# NUDGE: Lightweight Non-Parametric Fine-Tuning of Embeddings for Retrieval

**Sepanta Zeighami**
UC Berkeley
zeighami@berkeley.edu

**Zac Wellmer**
zac@1984.ai

**Aditya Parameswaran**
UC Berkeley
adityagp@berkeley.edu

## Abstract

$k$-Nearest Neighbor search on dense vector embeddings ($k$-NN retrieval) from pre-trained embedding models is the predominant retrieval method for text and images, as well as Retrieval-Augmented Generation (RAG) pipelines. In practice, application developers often fine-tune the embeddings to improve their accuracy on the dataset and query workload in hand. Existing approaches either fine-tune the pre-trained model itself or, more efficiently, but at the cost of accuracy, train adaptor models to transform the output of the pre-trained model. We present NUDGE, a family of novel *non-parametric* embedding fine-tuning approaches that are significantly more accurate and efficient than both sets of existing approaches. NUDGE directly modifies the embeddings of data records to maximize the accuracy of $k$-NN retrieval. We present a thorough theoretical and experimental analysis of NUDGE's non-parametric approach. We show that even though the underlying problem is NP-Hard, constrained variations can be solved efficiently. These constraints additionally ensure that the changes to the embeddings are modest, avoiding large distortions to the semantics learned during pre-training. In experiments across five pre-trained models and nine standard text and image retrieval datasets, *NUDGE runs in minutes and often achieves NDCG@10 at least 10% higher than existing fine-tuning methods. On average, NUDGE provides 3.3× and 4.3× higher increase in accuracy and runs 200× and 3× faster, respectively, over fine-tuning the pre-trained model and training adaptors.*

## 1 Introduction

$k$-Nearest Neighbor search on dense vector embeddings ($k$-NN retrieval) from pre-trained embedding models is the de-facto standard for text and image retrieval, as well as in Retrieval-Augmented Generation (RAG) pipelines. (Lewis et al., 2020; Li et al., 2023a; Gao et al., 2023; Patil et al., 2023; Du et al., 2022; Liu et al., 2021b). Given $n$ data records (e.g., text chunks or images), $k$-NN retrieval embeds them using a pre-trained model as $d$-dimensional vectors in $\mathcal{R}^d$. To answer a query, it similarly embeds the query in $\mathcal{R}^d$ and retrieves the top-$k$ data records whose embeddings have the highest cosine similarity (or inner product) with the query embedding. By simply performing a top-$k$ look-up (often through vector databases), the simplicity and efficiency of $k$-NN retrieval has made it increasingly popular and often preferred to other retrieval paradigms, e.g., late interaction (Khattab & Zaharia, 2020; Santhanam et al., 2021) or generative retrieval (Tay et al., 2022; Wang et al., 2022). However, the out-of-the-box pre-trained model is often not sufficiently accurate on the dataset or queries in hand, and fine-tuning is typically used to improve the accuracy.

There are two standard approaches to fine-tuning embeddings: fine-tuning the pre-trained models directly (referred to henceforth as *PTFT*) or training *adaptor models* on top of the pre-trained models (Zhao et al., 2024; Zhou et al., 2024; Aarsen, 2024; Suvansh Sanjeev, 2024; LlamaIndex, 2024b). PTFT can be more accurate but comes with practical challenges: it (1) requires access to the model parameters and must rely on third-party APIs for closed-source models [1], (2) is computationally expensive, and (3) incurs further hosting and maintenance costs for deployment of the fine-tuned model. An alternative is to learn an *Adaptor*, $\hat{g}_\theta$, a transformation of the output of the (frozen) pre-trained model, $\hat{f}$, so that the function $\hat{g}_\theta \circ \hat{f}$ generates accurate data and/or query embeddings. Training $\hat{g}_\theta$ can be done *model-agnostically*, that is, with *only black-box* access to the pre-trained model, addressing (1). Moreover, Adaptors are typically small models, such as linear

---

[1]E.g., OpenAI currently does not provide an interface for fine-tuning embedding model (OpenAI, 2024b).

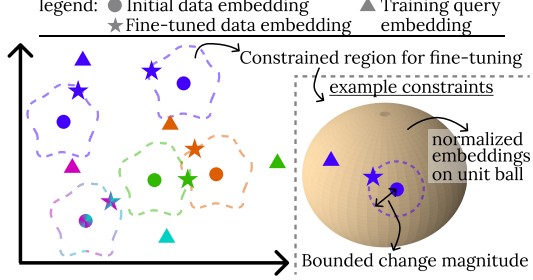

Figure 1: NUDGEs change data embeddings within a constrained region to maximize similarity with training queries. Data embeddings are colored based on queries for which they are the ground-truth answers.

Table 1: Comparison of fine-tuning methods on standard text datasets (PTFT refers to fine-tuning the pre-trained model). See Sec. 4 for experimental setting.

|  | **PTFT** | **Adaptors** | **NUDGE (ours)** |
|---|---|---|---|
| **Needs Model Parameters** | Yes | No | **No** |
| **Fine-Tuning Time (mins.)** | 447 | 7.99 | **2.18** |
| **Accuracy Boost (avg / max%)** | 3.8 / 15.6 | 2.9 / 12.4 | **12.4 / 24.8** |

models (LlamaIndex, 2024b), addressing (2), and lowering the associated costs in (3). Nonetheless, experimental results show, at best, modest accuracy gains from using Adaptors. *We, therefore, lack a fine-tuning approach for k-NN retrieval that is simultaneously effective, efficient, and easy-to-use.*

We present NUDGE, a family of approaches to *non-parametrically fine-tune embeddings efficiently*. NUDGE methods (or *NUDGEs*) are surprisingly effective, model-agnostic, and incur no additional deployment cost. NUDGEs take a novel non-parametric view of embedding fine-tuning: they view the embeddings themselves as parameters of the $k$-NN retrieval algorithm and directly modify the embeddings of data records to maximize the accuracy of $k$-NN retrieval. Although we show that the underlying optimization problem is NP-Hard in general, and can lead to overfitting if data embeddings are allowed to change arbitrarily, NUDGEs efficiently solve constrained variations of this problem formulated to avoid overfitting. As shown in Fig. 1, NUDGEs change each data embedding to maximize the similarity between the data embedding and the training queries to which the data record is a correct answer, while constraining how and by how much the embedding can change. Intuitively, the constraints allow for enough modifications to the embeddings to improve accuracy on the dataset in hand while avoiding large distortion that would offset the semantics learned during pre-training. Fig. 1 shows an example of the constrained region used, where new embeddings are constrained to be normalized (i.e., to fall on the unit ball), and the magnitude of changes to the embeddings to be bounded. NUDGEs solve the constrained optimization problems in closed form, presenting simple and effective update formulae for embedding fine-tuning.

NUDGE's *non-parametric* formulation represents a new paradigm in embedding fine-tuning. It contrasts with *parametric* approaches adopted by PTFT and Adaptors that increase the similarity between query and answer pairs through contrastive-type losses (Zhao et al., 2024). The non-parametric nature of NUDGE affords it more flexibility in optimization to improve data embeddings for which training queries are available, but avoids embedding changes for data records without associated training queries. Parametric methods, on the other hand, have less flexibility in optimization, but any improvement may generalize to data records that don't have associated training queries. This makes NUDGE particularly well-suited for *closed-world* settings where a relatively fixed dataset is queried multiple times, while parametric methods are advantageous when generalizing to new data records (e.g., in a rapidly changing dataset) without associated training queries. In the latter setting, we show that NUDGE can be combined with parametric methods to improve embeddings of data records both with and without associated training queries.

We present a thorough experimental evaluation, showing that NUDGEs significantly outperform Adaptors and PTFT on standard text retrieval datasets from BEIR (Thakur et al., 2021) and KILT (Petroni et al., 2021) and image retrieval datasets COCO (Lin et al., 2014) and Flickr (Young et al., 2014). NUDGEs obtain common accuracy metrics Recall@10 and NDCG@10 *up to 16.0% higher than both PTFT and Adaptors and 24.4% higher than no fine-tuning, consistently outperforming parametric methods across 9 different datasets and 5 different embedding models.* We also comprehensively evaluate out-of-distribution generalization of NUDGEs compared with parametric methods. We show that although NUDGEs, on their own, have limited out-of-distribution generalization ability, they can be combined with parametric methods to inherit their out-of-distribution characteristics while complementing them with significant in-distribution accuracy improvements. Overall, NUDGEs are model-agnostic and can be used with any (potentially closed-source) embedding

model, are trained in minutes on datasets with millions of records, and the fine-tuned embeddings can be used directly without any additional model inference cost.

## 2 PRELIMINARIES

**Notation**. We use bar $^-$ on top of letters to denote raw data/queries that are not embedded. We use boldface capital letters to denote matrices (mostly used to represent embeddings), e.g., $\boldsymbol{X} \in \mathcal{R}^{r \times s}$ is an $r \times s$ matrix, and use $\boldsymbol{X}_i$ to refer to the $i$-th row of the matrix, which is a vector, $\boldsymbol{X}_i \in \mathcal{R}^s$. $\|\boldsymbol{x}\|$ refers to the L2 norm of a vector $\boldsymbol{x}$, $\|\boldsymbol{X}\|$ refers to the vector of row-wise norms, $(\|\boldsymbol{X}_1\|, ..., \|\boldsymbol{X}_r\|)$, and $\frac{\boldsymbol{X}}{\|\boldsymbol{X}\|}$ refers to $\boldsymbol{X}$ with normalized rows. We use $[s]$ to refer to the set $\{1, 2, ..., s\}$.

$k$**-Nearest Neighbor Retrieval**. Given a dataset, $\bar{D}$ of $n$ records (e.g., text chunks or images), $k$-NN retrieval first embeds the data records using an embedding model, $\mathcal{E}_D$, to generate an embedding matrix $\boldsymbol{D} \in \mathcal{R}^{n \times d}$ where $d$ is the embedding dimensionality. The $i$-th row, $\boldsymbol{D}_i$, of $\boldsymbol{D}$ is the embedding of the $i$-th record, $\bar{D}_i$, that is, $\boldsymbol{D}_i = \mathcal{E}_D(\bar{D}_i)$. To answer a query $\bar{q}$, $k$-NN retrieval first embeds the query using an embedding model, $\mathcal{E}_Q$ (often $\mathcal{E}_Q$ and $\mathcal{E}_D$ are the same, but can be different, especially in a multi-modal setting) to obtain $\boldsymbol{q} = \mathcal{E}_Q(\bar{q})$. $k$-NN retrieval then returns the top-$k$ records in $\bar{D}$ whose embeddings have the highest similarity to $\boldsymbol{q}$, measured by either the inner product or cosine similarity, i.e., the $k$ records $\bar{D}_{i_1}, ..., \bar{D}_{i_k}$ that correspond to the $k$ highest values in the set $\{\boldsymbol{D}_1 \cdot \boldsymbol{q}, ..., \boldsymbol{D}_n \cdot \boldsymbol{q}\}$ when using the inner product similarity metric. By default, we use the inner product, which can be applied to normalized embeddings to obtain cosine similarity.

**Ground-Truth Answers**. Queries can require retrieving multiple data records, and each data record can have a different degree of relevance to the query. For simplicity, here, we present our results in the setting where a query, $\bar{q}$, requires retrieving a single ground-truth data record. We refer to the ground-truth data record for a query, $\bar{q}$, as the *ground-truth answer* to the query and often refer to the data record with its index $y$, $y \in [n]$. Extensions to multiple ground-truth data records (each with potentially different degrees of relevance to the query) is straightforward and presented in Appx. D.

**Fine-Tuning**. Fine-tuning aims to improve retrieval accuracy for the dataset $\bar{D}$ through optimizing embeddings. We let $\boldsymbol{D}^* \in \mathcal{R}^{n \times d}$ denote the fine-tuned data embeddings obtained after fine-tuning, where $\boldsymbol{D}_i^* \in \mathcal{R}^d$ is the fine-tuned embedding for the $i$-th data record. We consider a supervised setting where a query set with corresponding ground-truth answers is available, consisting of the query set, $\bar{Q}$, and a set, $Y$, of ground-truth answers, where $Y_j$ is the index of the ground-truth answer for the $j$-th query, $\bar{q}_j$. We split this set into two, a training set $\bar{Q}^T, Y^T$ and a validation set $\bar{Q}^V, Y^V$, with $n_T$ and $n_V$ queries, respectively. Let $\boldsymbol{Q}^T \in \mathcal{R}^{n_T \times d}$ and $\boldsymbol{Q}^V \in \mathcal{R}^{n_V \times d}$ be matrices containing embeddings for training and validation queries. The training set can be collected over time from user interactions with the system, by collecting labels, or by generating synthetic training data using LLMs (e.g., LlamaIndex (2024a); Meng et al. (2022)).

## 3 NON-PARAMETRIC EMBEDDING FINE-TUNING

Our *NUDGE* approach views embeddings as parameters of the $k$-NN retrieval algorithm, optimizing the embeddings directly to improve retrieval accuracy. In Sec. 3.1, we formalize the notion of non-parametric embedding fine-tuning by stating two optimization problems, one directly maximizing a retrieval accuracy metric and one maximizing similarity between queries and their ground-truth answers. Maximizing accuracy is the final goal, but similarity is a simpler surrogate to optimize in practice. Nonetheless, we show that the former is NP-hard and the latter is unbounded (i.e., the objective can be improved indefinitely). Moreover, since both optimization problems allow embeddings to arbitrarily change (i.e., are unconstrained), directly solving either can lead to overfitting. In Sec. 3.2, we present NUDGE, a family of approaches that solve a combination of constrained variations of the two optimization problems to address generalization and efficiency challenges.

### 3.1 UNCONSTRAINED NON-PARAMETRIC EMBEDDING FINE-TUNING PROBLEMS

Let $\boldsymbol{\Delta} \in \mathcal{R}^{n \times d}$ be the modification to be learned to the embeddings, so that its $i$-th row, $\boldsymbol{\Delta}_i$, is the modification to $\boldsymbol{D}_i$. That is, after fine-tuning, the final embedding is $\boldsymbol{D}^* = \boldsymbol{D} + \boldsymbol{\Delta}$. We use $\boldsymbol{Q}$ and $Y$ to refer to a generic query embedding matrix with corresponding ground-truth answers containing $N$ queries. $\boldsymbol{Q}$ and $Y$ can respectively be either $\boldsymbol{Q}^T$ and $Y^T$ or $\boldsymbol{Q}^V$ and $Y^V$. Recall that $Y_i$ is the index of the ground-truth answer to the $i$-th query, so that $\boldsymbol{D}_{Y_i}$ and $\boldsymbol{\Delta}_{Y_i}$ are $d$-dimensional vectors, respectively, referring to the data embedding of the ground-truth answer to $i$-th query, and the modification to be learned to the data embedding of the ground-truth answer to $i$-th query.

**MaxA-EFT**. *Maximum Accuracy Embedding Fine-Tuning Problem, MaxA-EFT*, is the problem of finding $\boldsymbol{\Delta}$ to fine-tune data embeddings that maximizes the number of queries in $\boldsymbol{Q}$ answered correctly, formalized as follows. For a query embedding matrix $\boldsymbol{Q}$ with ground-truth answers $Y$, let $\mathcal{I}_i(\boldsymbol{\Delta})$, for $i \in [N]$, be the indicator variable denoting if the $i$-th query is *answered correctly after fine-tuning with* $\boldsymbol{\Delta}$. Formally, $\mathcal{I}_i(\boldsymbol{\Delta}) = 1$ if the following holds, and zero otherwise

$$\boldsymbol{Q}_i \cdot (\boldsymbol{D}_{Y_i} + \boldsymbol{\Delta}_{Y_i}) > \boldsymbol{Q}_i \cdot (\boldsymbol{D}_j + \boldsymbol{\Delta}_j), \quad \forall j \in ([n] \setminus Y_i), \tag{1}$$

Where $[n] \setminus Y_i$ is the set data records indices that are not a ground-truth answer to the $i$-th query.

**Problem 1** (MaxA-EFT). *MaxA-EFT is the problem of finding $\boldsymbol{\Delta}$ to maximize the number of queries answered correctly after fine-tuning with $\boldsymbol{\Delta}$, i.e.,* $\arg\max_{\boldsymbol{\Delta} \in \mathcal{R}^{n \times d}} \sum_{i \in [N]} \mathcal{I}_i(\boldsymbol{\Delta})$.

**Theorem 1.** *MaxA-EFT is NP-Hard.*

Theorem 1 is proved by reduction from the Maximum Feasible Linear Subsystem problem as studied in Amaldi & Kann (1995), see Appx. B.1. Apart from the NP-hardness, MaxA-EFT allows data embeddings to be arbitrarily changed by $\boldsymbol{\Delta}$. This change can distort the semantics captured in $\boldsymbol{D}$ by the pre-trained model, and lead to poor generalization to queries outside of $\boldsymbol{Q}$.

**MaxS-EFT**. An alternative formulation is *Maximum Similarity Embedding Fine-Tuning Problem,*

$$\arg\max_{\boldsymbol{\Delta} \in \mathcal{R}^{n \times d}} \sum_{i \in [N]} \boldsymbol{Q}_i \cdot (\boldsymbol{D}_{Y_i} + \boldsymbol{\Delta}_{Y_i}), \tag{2}$$

referred to as *MaxS-EFT*. Here, we change data embeddings to maximize the similarity between queries and ground-truth answers, a standard optimization objective (e.g., Henderson et al. (2017), SentenceTransformers (2024b)). However, the non-parametric formulation makes Eq. 2 an unconstrained optimization problem with a linear objective, so the problem is unbounded and has no optimal solution. Moreover, setting $\boldsymbol{\Delta}_{Y_i}$ so $\boldsymbol{Q}_i \cdot \boldsymbol{\Delta}_{Y_i} > 0$ and increasing the magnitude of $\boldsymbol{\Delta}_{Y_i}$ arbitrarily improves the objective, yielding trivial solutions with poor generalization to unseen queries.

## 3.2 NUDGE APPROACHES

Because of the potential for overfitting and the computational challenges due to NP-hardness, we do not solve either MaxA-EFT or MaxS-EFT directly. Instead, we introduce NUDGE, a family of approaches that solve constrained variations of MaxA-EFT and MaxS-EFT, designed to avoid overfitting, while being efficient. We discuss two main approaches, NUDGE-M and NUDGE-N, in Secs. 3.2.1 and 3.2.2 and present other practical extensions in Appx. C.2.

### 3.2.1 NUDGE-M: NUDGE WITH BOUNDED MAGNITUDE

NUDGE-M solves MaxS-EFT on the training set, but with the added constraint $\|\boldsymbol{\Delta}_i\| \leq \gamma, \forall i \in [n]$, for a scalar $\gamma \geq 0$. $\gamma$ controls how much each embedding can change during fine-tuning. NUDGE-M sets $\gamma$ by solving MaxA-EFT on the validation set. Intuitively, this approach (1) changes data embeddings to maximize the similarity between embeddings and queries on the training set, (2) ensures that the magnitude of the changes to the embeddings is bounded to avoid overfitting, and (3) decides how much the embeddings are allowed to change by maximizing validation accuracy. NUDGE-M provides a closed-form solution to do this. We provide an overview of the solution here and leave the details and formal proofs to Appx. B.2.

**Optimization Formulation**. Define `MaxS-M`$(\gamma)$, for a scalar $\gamma \geq 0$, as the set of optimal solutions to the following constrained version of MaxS-EFT with *bounded magnitude*:

$$\texttt{MaxS-M}(\gamma) = \arg\max \sum_{i \in [n_T]} \boldsymbol{Q}_i^T \cdot (\boldsymbol{D}_{Y_i^T} + \boldsymbol{\Delta}_{Y_i^T})$$

$$\text{s.t. } \boldsymbol{\Delta} \in \mathcal{R}^{n \times d}, \|\boldsymbol{\Delta}_i\| \leq \gamma \quad \forall i \in [n].$$

A $\boldsymbol{\Delta} \in$ `MaxS-M`$(\gamma)$ changes data embeddings by at most $\gamma$ while maximizing the similarity between training queries and their ground-truth answer. We set $\gamma$ to maximize the validation accuracy after fine-tuning with $\boldsymbol{\Delta} \in$ `MaxS-M`$(\gamma)$:

$$\max \sum_{i \in [n_V]} \mathcal{I}_i^V(\boldsymbol{\Delta})$$

$$\text{s.t. } \boldsymbol{\Delta} \in \texttt{MaxS-M}(\gamma), \gamma \geq 0.$$

$\mathcal{I}_i^V(\boldsymbol{\Delta})$ denotes $\mathcal{I}_i$ (see Eq. 1) on the validation set $\boldsymbol{Q}^V, Y^V$, so $\sum_{i \in [n_V]} \mathcal{I}_i^V(\boldsymbol{\Delta})$ is the validation accuracy after fine-tuning with $\boldsymbol{\Delta}$. This problem is referred to as *Bi-level Maximization with bounded Magnitude, BiMax-M*. We denote the optimal solution to BiMax-M by $\boldsymbol{\Delta}^M$.

**NUDGE-M**. NUDGE-M is an algorithm that optimally solves BiMax-M:

**Theorem 2.** *There exists an algorithm, referred to as NUDGE-M, that optimally solves BiMax-M in $O(n_V(nd + \log n_V) + n_T d)$. Specifically, NUDGE-M sets $\boldsymbol{\Delta}^M$ as*

$$\boldsymbol{\Delta}_i^M = \gamma^* \frac{\boldsymbol{G}_i}{\|\boldsymbol{G}_i\|}, \quad where \ \ \boldsymbol{G}_i = \sum_{j \in [n_T]} \mathbb{I}[i = Y_j^T] \boldsymbol{Q}_j^T, \quad \forall i \in [n], \tag{3}$$

*and $\mathbb{I}$ is the indicator function with a predicate argument, for an optimally chosen scalar $\gamma^*$.*

Eq. 3 presents the simple update rule used by NUDGE-M to fine-tune data embeddings, using which the fine-tuned embeddings are computed as $\boldsymbol{D}^* = \boldsymbol{D} + \boldsymbol{\Delta}^M$. Observe that $\boldsymbol{G}_i$ is the sum of query embeddings whose ground-truth answer is $\boldsymbol{D}_i$, so that data embeddings are moved towards the queries for which they are the ground-truth answers. We also note that $O(n_T d)$ is the complexity of a single iteration over training data, and $O(n_V nd)$ is the complexity of calculating validation accuracy once. Thus, ignoring the log term, the above time complexity is equal to a single training and validation iteration for parametric approaches (i.e., Adaptors or PTFT), and has smaller constant factors since it does not perform any model forward passes.

The optimal $\gamma^*$ in Eq. 3 is calculated by solving linear inequalities resulting from the definition of $\mathcal{I}_i^V$. We provide an overview here and leave the details to Appx. B.2. First, note that $\texttt{MaxS-M}(\gamma) = \gamma \frac{\boldsymbol{G}}{\|\boldsymbol{G}\|}$, found using the KKT points of the optimization problem. Thus, BiMax-M reduces to finding a $\gamma$ that maximizes $\sum_{i \in [n_V]} \mathcal{I}_i^V(\gamma \frac{\boldsymbol{G}}{\|\boldsymbol{G}\|})$. For each $i \in [n_V]$, substituting $\boldsymbol{\Delta} = \gamma \frac{\boldsymbol{G}}{\|\boldsymbol{G}\|}$ into the definition of $\mathcal{I}_i^V(\boldsymbol{\Delta})$ in Eq. 1, we have $\mathcal{I}_i^V(\gamma \frac{\boldsymbol{G}}{\|\boldsymbol{G}\|}) = 1$ if the following holds, and zero otherwise:

$$\boldsymbol{Q}_i^V \cdot (\boldsymbol{D}_{Y_i} + \gamma \frac{\boldsymbol{G}_{Y_i^V}}{\|\boldsymbol{G}_{Y_i^V}\|}) > \boldsymbol{Q}_i^V \cdot (\boldsymbol{D}_j + \gamma \frac{\boldsymbol{G}_j}{\|\boldsymbol{G}_j\|}), \quad \forall j \in [n] \setminus Y_i^V. \tag{4}$$

Eq. 4 is a set of inequalities in $\gamma$. Denote the solution to the inequalities by $I_i$, so that $\gamma \in I_i$ if and only if $\mathcal{I}_i^V(\gamma \frac{\boldsymbol{G}}{\|\boldsymbol{G}\|}) = 1$. Since the inequalities are linear in $\gamma$, $I_i$ is an interval in $\mathcal{R}$. Consequently,

$$\gamma^* = \arg \max_{\gamma} \sum_{i \in n_V} \mathbb{I}[\gamma \in I_i], \tag{5}$$

and the solution to Eq. 5 is a $\gamma$ that intersects the most number of intervals across all $I_i$, $i \in [n_V]$, which can be found by a single iteration over the intervals after sorting their start and end points.

### 3.2.2 NUDGE-N: NUDGE WITH NORMALIZED EMBEDDINGS

NUDGE-N additionally constrains the norm of the fine-tuned embedding. This constraint serves as an additional regularization, helping with out-of-distribution generalization (see Sec.4.2).

**Optimization Formulation**. Analogous to $\texttt{MaxS-M}(\gamma)$ but with an added constraint, we define $\texttt{MaxS-N}(\gamma)$ as the optimal solution to the following optimization problem:

$$\texttt{MaxS-N}(\gamma) = \arg \max \sum_{i \in [n_T]} \boldsymbol{Q}_i^T \cdot (\boldsymbol{D}_{Y_i^T} + \boldsymbol{\Delta}_{Y_i^T})$$

$$\text{s.t. } \boldsymbol{\Delta} \in \mathcal{R}^{n \times d}, \|\boldsymbol{\Delta}_i\|^2 \leq \gamma, \|\boldsymbol{D}_i + \boldsymbol{\Delta}_i\| = 1 \quad \forall i \in [n].$$

To find a suitable $\gamma$, we solve the same optimization problem as BiMax-M, except that we replace $\texttt{MaxS-M}(\gamma)$ with $\texttt{MaxS-N}(\gamma)$. We call this problem *Bi-level Maximization with Normalized embeddings, BiMax-N*, and refer to its optimal solution as $\boldsymbol{\Delta}^N$.

**NUDGE-N**. NUDGE-N is an algorithm that optimally solves BiMax-N:

**Theorem 3.** *There exists an algorithm, referred to as NUDGE-N, that optimally solves BiMax-N in $O(n_V(nd + \log n_V) + n_T d)$. Specifically, NUDGE-N sets $\boldsymbol{\Delta}^N$ as*

$$\boldsymbol{\Delta}_i^N = \begin{cases} \frac{\boldsymbol{G}_i}{\|\boldsymbol{G}_i\|} - \boldsymbol{D}_i, & \text{if } \frac{\boldsymbol{G}_i \cdot \boldsymbol{D}_i}{\|\boldsymbol{G}_i\|} \geq 1 - \frac{\gamma^*}{2} \\ \frac{\sqrt{\gamma^*(4-\gamma^*)}}{2} \boldsymbol{Z}_i - \frac{\gamma^*}{2} \boldsymbol{D}_i, & \text{otherwise,} \end{cases}, \quad \boldsymbol{Z}_i = \frac{\boldsymbol{G}_i - (\boldsymbol{D}_i \cdot \boldsymbol{G}_i) \boldsymbol{D}_i}{\|\boldsymbol{G}_i - (\boldsymbol{D}_i \cdot \boldsymbol{G}_i) \boldsymbol{D}_i\|} \tag{6}$$

*and $\boldsymbol{G}_i$ is as defined in Eq. 3, for an optimally chosen scalar $\gamma^*$.*

Eq. 6 is the update rule used by NUDGE-N to fine-tune data embeddings, using which the fine-tuned embeddings are obtained as $\boldsymbol{D}^* = \boldsymbol{D} + \boldsymbol{\Delta}^N$. This update moves the data embedding on the unit ball (to satisfy $\|\boldsymbol{D}_i + \boldsymbol{\Delta}_i\| = 1$) between $\boldsymbol{D}_i$ and $\frac{\boldsymbol{G}_i}{\|\boldsymbol{G}_i\|}$, where $\frac{\boldsymbol{G}_i}{\|\boldsymbol{G}_i\|}$ is the normalized sum of embeddings of queries whose ground-truth answer is $\boldsymbol{D}_i$. $\gamma^*$ determines how much to move the embedding, and $\boldsymbol{Z}_i$ determines the direction. $\boldsymbol{Z}_i$ is the normalized projection of $\boldsymbol{G}_i$ onto the tangent plane of the unit ball at $\boldsymbol{D}_i$, so that moving in the direction of $\boldsymbol{Z}_i$ maximally increases `MaxS-N` objective.

Finding the optimal $\gamma^*$ is similar to NUDGE-M. We first use KKT to solve `MaxS-N`$(\gamma)$ and substitute the resulting $\boldsymbol{\Delta}$ into the definition of $\mathcal{I}_i^V$ in Eq. 1. The optimal $\gamma^*$ is then found as the value that satisfies the maximum number of the resulting inequalities. The resulting inequalities are, in this case, quadratic (compared with linear for NUDGE-M) but can still be solved in closed-form. Nevertheless, from a practical perspective, solving the quadratic equations is tedious as it requires considering various special cases. We observed that performing a grid search to find $\gamma^*$ that maximizes validation accuracy finds accurate solutions and is almost as efficient (experiments in Sec. 4.3 show NUDGE-N is not very sensitive to the value of $\gamma^*$). We use this implementation in practice.

### 3.3 EXTENSIONS AND ALTERNATIVES

**Combining NUDGE and Parametric Methods for Generalization**. NUDGE's non-parametric methodology distinguishes between data records that have associated queries at training time (referred to as in-distribution labels) and data records that don't (referred to as out-of-distribution labels). NUDGE is designed to improve embeddings of in-distribution labels, and does not modify the embedding of data records that don't appear in the training set. In contrast, parametric methods such as PTFT and Adaptor modify the embedding function and have the ability to generalize and improve embeddings of out-of-distribution labels as well. This makes NUDGE well-suited for *closed-world* settings where data records are relatively fixed, and generalization to new records is not needed. On the other hand, when generalization is needed, NUDGE can be combined with parametric methods to provide the best of both worlds: improving accuracy for queries with in-distribution labels through NUDGE and on queries with out-of-distribution labels through parametric methods. Sec. 4.2 empirically studies generalization ability of NUDGE when combined with parametric methods.

**Alternative Constraint Formulations**. Constraints are an important part of NUDGE, controlling how embeddings are updated. The L2 norm constraints in NUDGE-M and NUDGE-N are chosen since we focus on dense embeddings and bounding L2 norm leads to dense embedding updates. However, other constraints are possible, for example, using L1 norm instead of L2 norm to create sparse updates or using adaptive thresholds in NUDGE-M and NUDGE-N to increase flexibility in fine-tuning. We studied both such approaches, deriving fine-tuning update rule using L1 norm in Appendix C.1 and empirically evaluating both use of L1 norm and adaptive thresholds in Appendix E.6. We observed limited benefits from using such alternative constraint formulations.

## 4 EXPERIMENTS

We present results on standard text and image retrieval benchmarks and multiple pre-trained models. For the sake of space, additional detailed results and ablation studies are deferred to Appx. E.

**Datasets**. For text retrieval datasets we use 7 standard datasets: SciFacts (Wadden et al., 2020), Fever (Fever, 2024), ArguAna (Arguana, 2024) (we use their BEIR (Thakur et al., 2021) versions), TriviaQA Joshi et al. (2017), HotpotQA (Yang et al., 2018), and Natural Questions(Kwiatkowski et al., 2019) (we use their KILT (Petroni et al., 2021) versions), and NF-Corpus (Boteva et al., 2016) (although all datasets have a BEIR version, we use non-BEIR versions whenever that is larger). We use the datasets as is, without any preprocessing step, except for datasets from KILT, where we only use Wikipedia pages that contain an answer to at least one query (i.e., pages where we expect fine-tuning to have an impact) due to computational constraints (embedding and storing the embeddings for all Wikipedia pages requires significant computational resources and is unlikely to impact the results). For image retrieval, we use COCO (Lin et al., 2014) (we use the dataset from 2014) and Flickr (Young et al., 2014) datasets. We use image captions as queries to retrieve the corresponding image. For all text and image datasets, we use 0.7-0.1-0.2 train, validation and test split, but limit test and validation sizes to at most 10,000 queries if there is more. Statistics about data and query size are presented in Table 2.

**Pre-Trained Models**. We report results on fine-tuning embeddings for 5 different pre-trained models, 3 for text and 2 for image retrieval, where we consider models of different sizes and embedding dimensions. For text, we use BGE-small (Xiao et al., 2023) with 33M parameters and embedding

Table 2: Total number of queries and records in the datasets

|  | NF-Corpus | SciFact | ArguAna | Fever | NQ | TriviaQA | HotpotQA | COCO | Flickr |
|---|---|---|---|---|---|---|---|---|---|
| **Query** | 2,429 | 1,109 | 1,401 | 123,142 | 79,782 | 58,245 | 74,259 | 414,113 | 155,070 |
| **Data** | 3,633 | 5,183 | 8,674 | 5,416,568 | 7,631,395 | 7,631,395 | 7,631,395 | 82,783 | 31,014 |

Table 3: Average results across text datasets grouped by the embedding model used

| Emb. Model → | BGE-S | | | GTE-L | | | TE3-L | | |
|---|---|---|---|---|---|---|---|---|---|
| ↓ Method | R@1 | R@10 | NDCG@10 | R@1 | R@10 | NDCG@10 | R@1 | R@10 | NDCG@10 |
| NUDGE-M | 49.7 | 66.6 | 57.0 (+8.4) | 52.1 | 73.4 | 61.0 (+7.7) | 54.1 | **75.6** | 63.2 (+11.0) |
| NUDGE-N | **52.0** | **72.6** | **61.1 (+12.4)** | **53.4** | **74.8** | **62.7 (+9.4)** | **55.2** | **76.0** | **63.9 (+11.7)** |
| Adapter | 39.5 | 65.5 | 51.6 (+2.9) | 45.1 | 68.4 | 55.7 (+2.4) | 46.9 | 66.2 | 54.4 (+2.2) |
| PTFT | 40.9 | 66.1 | 52.5 (+3.8) | N/A | N/A | N/A | N/A | N/A | N/A |
| No Fine-Tuning | 37.0 | 62.4 | 48.7 | 41.6 | 67.0 | 53.3 | 40.0 | 67.5 | 52.2 |

dimension 384, GTE-large (Li et al., 2023b) with 434M parameters and embedding dimension 1024 and OpenAI's text-embedding-large-3 (OpenAI, 2024a), a closed-source model with embedding dimensions 3072. The three models are respectively referred to as BGE-S, GTE-L and TE3-L. For image retrieval, we use two CLIP variants (Radford et al., 2021), ViT-B/32 and ViT-L/14, which have, respectively, 151M and 427M parameters and 512 and 768 embedding dimensions. We respectively call them CLIP-B and CLIP-L.

**Baselines**. We report results using the embeddings without fine-tuning, called No Fine-Tuning, in addition to training Adaptors and fine-tuning the pre-trained model, referred to as PTFT (see Appx. E.1 for implementation details). Due to computational constraints, we report results for the latter only for the small open-source model BGE-S. For both, we present two versions. By default, we use the Multiple Negative Ranking (MNR) loss (Henderson et al., 2017) for training, which is the standard contrastive fine-tuning loss when positive query/answer pairs are available, suggested by SentenceTransformers (SentenceTransformers, 2024a) and used by LlamaIndex (LlamaIndex, 2024d) for fine-tuning (although LlamaIndex only uses a single positive example per query (LlamaIndex, 2024e)). Despite our hyperparameter tuning effort, we observed no accuracy improvements on some datasets through fine-tuning with this loss (see Table 5). We then modified the loss, so that only negative samples whose cosine similarity is at least equal to a threshold are included in the loss (see E.1 for details), which improved accuracy on some datasets but worsened it on other. We use the suffix -L to denote the baselines using this modified loss. We report results using this loss to gain more insight into the baseline's behavior. We use cosine similarity as the retrieval distance metric for No Fine-Tuning, Adaptor, and PTFT.

**Metrics**. We report typical metrics Recall@$k$ (R@$k$ for short) and NDCG@$k$ with $k = 10$ by default, following MTEB benchmark (Muennighoff et al., 2022). We also report Recall@1, the validation metric, and the change in NDCG@$k$ compared with No Fine-Tuning in parenthesis (+/-).

### 4.1 BASELINE RESULTS

**Summary of results**. Tables 3-4 present our accuracy results averaged across all text or image datasets for different models. Table 5 presents the per dataset results for BGE-S. The per dataset results for other models followed similar trends and are deferred to Appx. E.2.

As Tables 3-4 show, both NUDGE-M and NUDGE-N provide significant accuracy gains, providing up to 14.3% NDCG@10 boost over No Fine-Tuning while PTFT and Adaptor only improve NDCG@10 up to 4.0%, when averaged across datasets, for both text and image retrieval. Interestingly, the accuracy gains from NUDGE-M and NUDGE-N depends on the pre-trained model. GTE-L outperforms TE3-L without fine-tuning, but using NUDGE TE3-L outperforms GTE-L.

Moreover, Table 8 shows the total fine-tuning time to run BGE-S on our text datasets (i.e., time obtain the associated results in Table 3) using an Nvidia A100 GPU as well as using 32 core Intel Broadwell CPUs. The reported time excludes the time to embed the data records (which is the same across all methods). NUDGE variants run in 1-2 minutes with GPU and in less than 11 minutes using CPUs, which is, respectively, more than 3 and 11 times faster than Adaptor and more than 200 times faster than PTFT, which cannot be run on CPUs in a reasonable time-frame. For both Adaptor and PTFT, the reported run times are for optimized implementations that include early stopping and optimizations for efficient calculation of validation accuracy (see Appx. E.1).

Table 4: Average results across image datasets grouped by the embedding model used

| Emb. Model → | CLIP-B | | | CLIP-L | | |
|---|---|---|---|---|---|---|
| ↓ Method | R@1 | R@10 | NDCG@10 | R@1 | R@10 | NDCG@10 |
| NUDGE-M | **28.6** | **55.1** | **40.9 (+14.1)** | **30.1** | **58.1** | **43.2 (+10.7)** |
| NUDGE-N | **28.8** | **55.3** | **41.1 (+14.3)** | **30.1** | **58.2** | **43.3 (+10.8)** |
| Adapter | 18.5 | 43.5 | 29.9 (+3.0) | 24.1 | 50.9 | 36.5 (+4.0) |
| No Fine-Tuning | 15.9 | 40.1 | 26.9 | 20.5 | 46.5 | 32.5 |

Table 5: NDCG@10 results for using BGE-S on text datasets

| Method | ArguAna | Fever | HotpotQA | NF-Corpus | NQ | SciFact | TriviaQA |
|---|---|---|---|---|---|---|---|
| NUDGE-M | **47.9** (+0) | **95.0 (+13.7)** | 63.1 (+5.0) | 40.0 (+6.1) | 38.6 (+17.5) | 79.2 (+6.5) | 35.5 (+9.6) |
| NUDGE-N | **47.9** (+0) | 93.5 (+12.3) | **65.2 (+7.1)** | 44.6 (+10.7) | 45.9 (+24.8) | **87.8 (+15.1)** | **43.0 (+17.2)** |
| Adapter | **47.9** (+0) | 81.2 (+0) | 58.1 (+0) | 37.4 (+3.5) | 27.4 (+6.2) | 83.5 (+10.9) | 25.9 (+0) |
| Adapter-L | **47.9** (+0) | 88.4 (+7.2) | 62.7 (+4.6) | 35.7 (+1.9) | 33.6 (+12.0) | 85.1 (+12.4) | 27.2 (+1.2) |
| PTFT | **47.9** (+0) | 81.2 (+0) | 58.1 (+0) | 46.1 (+12.2) | 28.1 (+7.0) | 80.4 (+7.7) | 25.9 (+0) |
| PTFT-L | **47.9** (+0) | 84.1 (+2.9) | 62.1 (+4.0) | 36.0 (+2.2) | 36.8 (+15.6) | 72.7 (+0) | 26.0 (+0.1) |
| No Fine-Tuning | **47.9** | 81.2 | 58.1 | 33.8 | 21.2 | 72.7 | 25.9 |

**Detailed Results**. Table 5 shows the detailed retrieval accuracy on text datasets for BGE-S. NUDGE significantly outperforms parametric methods on almost all datasets. NUDGE-N outperforms NUDGE-M on most datasets, showing the benefit of constraining embeddings to be normalized.

The results show that parametric approaches are unreliable and fail to provide significant accuracy improvements despite requiring significantly more computational resources. The reported results are after hyperparameter tuning as well as tweaking the loss function (i.e., -L variants). We observed that the latter does help improve (and sometimes worsen) accuracy for parametric approaches on some datasets, with Adaptor-L and PTFT-L providing accuracy improvements on Fever and HotpotQA where Adaptor and PTFT provided none. Meanwhile, NUDGE consistently provides significant accuracy boosts *without any hyperparameter tuning*. We provide a detailed discussion of the failure modes of the parametric approaches in Appx. E.3.

**Impact of label distribution**. Across datasets, we see NUDGE-M and NUDGE-N perform similarly on some datasets, while NUDGE-N outperforms NUDGE-M in other settings. As we discuss in details in Appendix E.12, difference between NUDGE-N and NUDGE-M can be explained based on the label distribution. Specifically, NUDGE-M performs worse on datasets where many queries ask for labels that are not seen during training (e.g., NQ, HotpotQA, SciFact), while it performs better when label ditribution is skewed towards a fixed set of labels (e.g., Fever, CoCo and Flickr).

## 4.2 OUT-OF-DISTRIBUTION GENERALIZATION

Next, we study the impact of fine-tuning on out-of-distribution queries and labels. To study impact of out-of-distribution queries, separately for each dataset, we use K-means to cluster all the queries in two clusters, referred to as $C_1$ and $C_2$. We split $C_1$ into 3 sets, $C_1^{\text{train}}$, $C_1^{\text{val}}$ and $C_1^{\text{test}}$, where $C_1^{\text{train}}$ and $C_1^{\text{val}}$ are used for training and validation. We report test results on $C_1^{\text{test}}$ as in-distribution and on $C_2$ as out-of-distribution results. To study impact of out-of-distribution labels, we split the test set into two sets. We define queries with out-of-distribution labels as queries whose ground-truth answer contains a data record not in the training set. We refer to the rest of the test set queries with in-distribution labels. We abbreviate in-distribution as InD and out-of-distribution as OoD. We also include results for A+NUDGE-N and P-NUDGE-N, which are methods that perform NUDGE-N on embeddings obtained after performing Adaptor and PTFT, respectively.

Average results across text datasets and with BGE-S model are shown in Table 6. As expected, NUDGE variants significantly outperform Adaptor and PTFT on both in-distribution queries and labels. Moreover, A+NUDGE-N and P+NUDGE-N provide the best results for OoD Label, showing that combining NUDGE-N with parametric methods helps NUDGE-N provide better generalization to unseen labels while preserving NUDGE-N's significant accuracy boosts in In-Distribution settings. However, surprisingly, we see little benefit to parametric methods in OoD settings, which is counter-intuitive as we expect such approaches to provide better generalization ability than NUDGE.

To investigate this further, we plot the per dataset generalization results for 3 datasets in Table 7 (results for other datasets is presented in Appendix E). First, we observe that Adapter and PTFT

Table 6: Generalization results using BGE-S. NDCG@10 average over text datasets

| Method | In-Dist. Query | Out-Of-Dist. Query | In-Dist. Label | Out-Of-Dist. Label |
|---|---|---|---|---|
| NUDGE-M | 57.6 (+8.8) | 36.6 (-11.6) | **72.5 (+24.7)** | 39.5 (-6.4) |
| NUDGE-N | **60.8 (+12.0)** | **49.9 (+1.7)** | **72.5 (+24.6)** | 46.8 (+1.0) |
| NUDGE-M+N | 58.7 (+9.9) | **49.8 (+1.6)** | 67.4 (+19.6) | 46.9 (+1.0) |
| A+NUDGE-N | 60.2 (+11.4) | 49.1 (+0.9) | **72.9 (+25.0)** | **47.3 (+1.5)** |
| P+NUDGE-N | **61.1 (+12.4)** | 49.2 (+1.0) | 72.0 (+24.1) | **47.6 (+1.7)** |
| Adapter | 51.0 (+2.3) | 47.7 (-0.5) | 52.5 (+4.6) | 46.6 (+0.8) |
| PTFT | 53.7 (+4.9) | 48.2 (0.0) | 53.9 (+6.0) | 46.0 (+0.1) |
| No Fine-Tuning | 48.8 | 48.2 | 47.9 | 45.9 |

Table 7: Generalization results using BGE-S for 3 datasets. Showing increase/decrease in NDCG@10 over No Fine-Tuning (IQ: InD Query, OQ: OoD Query, IL: InD Label, OL: OoD Label)

| Method | NQ | | | | NFCorpus | | | | SciFact | | | |
|---|---|---|---|---|---|---|---|---|---|---|---|---|
| | IQ | OQ | IL | OL | IQ | OQ | IL | OL | IQ | OQ | IL | OL |
| NUDGE-M | 23.7 | -19.2 | **63.6** | -17.4 | 6.2 | 0.0 | 8.0 | -11.8 | 6.0 | 0.0 | 12.6 | -2.7 |
| NUDGE-N | 28.6 | 1.5 | 60.6 | -2.5 | 9.1 | **4.2** | 12.7 | -2.7 | **12.5** | -0.5 | 25.9 | -1.4 |
| NUDGE-M+N | 28.5 | 1.4 | 60.8 | -2.8 | 8.4 | **4.4** | 8.7 | -3.3 | 3.9 | 0.0 | 6.5 | 0.0 |
| A+NUDGE-N | 29.8 | 3.4 | 62.3 | 1.5 | 9.3 | **4.2** | 12.1 | 2.5 | 7.3 | -8.0 | **27.5** | -7.1 |
| P+NUDGE-N | **33.6** | 1.9 | 56.0 | **6.5** | **11.0** | 1.7 | **18.0** | 3.1 | 8.2 | -2.9 | 20.3 | -10.1 |
| Adapter | 8.2 | **4.3** | 7.1 | 5.5 | 0.3 | 0.3 | 3.7 | **5.3** | 7.3 | -8.0 | 21.6 | -5.4 |
| PTFT | 15.0 | 0.8 | 7.9 | **6.3** | **11.0** | 1.7 | 14.2 | **5.0** | 8.0 | -2.9 | 19.7 | -10.3 |

do, in fact, improve OoD accuracy in NQ and NFCorpus datasets. However, the accuracy gains are offset by performance regression in SciFact dataset, where both Adaptor and PTFT show poor OoD generalization, performing much worse than No Fine-Tuning. We note that SciFact is the smallest dataset in our experiments with less than 1,000 training queries, making generalization particularly difficult. Overall, this shows that the ability of parametric approaches to generalize also comes with the risk of overfitting to the training set.

Regarding NUDGE-N, it is non-parametric and thus performs similar to No Fine-Tuning in OoD settings, while it provides significant accuracy boosts in InD settings (around 20% higher accuracy compared with Adaptor or PTFT on InD Label in Table 6). It misses out on the opportunity to generalize in NQ and NFCorpus (performing worse than Adaptor and PTFT in OoD settings in those datasets), but it also does not experience significant accuracy regression on SciFact (and performs better than Adaptor and PTFT on that dataset). For better generalization, NUDGE can be combined with Adaptor and PTFT to inherit their OoD generalization ability while achieving significantly higher accuracies in the InD settings, as evidenced by A+NUDGE-N and P+NUDGE-N methods.

Finally, we note that although NUDGE-M performs well on in-distribution samples, its performance deteriorates on OoD queries. To understand this, in Tables 6 and 7, we also included NUDGE-M+N which normalizes the embeddings obtained by NUDGE-M. We see that NUDGE-M+N avoids such an accuracy deterioration on OoD queries, suggesting that unnormalized embeddings are the main reason behind the poor OoD generalization of NUDGE-M. Indeed, when retrieving top-$k$ results using inner product similarity, fine-tuned embeddings with large magnitude can adversely impact the answer to queries that are far from fine-tuned data records, thus causing performance degradation on OoD queries (also see the discussion on the impact of label distribution in Sec. 4.1).

### 4.3 IMPACT OF $\gamma$

Fig. 2 shows the impact of $\gamma$ on NUDGE variants, obtained by using the update rule for NUDGE-N and NUDGE-M, respectively from Eq. 6 and Eq. 3 with different values of $\gamma$. Interestingly, for NUDGE-N, similar values of $\gamma$ work well across datasets, suggesting that $\gamma$ relates to inherent characteristics of the embedding space independent of the dataset. However, for NUDGE-M the best value of $\gamma$ changes more significantly across datasets. We further discuss in Appendix E.12 that label distribution can explain this behavior. For label distributions skewed towards a fixed set of labels large $\gamma$ does not hurt accuracy, but the opposite is observed on heavy tailed label distributions.

Table 8: Run time using BGE-S

| Method | Time GPU (mins.) | Time CPU (mins.) |
|--------|-----------------|-----------------|
| NUDGE-M | **1.14** | **7.12** |
| NUDGE-N | 2.18 | 11.0 |
| Adaptor | 7.99 | 77.8 |
| PTFT | 447 | N/A |

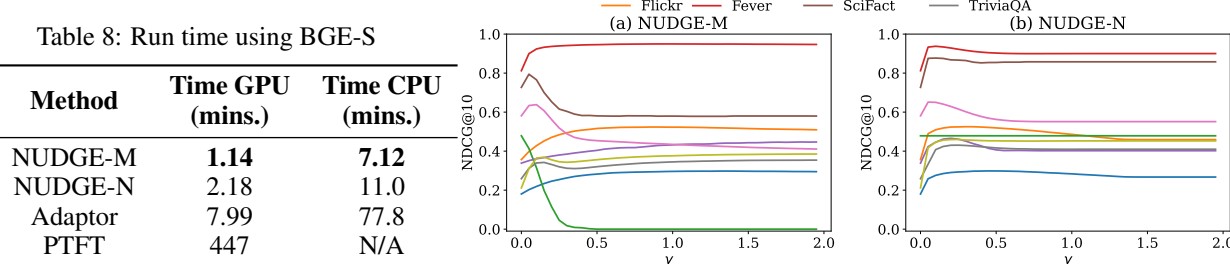

Figure 2: Impact of $\gamma$ on accuracy (BGE-S for text, CLIP-B for image)

## 5 RELATED WORK

Decades of research has explored various retrieval methods, including sparse (Robertson et al., 2009), dense (Karpukhin et al., 2020), late interaction Khattab & Zaharia (2020), and generative retrieval Tay et al. (2022) to name a few. Compared with such retrieval paradigms, $k$-NN retrieval has recently gained increased popularity (see Zhao et al. (2024) for a survey of recent studies) due to its simplicity and efficiency. $k$-NN retrieval uses embeddings from pre-trained models, thus avoiding the need for extensive training on the dataset in hand, and performs retrieval through a simple vector index look-up avoiding expensive model inference at query time.

This paper focuses on improving $k$-NN retrieval accuracy given access to a pre-trained embedding model. Related work can be divided into two categories. The first category, including this paper, aims to improve the embeddings through fine-tuning for the specific dataset and query workload. Fine-tuning the pre-trained model itself, through a similar training strategy used during pre-training is possible, Zhao et al. (2024); Tang et al. (2022); Tam et al. (2022); Liu et al. (2021a); Pires et al. (2019); Yu et al. (2022); Ma et al. (2024), but requires access to the model parameters, is computationally expensive and incurs extra hosting and maintenance costs the fine-tuned models at deployment time. A common alternative in practice is to train Adaptors (Suvansh Sanjeev, 2024; LlamaIndex, 2024b) to transform the output of the models. As our experiments show, adaptors provide limited accuracy gains. On the other hand, our approach, NUDGE, is efficient, model agnostic, does not require hosting and maintaining any model at deployment time, and provides a significant accuracy boost. Work in the second category, orthogonal to this paper, modify the retrieval algorithm to improve accuracy, e.g., through query rewriting or reranking. (see Zhao et al. (2024); Gao et al. (2023) for a survey). Notable examples include Gao et al. (2022); Wang et al. (2023), which rewrite or expand queries with hypothetical documents, Sarthi et al. (2024), which adds summary text chunks to the dataset to improve retrieval, Sachan et al. (2022) which reranks passages based on probability of observing the query, and Lin et al. (2024) which extracts document structure for query answering. Our work could be combined with any of these approaches.

Beyond retrieval, our work is related to recent work on fine-tuning pre-trained models for various purposes (Ouyang et al., 2022; Rafailov et al., 2024; Ziegler et al., 2019; Zhang et al., 2024; Patil et al., 2023), but shows the novel insight that non-parametric fine-tuning of model output, instead of fine-tuning model parameters, can provide significant accuracy improvements. Similar to Ouyang et al. (2022); Rafailov et al. (2024); Ziegler et al. (2019), our optimization constrains how much model output can change during fine-tuning. In NUDGE, this constraint represents the intuition that among all fine-tuned embeddings that achieve a specific training loss, the one with the least change from non-fine-tuned embedding should be preferred. Although this intuition can be broadly applicable to fine-tuning, the KL-divergence constraint in Ouyang et al. (2022); Ziegler et al. (2019) has additional significance due to the use of a reward model.

## 6 CONCLUSION

We studied the problem of fine-tuning embeddings to improve $k$-NN retrieval accuracy. We presented NUDGE, a novel non-parametric approach to embedding fine-tuning that is efficient, provides significant accuracy boosts, and does not require any model hosting or maintenance at deployment time. Our experimental results show that NUDGE improves accuracy by up to 16.0% compared with existing methods and up to 24.4% compared with no fine-tuning. Future work includes incorporating NUDGE inside vector databases and generating and maintaining query sets for fine-tuning.

## REPRODUCIBILITY STATEMENT

End-to-end source code is available at `https://github.com/szeighami/nudge` that downloads the datasets from publicly available sources, runs our methods and the baselines, and reproduces our results. The codebase contains a docker file and instructions to run the code in a docker container to ensure the code is easily executable and runs with the same configuration to reproduce our results exactly. The code contains any and all data processing steps (as already described in Sec. 4), hyperparameter and other experimental settings (as described in Sec. 4 and detailed in Appx. E.1), and implementation of all the methods and baselines.

### ACKNOWLEDGMENTS

We acknowledge support from grants DGE-2243822, IIS-2129008, IIS-1940759, and IIS-1940757 awarded by the National Science Foundation, funds from the State of California, funds from the Alfred P. Sloan Foundation, as well as EPIC lab sponsors: G-Research, Adobe, Microsoft, Google, and Sigma Computing.

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

## A    APPENDIX OVERVIEW

This appendix is organized as follows:

- Appx. B contains the proofs of the theoretical results in the paper.

- Appx. C discusses other NUDGE variants.

- Appx. D discusses the extension of NUDGE to multi-label settings.

- Appx. E provides details about the experimental setting and additional experiments.

## B    PROOFS

Appx. B.1-B.3, respectively, provide the proofs of Theorems. 1-3. The proofs of technical lemmas are presented in Appx. B.4

### B.1    PROOF OF THEOREM 1

We show a reduction from *homogeneous maximum feasible linear subsystem, Max-FLS,* shown to be NP-Hard in Amaldi & Kann (1995). We first formally state the decision version of the problem.

**Problem 2** (Homogeneous Max-FLS). *Given a linear system $\boldsymbol{A}\boldsymbol{x} < \boldsymbol{0}$, where $\boldsymbol{A}$ is of size $s \times t$, and an integer $K$ with $1 < K < s$, does there exist a solution $\boldsymbol{x} \in \mathcal{R}^t$ satisfying at least $K$ inequalities of the system?*

We show a reduction from Homogeneous Max-FLS to the decision version of MaxA-EFT. The decision version of MaxA-EFT asks whether there exists $\boldsymbol{\Delta}$ such that the training accuracy is at least $K$, i.e., whether at least $K$ different $\mathcal{I}_i(\boldsymbol{\Delta})$, $i \in [n_T]$ can be satisfied.

Given $\boldsymbol{A}$, the reduction defines an instance of MaxA-EFT by specifying $\boldsymbol{D}$, $\boldsymbol{Q}$, and $Y$. Specifically, we let $\boldsymbol{Q} = \boldsymbol{A}$, $\boldsymbol{D} = \boldsymbol{0} \in \mathcal{R}^{2 \times t}$ (i.e., a $2 \times t$ matrix only consisting of zeros) and $Y = \{Y_1, ..., Y_s\}$, $Y_i = 0 \; \forall i \in [s]$ (i.e., this instance only has 2 data records, embedding dimension is $t$ and there are $s$ training queries).

We next show that there exists $\boldsymbol{\Delta}$ so that the training accuracy of MaxA-EFT is at least $K$ if and only if there exists a solution $\boldsymbol{x}$ to Homogeneous Max-FLS satisfying at least $K$ inequalities of the system.

Observe that, by the above construction, $\mathcal{I}_i(\boldsymbol{\Delta})$ is satisfied if and only if

$$\boldsymbol{Q}_i \cdot (\boldsymbol{\Delta}_1 - \boldsymbol{\Delta}_0) < 0.$$

Suppose the training accuracy is at least $K$ for some $\boldsymbol{\Delta}$. Let $\boldsymbol{x} = \boldsymbol{\Delta}_1 - \boldsymbol{\Delta}_0$, and let $\{i_1, ..., i_K\}$ be the index of $K$ training samples answered correctly. Thus, we have $\boldsymbol{A}_i \cdot \boldsymbol{x} < 0$ for all $i \in \{i_1, ..., i_K\}$, showing the existing $K$ inequalities in the subsystem that are satisfied.

Conversely, assume some $\boldsymbol{x}$ satisfies at least $K$ inequalities in the subsystem. Let $\boldsymbol{\Delta}_0 = \boldsymbol{0}$ and $\boldsymbol{\Delta}_1 = \boldsymbol{x}$. We have that at least $K$ inequalities in

$$\boldsymbol{Q}\boldsymbol{\Delta}_1 < \boldsymbol{0}$$

are satisfied, so that the training accuracy is at least $K$. The reduction is polynomial time, thus showing MaxA-EFT is NP-hard. $\qquad\square$

### B.2    PROOF OF THEOREM 2

**Setup**. Recall that for any $i \in [n]$,

$$\boldsymbol{G}_i = \sum_{j \in [n_T]} \mathbb{I}[i = Y_j^T] \boldsymbol{Q}_j^T,$$

so that

$$\underset{\boldsymbol{\Delta} \in \mathcal{R}^{n \times d}}{\arg\max} \sum_{i \in [n_T]} \boldsymbol{Q}_i^T \cdot (\boldsymbol{D}_{Y_i^T} + \boldsymbol{\Delta}_{Y_i^T}) = \underset{\boldsymbol{\Delta} \in \mathcal{R}^{n \times d}}{\arg\max} \sum_{i \in [n]} \boldsymbol{G}_i \cdot \boldsymbol{\Delta}_i = \underset{\boldsymbol{\Delta} \in \mathcal{R}^{n \times d}}{\arg\max} \sum_{i \in [n], \|\boldsymbol{G}_i\| \neq 0} \boldsymbol{G}_i \cdot \boldsymbol{\Delta}_i.$$

The proofs here use the above formulation for MaxS-M. Moreover, we set $\boldsymbol{\Delta}_i^\gamma = \boldsymbol{0}$ for any $i$ where $\|\boldsymbol{G}_i\| = 0$, given that they do not appear in the above objective. Note that even when $\boldsymbol{\Delta}_i^\gamma = \boldsymbol{0}$ for any $i \in [n]$, the $i$-th record still influences the BiMax-M objective and needs to be taken into account when solving the outer optimization in BiMax-M.

---

**Algorithm 1** NUDGE-M algorithm

---

**Require:** A training set $\boldsymbol{Q}^T, Y^T$, validation set $\boldsymbol{Q}^V, Y^V$ and data embeddings $\boldsymbol{D}$
**Ensure:** Fine-tuned data embeddings
 1: $\boldsymbol{G}_i \leftarrow \sum_{j \in [n_T]} \mathbb{I}[i = Y_j^T]\boldsymbol{Q}_j^T$ for all $i \in [n]$
 2: Calculate $\mathcal{G}_{i,j}, \mathcal{S}_{i,j}$ for all $i \in [n_V], j \in [n]$
 3: $I \leftarrow \emptyset$
 4: **for** $i$ **in** $[n_V]$ **do**
 5:     **if** $\mathcal{S}_{i,j} > \mathcal{S}_{i,Y_i^V}$ for any $j \in \{j \in [n] \setminus Y_i^V | \mathcal{G}_{i,Y_i^V} - \mathcal{G}_{i,j} = 0\}$ **then**
 6:         **Continue**                        $\triangleright \ I_i = \emptyset$ so skip the query
 7:     $P \leftarrow \{j \in [n] \setminus Y_i^V | \mathcal{G}_{i,Y_i^V} - \mathcal{G}_{i,j} > 0\}$
 8:     $N \leftarrow \{j \in [n] \setminus Y_i^V | \mathcal{G}_{i,Y_i^V} - \mathcal{G}_{i,j} < 0\}$
 9:     $l \leftarrow \max_{j \in P} \frac{\mathcal{S}_{i,j} - \mathcal{S}_{i,Y_i^V}}{\mathcal{G}_{i,Y_i^V} - \mathcal{G}_{i,j}}, u \leftarrow \min_{j \in N} \frac{\mathcal{S}_{i,j} - \mathcal{S}_{i,Y_i^V}}{\mathcal{G}_{i,Y_i^V} - \mathcal{G}_{i,j}}$
10:     **if** $u < 0$ **or** $u < l$ **then**
11:         **Continue**                       $\triangleright \ I_i = \emptyset$ so skip the query
12:     $I_i \leftarrow (\max\{l, 0\}, u)$
13:     $I.\text{append}(I_i)$
14: $\gamma^* \leftarrow \arg\max_\gamma \sum_{I_i \in I} \mathbb{I}[\gamma \in I_i]$
15: $\boldsymbol{D}_i^* \leftarrow \boldsymbol{D}_i + \gamma^* \mathbb{I}[\|\boldsymbol{G}_i\| \neq 0]\frac{\boldsymbol{G}_i}{\|\boldsymbol{G}_i\|}$
16: **return** $\boldsymbol{D}^*$

---

**Finding MaxS-M.** We first find $\texttt{MaxS-M}(\gamma)$ by solving the following optimization problem:

$$\underset{\boldsymbol{\Delta} \in \mathcal{R}^{n \times d}}{\arg\max} \quad \sum_{i \in [n]} \boldsymbol{G}_i \cdot \boldsymbol{\Delta}_i$$
$$\text{s. t.} \quad \|\boldsymbol{\Delta}_i\| \leq \gamma, \quad \forall i \in [n].$$

In this problem, the constraints are independent for each $i \in [n]$, $\|\boldsymbol{G}_i\| \neq 0$, and the objective is simply a summation across $\boldsymbol{G}_i \cdot (\boldsymbol{D}_i + \boldsymbol{\Delta}_i)$ values, so that a solution $\boldsymbol{\Delta}^*$ is optimal for this problem if and only if for each $i \in [n]$, $\boldsymbol{\Delta}_i^*$ is an optimal solution to

$$\underset{\boldsymbol{\Delta} \in \mathcal{R}^d}{\arg\max} \quad \boldsymbol{G}_i \cdot \boldsymbol{\Delta}_i$$
$$\text{s. t.} \quad \|\boldsymbol{\Delta}_i\| \leq \gamma.$$

Solving this problem, we have:

**Lemma 1.** *For any $i \in [n]$, whenever $\|\boldsymbol{G}_i\| \neq 0$ and $\gamma \geq 0$, the optimal solution to*

$$\underset{\boldsymbol{\Delta} \in \mathcal{R}^d}{\arg\min} \quad -\boldsymbol{G}_i \cdot \boldsymbol{\Delta}_i$$
$$\text{s. t.} \quad \|\boldsymbol{\Delta}_i\| \leq \gamma,$$

*is*

$$\boldsymbol{\Delta}_i^\gamma = \gamma \frac{\boldsymbol{G}_i}{\|\boldsymbol{G}_i\|}.$$

**Solving BiMax-M.** The goal is now to find a parameter $\gamma \geq 0$ such that the maximum number of validation queries are answered correctly. Observe that, by definition, the $i$-th validation query, $i \in [n_V]$ is correctly answered when $\forall j \in [n_V] \setminus Y_i^V$,

$$\boldsymbol{Q}_i^V \cdot \boldsymbol{D}_{Y_i} + \gamma \mathbb{I}[\|\boldsymbol{G}_{Y_i^V}\| \neq 0]\boldsymbol{Q}_i^V \cdot \frac{\boldsymbol{G}_{Y_i^V}}{\|\boldsymbol{G}_{Y_i^V}\|} > \boldsymbol{Q}_i^V \cdot \boldsymbol{D}_j + \gamma \mathbb{I}[\|\boldsymbol{G}_j\| \neq 0]\boldsymbol{Q}_i^V \cdot \frac{\boldsymbol{G}_j}{\|\boldsymbol{G}_j\|}.$$

where, we abuse notation and assume $\frac{\mathbb{I}[\|\boldsymbol{G}_j\| \neq 0]}{\|\boldsymbol{G}_j\|} = 0$ if $\|\boldsymbol{G}_j\| = 0$ for any $j$. Define

$$\mathcal{G}_{i,j} = \mathbb{I}[\|\boldsymbol{G}_j\| \neq 0]\boldsymbol{Q}_i^V \cdot \frac{\boldsymbol{G}_j}{\|\boldsymbol{G}_j\|}, \quad \text{and} \quad \mathcal{S}_{i,j} = \boldsymbol{Q}_i^V \cdot \boldsymbol{D}_j$$

for any $j \in [n]$. Thus, we have that the $i$-th validation query is answered correctly when $\forall j \in [n_V] \setminus Y_i^V$

$$\gamma \in I_{i,j} = \begin{cases} \left(-\infty, \frac{\mathcal{S}_{i,j} - \mathcal{S}_{i,Y_i^V}}{\mathcal{G}_{i,Y_i^V} - \mathcal{G}_{i,j}}\right), & \text{if } \mathcal{G}_{i,Y_i^V} - \mathcal{G}_{i,j} < 0 \\ \left(\frac{\mathcal{S}_{i,j} - \mathcal{S}_{i,Y_i^V}}{\mathcal{G}_{i,Y_i^V} - \mathcal{G}_{i,j}}, \infty\right), & \text{if } \mathcal{G}_{i,Y_i^V} - \mathcal{G}_{i,j} > 0 \\ \mathcal{R}, & \text{if } \mathcal{G}_{i,Y_i^V} - \mathcal{G}_{i,j} = 0 \text{ and } \mathcal{S}_{i,j} - \mathcal{S}_{i,Y_i^V} < 0 \\ \emptyset, & \text{otherwise.} \end{cases}$$

$I_{i,j}$ defines an interval in $\mathcal{R}$ so that $\gamma \in I_{i,j}$ means the correct answer to the $i$-th validation query has higher similarity to the query compared with the $j$-th data record, for $j \neq Y_i^V$. Thus, to answer the $i$-th query correctly, we must have $\gamma \in I_i = \cap_{j \neq Y_i^V} I_{i,j}$. To maximize the number of queries answered correctly, after finding $I_i$ for all $i$, we simply need to find a $\gamma$ value that intersects the most number of intervals among $I_1, ..., I_{n_V}$, i.e., $\arg\max_\gamma \sum_{I_i \in I} \mathbb{I}[\gamma \in I_i]$, where $I = \{I_1, ..., I_{n_V}\}$. Finding a point where maximum intervals overlap is a basic algorithmic problem and can be done by a single iteration through the intervals after sorting their start and end points.

**NUDGE-M**. Alg. 1 presents NUDGE-M, which formalizes the above procedure, and also incorporates the constraint $\gamma \geq 0$. Lines 4-13 find the intervals $I_i$, $\forall i \in [n_V]$ and add it to a list $I$ (intersection of half intervals can be found by calculating the maximum of lower bounds and minimum of upper bounds of the intervals, done in lines 7-9). The algorithm returns new data embeddings by simply performing a single addition.

**Time complexity**. Alg. 1 can be implemented so that lines 9-5 with a single pass over the dataset. Therefore, finding $I_i$ for all queries (lines 4-13) take $O(n_V \times n \times d)$. Calculating $G$ takes time $O(n_T \times d)$. Finding $\gamma^*$ in line 14 is the basic problem of finding the maximum number of overlapping ranges and can be done in $O(n_V \log(n_V))$, by first sorting the ranges in $I$ based on their lower bound and iteratively traversing the sorted list and keeping track of the number of overlapping ranges. Thus, Alg. 1 can be implemented in $O(n_V(n \times d + \log(n_V) + n_T \times d))$.

### B.3 PROOF OF THEOREM 3

**Setup**. In the discussion below, for simplicity, we assume for any $i \in [n]$ $\boldsymbol{G}_i \cdot \boldsymbol{D}_i \geq 0$. Although the discussion below can be extended to consider $\boldsymbol{G}_i \cdot \boldsymbol{D}_i < 0$, the results will be more tedious (see Lemma 2). Moreover, in practice we expect $\boldsymbol{G}_i \cdot \boldsymbol{D}_i \geq 0$ to hold, as otherwise queries and their correct answers will have negative similarity suggesting either a poor training dataset or pre-trained embedding model. We simply set $\boldsymbol{G}_i = \boldsymbol{0}$ whenever $\boldsymbol{G}_i \cdot \boldsymbol{D}_i < 0$. Moreover, similar to Appx. B.2, we rewrite the MaxS-N objective as

$$\arg\max_{\boldsymbol{\Delta} \in \mathcal{R}^{n \times d}} \sum_{i \in [n_T]} \boldsymbol{Q}_i^T \cdot (\boldsymbol{D}_{Y_i^T} + \boldsymbol{\Delta}_{Y_i^T}) = \arg\max_{\boldsymbol{\Delta} \in \mathcal{R}^{n \times d}} \sum_{i \in [n]} \boldsymbol{G}_i \cdot \boldsymbol{\Delta}_i = \arg\max_{\boldsymbol{\Delta} \in \mathcal{R}^{n \times d}} \sum_{i \in [n], \|\boldsymbol{G}_i\| \neq 0} \boldsymbol{G}_i \cdot \boldsymbol{\Delta}_i,$$

and set $\boldsymbol{\Delta}_i^\gamma = \boldsymbol{0}$ for any $i$ where $\|\boldsymbol{G}_i\| = 0$. Furthermore, we assume the embeddings $\boldsymbol{D}_i$ are normalized, (i.e., $\|\boldsymbol{D}_i\| = 1, \forall i \in [n]$), or we otherwise normalize them.

**Finding MaxS-N**. We first find MaxS-N$(\gamma)$ by solving the following optimization problem:

$$\begin{aligned} \arg\max_{\boldsymbol{\Delta} \in \mathcal{R}^{n \times d}} \quad & \sum_{i \in [n]} \boldsymbol{G}_i \cdot \boldsymbol{D}_i \\ \text{s. t.} \quad & \|\boldsymbol{\Delta}_i\|^2 \leq \gamma, \quad \forall i \in [n] \\ & \|\boldsymbol{\Delta}_i + \boldsymbol{D}_i\| = 1, \forall i \in [n]. \end{aligned}$$

In this problem, the constraints are independent for each $i \in [n]$, and the objective is simply a summation across $\boldsymbol{G}_i \cdot (\boldsymbol{D}_i + \boldsymbol{\Delta}_i)$ values, so that a solution $\boldsymbol{\Delta}^*$ is optimal for this problem if and only if for each $i \in [n]$, $\boldsymbol{\Delta}_i^*$ is an optimal solution to

$$\begin{aligned} \arg\max_{\boldsymbol{\Delta} \in \mathcal{R}^d} \quad & \boldsymbol{G}_i \cdot \boldsymbol{D}_i \\ \text{s. t.} \quad & \|\boldsymbol{\Delta}_i\|^2 \leq \gamma, \\ & \|\boldsymbol{\Delta}_i + \boldsymbol{D}_i\| = 1. \end{aligned}$$

Solving this problem, we have:

**Lemma 2.** *For any $i \in [n]$, let $\theta_i$ be the angle between $\boldsymbol{G}_i$ and $\boldsymbol{D}_i$. Whenever $\theta_i \in [0, \frac{\pi}{2}]$, $\gamma \geq 0$, $\|\boldsymbol{D}_i\| = 1$ and $\|\boldsymbol{G}_i\| \neq 0$ the optimal solution to the optimization problem*

$$\underset{\boldsymbol{\Delta} \in \mathcal{R}^d}{\arg\min} \quad -\boldsymbol{G}_i \cdot \boldsymbol{\Delta}_i$$
$$\text{s. t.} \quad \|\boldsymbol{\Delta}_i\|^2 \leq \gamma,$$
$$\|\boldsymbol{\Delta}_i + \boldsymbol{D}_i\| = 1,$$

*is*

$$\boldsymbol{\Delta}_i^\gamma = \begin{cases} \frac{\boldsymbol{G}_i}{\|\boldsymbol{G}_i\|} - \boldsymbol{D}_i, & \text{if } \cos\theta_i \geq 1 - \frac{\gamma}{2} \\ \frac{\sqrt{\gamma(4-\gamma)}(\boldsymbol{G}_i - (\boldsymbol{D}_i \cdot \boldsymbol{G}_i)\boldsymbol{D}_i)}{2\|\boldsymbol{G}_i - (\boldsymbol{D}_i \cdot \boldsymbol{G}_i)\boldsymbol{D}_i\|} - \frac{\gamma}{2}\boldsymbol{D}_i, & \text{otherwise.} \end{cases}$$

For simplicity of notation, we denote by $\boldsymbol{Z}_i \in \mathcal{R}^d$ the vector

$$\boldsymbol{Z}_i = \frac{\boldsymbol{G}_i - (\boldsymbol{D}_i \cdot \boldsymbol{G}_i)\boldsymbol{D}_i}{\|\boldsymbol{G}_i - (\boldsymbol{D}_i \cdot \boldsymbol{G}_i)\boldsymbol{D}_i\|}.$$

We also denote by $\mathcal{Z}_{i,j} = \boldsymbol{Q}_i^V \cdot \boldsymbol{Z}_j$, by $\mathcal{D}_{i,j} = \boldsymbol{Q}_i^V \cdot \boldsymbol{D}_j$ and by $\mathcal{G}_{i,j} = \mathbb{I}[\|\boldsymbol{G}_j\| \neq 0]\boldsymbol{Q}_i^V \cdot \frac{\boldsymbol{G}_j}{\|\boldsymbol{G}_j\|}$ for all $i \in [n_V], j \in [n]$.

**Solving BiMax-N**. Next, we consider solving BiMax-N, $\forall j \in [n]$. Substituting $\Delta^\gamma$ into the definition of $\mathcal{I}_i^V(\boldsymbol{\Delta}^\gamma)$ for each $i \in [n_V]$, $\mathcal{I}_i^V(\boldsymbol{\Delta}^\gamma) = 1$ if and only if for all $j \in [n] \setminus Y_i^V$

$$\boldsymbol{Q}_i^V \cdot (\boldsymbol{D}_{Y_i^V} + \boldsymbol{\Delta}_{Y_i^V}^\gamma) > \boldsymbol{Q}_i^V \cdot (\boldsymbol{D}_j + \boldsymbol{\Delta}_j^\gamma). \tag{7}$$

First, consider the case simpler setting when $\|\boldsymbol{G}_j\| \neq 0$ for all $j \in [n]$. Let $\bar{\theta}_{i,j} = \max\{\theta_i, \theta_j\}$ and $\underline{\theta}_{i,j} = \min\{\theta_i, \theta_j\}$. Substituting the values from Lemma 2, we have Eq. 7 is equivalent to

$$\begin{cases} (1-\frac{\gamma}{2})\mathcal{D}_{i,Y_i^V} + \frac{\sqrt{\gamma(4-\gamma)}}{2}\mathcal{Z}_{i,Y_i^V} > (1-\frac{\gamma}{2})\mathcal{D}_{i,j} + \frac{\sqrt{\gamma(4-\gamma)}}{2}\mathcal{Z}_{i,j}, & \text{if } \gamma < 2 - 2\cos\underline{\theta}_{Y_i^V,j} \\ (1-\frac{\gamma}{2})\mathcal{D}_{i,Y_i^V} + \frac{\sqrt{\gamma(4-\gamma)}}{2}\mathcal{Z}_{i,Y_i^V} > \mathcal{G}_{i,j}, & \begin{aligned}&\text{if } \gamma \geq 2 - 2\cos\underline{\theta}_{Y_i^V,j}, \\ &\gamma < 2-2\cos\bar{\theta}_{Y_i^V,j}, \theta_{Y_i^V} < \theta_j\end{aligned} \\ \mathcal{G}_{i,Y_i^V} > (1-\frac{\gamma}{2})\mathcal{D}_{i,j} + \frac{\sqrt{\gamma(4-\gamma)}}{2}\mathcal{Z}_{i,j}, & \begin{aligned}&\text{if } \gamma \geq 2 - 2\cos\underline{\theta}_{Y_i^V,j}, \\ &\gamma < 2-2\cos\bar{\theta}_{Y_i^V,j}, \theta_{Y_i^V} > \theta_j\end{aligned} \\ \mathcal{G}_{i,Y_i^V} > \mathcal{G}_{i,j}, & \text{otherwise.} \end{cases} \tag{8}$$

Observe that finding the values of $\gamma$ for which the inequalities in Eq. 8 hold requires solving inequalities of the form $a\sqrt{\gamma(4-\gamma)} + b\gamma + c > 0$, for some $a, b, c \in \mathcal{R}$.

**Lemma 3.** *Let $f(\gamma) = a\sqrt{\gamma(4-\gamma)} + b\gamma + c$ for $a, b, c \in \mathcal{R}$. We say there are $k$ roots, $\gamma_0, ..., \gamma_{k-1}$, if there exists $k$ distinct $\gamma_i \in (0, 4)$ for which $f(\gamma_i) = 0$. We must have $k \in \{0, 1, 2\}$, if two roots exist, we have*

$$\gamma_0 = \frac{2a^2 - bc - a\sqrt{\delta}}{a^2 + b^2}, \ \gamma_1 = \frac{2a^2 - bc + a\sqrt{\delta}}{a^2 + b^2}, \ \text{for } \delta = 4a^2 - 4bc - a^2c^2,$$

*and if only one root exists*

$$\gamma_0 \in \{\frac{2a^2 - bc - a\sqrt{\delta}}{a^2 + b^2}, \frac{2a^2 - bc + a\sqrt{\delta}}{a^2 + b^2}\}.$$

*Moreover, for any $\gamma \in (0, 4)$, $f(\gamma) > 0$ holds if and only if*

$$\gamma \in \begin{cases} (0, 4) & \text{if zero roots and } (c > 0 \text{ or } (c = 0, b > 0) \text{ or } (c = b = 0, a > 0)) \\ \emptyset & \text{if zero roots and } (c < 0 \text{ or } (c = 0, b < 0) \text{ or } (c = b = 0, a \leq 0)) \\ (\gamma_0, \gamma_1) & \text{if two roots and } c < 0 \\ (0, \gamma_0) \cup (\gamma_1, 4) & \text{if two roots and } c > 0 \\ (0, \gamma_0) & \text{if one root and } (c > 0 \text{ or } c = 0, a > 0) \\ (\gamma_0, 4) & \text{if one root and } (c < 0 \text{ or } c = 0, a < 0). \end{cases} \tag{9}$$

Thus, for the $i$-th query and the $j$-th data record, $i \in [n_V]$, $j \in [n] \setminus Y_i^V$, we can use Eq. 9 together with Eq. 8 to obtain a set $I_{i,j} \subseteq (0, 4)$ such that Eq. 7 holds if and only if $\gamma \in I_{i,j}$. Moreover, $I_{i,j}$ will consist of at most 3 intervals in $\subseteq (0, 4)$. Let $I_i = \cap_{j \in [n] \neq Y_i^V} I_{i,j}$, and note that the $i$-th query will be answered accurately if and only if $\gamma \in I_i$. Finally, $\gamma^* = \arg\max_\gamma \sum_{i \in [n_V]} \mathbb{I}[\gamma \in I_i]$ is an optimal solution to the MAxSS problem.

Now whenever $\|\boldsymbol{G}_j\| = 0$ for any $j \in [n]$, Eq. 7 changes since we need to also consider $\boldsymbol{\Delta}_{Y_i^V}^\gamma = \boldsymbol{0}$, $\boldsymbol{\Delta}_j^\gamma = \boldsymbol{0}$ or both. Each case leads to a similar set of inequalities to Eq. 8, which are similarly solved using Lemm 3. Moreover, we calculate validation accuracy separately for when $\gamma = 0$.

**NUDGE-N**. NUDGE-N follows the above procedure. It first calculates $\boldsymbol{G}$ and $\boldsymbol{Z}$, so that for the $i$-th query, and $j$-th data points, $j \neq Y_i^V$ applying Lemma 3 to Eq. 8 gives $I_{i,j}$. Then, it computes $I_i$ by a single pass over $I_{i,j}$ values, and finally finds a $\gamma$ value that intersects most $I_i$'s by sorting their beginning and end and iteration through the list.

**Time Complexity**. Calculating $\boldsymbol{G}$ takes time $O(dn_T)$, and $\boldsymbol{Z}$ takes $O(dn)$. Then, for the $i$-th query, it finds $I_i$ which takes $O(nd)$, since each $I_{i,j}$ only contains a constant number of intervals and can be computed in $O(d)$. Finally, finding $\gamma^*$ that intersects the most intervals, can be done in $O(n_V \log n_V)$. Thus, in total, NUDGE-N takes total of $O(n_V(nd + \log n_V) + n_T d)$.

### B.4 Proof of Technical Lemmas

#### B.4.1 Proof of Lemma 1

If $\gamma = 0$ the only feasible, and therefore optimal, solution is $\boldsymbol{\Delta} = \boldsymbol{0}$. Next, consider $\gamma > 0$. Observe that the optimization problem is convex, so we find $\boldsymbol{\Delta}_i$ values that satisfy the KKT conditions which, since the Slater's condition holds whenever $\gamma > 0$, provide necessary and sufficient conditions for optimality.

We have that the Lagrangian is

$$L(\boldsymbol{\Delta}_i, \mu, \lambda) = -\boldsymbol{G}_i \cdot \boldsymbol{D}_i + \frac{1}{2}\mu(\|\boldsymbol{\Delta}_i\|^2 - \gamma^2),$$

so that the KKT conditions are

$$-\boldsymbol{G}_i + \mu\boldsymbol{\Delta}_i = \boldsymbol{0}, \tag{10}$$

$$\mu(\|\boldsymbol{\Delta}_i\|^2 - \gamma^2) = 0, \tag{11}$$

$$\mu \geq 0. \tag{12}$$

If $\mu = 0$, we must have $\|\boldsymbol{G}_i\| = 0$, which is a contradiction. Thus, $\mu > 0$, and

$$\|\boldsymbol{\Delta}_i\|^2 - \gamma^2 = 0. \tag{13}$$

From Eq. 10,

$$\boldsymbol{\Delta}_i = \frac{\boldsymbol{G}_i}{\mu} \tag{14}$$

Substituting this into Eq. 13, and since $\mu > 0$, we have

$$\mu = \frac{\|\boldsymbol{G}_i\|}{\gamma},$$

and substituting back into Eq. 14, we have

$$\boldsymbol{\Delta}_i = \gamma\frac{\boldsymbol{G}_i}{\|\boldsymbol{G}_i\|}.$$

#### B.4.2 Proof of Lemma 2

We replace $\|\boldsymbol{\Delta}_i + \boldsymbol{D}_i\| = 1$ with $\|\boldsymbol{\Delta}_i + \boldsymbol{D}_i\| \leq 1$ so that the optimization is over a convex set, and solve the problem with $\|\boldsymbol{\Delta}_i + \boldsymbol{D}_i\| \leq 1$ constraint. As we will see, the only solution to the problem will have $\|\boldsymbol{\Delta}_i + \boldsymbol{D}_i\| = 1$, implying that it is the optimal solution to the original problem as well. Thus, consider the convex problem with $\|\boldsymbol{\Delta}_i + \boldsymbol{D}_i\| \leq 1$ constraint, or equivalantly $\|\boldsymbol{\Delta}_i + \boldsymbol{D}_i\|^2 \leq 1$.

We find $\boldsymbol{\Delta}_i$ values that satisfy the KKT conditions, which, since the Slater's condition holds whenever $\gamma > 0$, provide necessary and sufficient conditions for optimality.

We have that the Lagrangian is

$$L(\mathbf{\Delta}_i, \mu, \lambda) = -\mathbf{G}_i \cdot \mathbf{D}_i + \frac{1}{2}\mu(\|\mathbf{\Delta}_i\|^2 - \gamma) + \frac{1}{2}\lambda(\|\mathbf{\Delta}_i\|^2 + 2\mathbf{\Delta}_i \cdot \mathbf{D}_i),$$

so that the KKT conditions are

$$-\mathbf{G}_i + \mu\mathbf{\Delta}_i + \lambda\mathbf{\Delta}_i + \lambda\mathbf{D}_i = \mathbf{0}, \tag{15}$$

$$\lambda(\|\mathbf{\Delta}_i\|^2 + 2\mathbf{\Delta}_i \cdot \mathbf{D}_i) = 0, \tag{16}$$

$$\mu(\|\mathbf{\Delta}_i\|^2 - \gamma) = 0, \tag{17}$$

$$\|\mathbf{\Delta}_i\|^2 \leq \gamma, \tag{18}$$

$$\|\mathbf{\Delta}_i\|^2 + 2\mathbf{\Delta}_i \cdot \mathbf{D}_i \leq 0, \tag{19}$$

$$\mu, \lambda \geq 0, \tag{20}$$

where Eq. 15 is obtained by setting the gradient of the Lagrangian to zero, and Eq. 19 and 16 are obtained by substituting

$$\|\mathbf{D}_i + \mathbf{\Delta}_i\|^2 = \|\mathbf{D}_i\|^2 + \|\mathbf{\Delta}_i\|^2 + 2\mathbf{\Delta}_i \cdot \mathbf{D}_i = 1 + \|\mathbf{\Delta}_i\|^2 + 2\mathbf{\Delta}_i \cdot \mathbf{D}_i,$$

since $\|\mathbf{\Delta}_i\| = 1$.

To find the points satisfying all Eq. 15-20, we consider 4 setting depending on whether $\lambda = 0$ or $\mu = 0$ or not.

**Case 1,** $\lambda = 0, \mu = 0$. Substituting $\lambda = 0$ and $\mu = 0$ in Eq. 15, we have $\mathbf{G}_i = \mathbf{0}$. Thus, no solution with $\lambda = 0$ and $\mu = 0$ exists since $\mathbf{G}_i \neq \mathbf{0}$.

**Case 2,** $\lambda = 0, \mu > 0$. Having $\lambda = 0$ and $\mu > 0$, we have from Eq. 15

$$\mathbf{\Delta}_i = \frac{1}{\mu}\mathbf{G}_i,$$

substituting which into Eq. 17 with $\mu > 0$ w have

$$\frac{1}{\mu^2}\|\mathbf{G}_i\|^2 = \gamma,$$

$$\mu = \frac{\|\mathbf{G}_i\|}{\sqrt{\gamma}},$$

and therefore $\mathbf{\Delta}_i = \sqrt{\gamma}\frac{\mathbf{G}_i}{\|\mathbf{G}_i\|}$ (KKT conditions become similar to the ones in Theorem 2). Because $\mu > 0$, we must have $\|\mathbf{\Delta}_i\|^2 = \gamma$ due to Eq. 17, so that from Eq. 19 we have

$$\gamma + 2\sqrt{\gamma}\frac{\mathbf{G}_i}{\|\mathbf{G}_i\|} \cdot \mathbf{D}_i \leq 0,$$

$$\gamma + 2\sqrt{\gamma}\cos\theta_i \leq 0,$$

where $\theta_i$ is the angle between $\mathbf{G}_i$ and $\mathbf{D}_i$. Thus, we must have

$$\cos\theta_i \leq -\frac{\sqrt{\gamma}}{2}.$$

Since $\cos\theta_i \geq 0$ and $\gamma > 0$ by assumption, there are no solutions with $\lambda = 0, \mu > 0$.

**Case 3,** $\lambda > 0, \mu = 0$. Substituting $\mu = 0$ in Eq. 15 we have

$$\mathbf{\Delta}_i = \frac{\mathbf{G}_i - \lambda\mathbf{D}_i}{\lambda}.$$

Substituting this in Eq. 16, we have

$$\|\frac{\mathbf{G}_i - \lambda\mathbf{D}_i}{\lambda}\|^2 + 2\frac{\mathbf{G}_i - \lambda\mathbf{D}_i}{\lambda} \cdot \mathbf{D}_i = 0,$$

$$\|\mathbf{G}_i\|^2 + \lambda^2 - 2\lambda\mathbf{G}_i \cdot \mathbf{D}_i + 2\lambda(\mathbf{G}_i \cdot \mathbf{D}_i - \lambda) = 0,$$

implying, since $\lambda > 0$,

$$\lambda = \|\mathbf{G}_i\|,$$

and therefore

$$\mathbf{\Delta}_i = \frac{\mathbf{G}_i}{\|\mathbf{G}_i\|} - \mathbf{D}_i.$$

Note that to satisfy Eq. 18, we must have

$$(\frac{\boldsymbol{G}_i}{\|\boldsymbol{G}_i\|} - \boldsymbol{D}_i) \cdot (\frac{\boldsymbol{G}_i}{\|\boldsymbol{G}_i\|} - \boldsymbol{D}_i) \leq \gamma,$$

simplifying which we have

$$\cos\theta_i \geq 1 - \frac{\gamma}{2}.$$

To summarize, we showed that setting

$$\boldsymbol{\Delta}_i = \frac{\boldsymbol{G}_i}{\|\boldsymbol{G}_i\|} - \boldsymbol{D}_i, \ \mu = 0, \ \text{and} \ \lambda = \|\boldsymbol{G}_i\|$$

satisfy all KKT conditions whenever $\cos\theta_i \geq 1 - \frac{\gamma}{2}$.

**Case 4,** $\lambda > 0, \mu > 0$. From Eq. 15, we have

$$\boldsymbol{\Delta}_i = \frac{\boldsymbol{G}_i - \lambda\boldsymbol{D}_i}{\lambda + \mu} \tag{21}$$

and since $\|\boldsymbol{\Delta}_i\| - \gamma = 0$, Eq. 16 simplifies to $\gamma + 2\boldsymbol{\Delta}_i \cdot \boldsymbol{D}_i = 0$, so that

$$2\frac{\boldsymbol{G}_i \cdot \boldsymbol{D}_i - \lambda\boldsymbol{D}_i \cdot \boldsymbol{D}_i}{\lambda + \mu} = -\gamma.$$

Therefore,

$$\mu = 2\frac{\lambda - \boldsymbol{G}_i \cdot \boldsymbol{D}_i}{\gamma} - \lambda. \tag{22}$$

Substituting this in Eq.'21, we have

$$\boldsymbol{\Delta}_i = \frac{\gamma}{2}\frac{\boldsymbol{G}_i - \lambda\boldsymbol{D}_i}{\lambda - \boldsymbol{G}_i \cdot \boldsymbol{D}_i}. \tag{23}$$

Now, substituting Eq. 23 in Eq. 17, we get

$$\frac{\gamma^2}{4(\lambda - \boldsymbol{G}_i \cdot \boldsymbol{D}_i)^2}(\boldsymbol{G}_i - \lambda\boldsymbol{D}_i) \cdot (\boldsymbol{G}_i - \lambda\boldsymbol{D}_i) = \gamma$$

So

$$\frac{\gamma}{4}(\|\boldsymbol{G}_i\|^2 - \lambda^2 - 2\lambda\boldsymbol{G}_i \cdot \boldsymbol{D}_i) = \lambda^2 + (\boldsymbol{G}_i \cdot \boldsymbol{D}_i)^2 - 2\lambda\boldsymbol{G}_i \cdot \boldsymbol{D}_i.$$

Rearranging, we obtain a quadratic equation in $\lambda$,

$$\lambda^2 - 2\boldsymbol{D}_i \cdot \boldsymbol{G}_i\lambda + \frac{\|\boldsymbol{G}_i\|^2(\gamma - 4\cos^2\theta_i)}{\gamma - 4} = 0,$$

where we have used the fact that $(\boldsymbol{D}_i \cdot \boldsymbol{G}_i)^2 = \|\boldsymbol{G}_i\|^2 \cos^2\theta_i$. Solving the equation and simplifying we have

$$\lambda = \boldsymbol{D}_i \cdot \boldsymbol{G}_i \pm \|\boldsymbol{G}_i\|\sin\theta_i\sqrt{\frac{\gamma}{4 - \gamma}}. \tag{24}$$

Note that we must also have $\mu > 0$, so substituting Eq. 24 into Eq. 22 and rearranging we must have

$$\pm\|\boldsymbol{G}_i\|\sin\theta_i\sqrt{\frac{\gamma}{4 - \gamma}}(\frac{2}{\gamma} - 1) - \boldsymbol{D}_i \cdot \boldsymbol{G}_i > 0,$$

$$\pm\sin\theta_i\sqrt{\frac{\gamma}{4 - \gamma}}(\frac{2}{\gamma} - 1) > \cos\theta_i. \tag{25}$$

Observe that, for $\theta_i \in [0, \frac{\pi}{2}]$, both $\sin\theta_i$ and $\cos\theta_i$ are non-negative, so that when $\gamma < 2$ only the positive branch is able to satisfy Eq. 25. In this case, we have

$$\lambda = \boldsymbol{D}_i \cdot \boldsymbol{G}_i + \|\boldsymbol{G}_i\|\sin\theta_i\sqrt{\frac{\gamma}{4 - \gamma}}. \tag{26}$$

Note that

$$\|\boldsymbol{G}_i\|\cos\theta_i + \|\boldsymbol{G}_i\|\sin\theta_i\sqrt{\frac{\gamma}{4 - \gamma}} > 0,$$

for $\theta_i \in [0, \frac{\pi}{2}]$, and thus $\lambda > 0$ is satisfied. Simplifying Eq. 25, observe that for $\theta_i \in [0, \frac{\pi}{2}]$ and $0 < \gamma < 2$,

$$\sin\theta_i\sqrt{\frac{(2-\gamma)^2}{\gamma(4-\gamma)}} > \cos\theta_i \iff \theta_i > \arctan(\sqrt{\frac{\gamma(4-\gamma)}{(2-\gamma)^2}}) \iff \cos\theta_i < \frac{1}{\sqrt{1 + \frac{\gamma(4-\gamma)}{(2-\gamma)^2}}} = 1 - \frac{\gamma}{2}.$$
(27)

To summarize, taking the positive branch in Eq. 24, we showed that

$$\boldsymbol{\Delta}_i = \frac{\sqrt{\gamma(4-\gamma)}(\boldsymbol{G}_i - (\boldsymbol{D}_i \cdot \boldsymbol{G}_i)\boldsymbol{D}_i)}{2\|\boldsymbol{G}_i\|\sin\theta_i} - \frac{\gamma}{2}\boldsymbol{D}_i,$$

$$\lambda = \boldsymbol{D}_i \cdot \boldsymbol{G}_i + \|\boldsymbol{G}_i\|\sin\theta_i\sqrt{\frac{\gamma}{4-\gamma}}, \text{ and}$$

$$\mu = \|\boldsymbol{G}_i\|\sin\theta_i\sqrt{\frac{(2-\gamma)^2}{\gamma(4-\gamma)}} - \boldsymbol{D}_i \cdot \boldsymbol{G}_i$$

satisfy the KKT conditions whenever

$$\cos\theta_i < 1 - \frac{\gamma}{2}.$$

Finally, consider the negative branch in Eq. 24 and $\gamma > 2$. To have $\mu > 0$, following an argument similar to Eq. 27, we must have

$$\cos\theta_i < |1 - \frac{\gamma}{2}|,$$

and similarly for $\lambda > 0$, we must have

$$\|\boldsymbol{G}_i\|\cos\theta_i - \|\boldsymbol{G}_i\|\sin\theta_i\sqrt{\frac{\gamma}{4-\gamma}} > 0 \iff \theta_i < \arctan(\sqrt{\frac{4-\gamma}{\gamma}}) \iff \cos\theta_i > \frac{1}{2}\sqrt{\gamma}.$$

However, $\frac{\sqrt{\gamma}}{2} \geq |1 - \frac{\gamma}{2}|$ for all $\gamma \in [2, 4]$, which implies $\mu > 0$ and $\lambda > 0$ cannot hold at the same time in this case. Finally, observe that $\|\boldsymbol{G}_i - (\boldsymbol{D}_i \cdot \boldsymbol{G}_i)\boldsymbol{D}_i\| = \|\boldsymbol{G}_i\|\sin\theta_i$. □

### B.4.3   PROOF OF LEMMA 3

We first find $\gamma$ values where $f(\gamma) = 0$. Observe that

$$a\sqrt{\gamma(4-\gamma)} + b\gamma + c = 0 \Rightarrow a^2\gamma(4-\gamma) - (b\gamma + c)^2 = 0,$$

so that $a^2\gamma(4-\gamma) - (b\gamma + c)^2 = 0$ is necessary for $f(\gamma) = 0$. Moreover solving the quadratic inequality, $a^2\gamma(4-\gamma) - (b\gamma + c)^2 = 0$ iff $\gamma \in \{\gamma_0, \gamma_1\}$ where

$$\gamma_0 = \frac{2a^2 - bc - a\sqrt{\delta}}{a^2 + b^2}, \quad \gamma_1 = \frac{2a^2 - bc + a\sqrt{\delta}}{a^2 + b^2}, \text{ for } \delta = 4a^2 - 4bc - a^2c^2.$$

Thus, $\gamma \in \{\gamma_0, \gamma_1\}$ is necessary for $f(\gamma) = 0$, which proves the first part of the lemma.

The proof of the rest of the lemma uses the above, the fact that $f$ is continuous, in addition to the following facts:

$$c > 0 \Rightarrow f(0) > 0, \ c < 0 \Rightarrow f(0) < 0 \tag{28}$$
$$c = 0, b > 0 \Rightarrow f(4) > 0, \ c = 0, b < 0 \Rightarrow f(4) < 0 \tag{29}$$
$$f \text{ has at most 1 local extrema, which is a maximum if } a > 0 \text{ and a minimum if } a < 0 \tag{30}$$

If there are zero roots, $f(\gamma) > 0$ either for all of $(0, 4)$ or none of it, which can be determined by checking the function value at 0 or 4 using Eq. 28 and 29 if $c \neq 0$ or $b \neq 0$. If $c = b = 0$, then we can solve $f(\gamma) > 0$ based on whether the function has a minimum or maximum using Eq. 30, or is a constant when $a = 0$.

If there is one root, $\gamma_0$, then $f(\gamma)$ is either positive after $\gamma_0$ or before it. If $c \neq 0$, we can find this by checking $f(0)$ using Eq. 28. If $c = 0$, then $f(\gamma)$ has two roots, at 0 and $\gamma_0$, and $f(\gamma) > 0$ can be determined based on whether the function has a minimum or maximum using Eq. 30. Both $c$ and $a$ cannot be zero, because the function $b\gamma$ cannot be zero both at $\gamma = 0$ and $\gamma = \gamma_0$ for $\gamma_0 \neq 0$.

If there are two roots, then $f(\gamma) > 0$ either between the two roots, or outside the interval between the two roots, which can be checked based on $f(0)$ using Eq 28. $c$ cannot be zero because $f(\gamma)$ cannot be zero at $\gamma = 0$, $\gamma = \gamma_0$ and $\gamma = \gamma_1$ for 3 distinct values fo $0, \gamma_0, \gamma_1$. □

## C    OTHER NUDGE VARIANTS

### C.1    SPARSITY CONSTRAINT

In NUDGE, how embeddings are fine-tuned (i.e., the form of the matrix $\mathbf{\Delta}$) depends on how constraints are defined in the formulation of the constrained non-parametric optimization problem. Our discussion so far considers constraints on the L2 norm of the change in embedding (in `MaxS-M` optimization problem in Sec. 3.2.1) and constraints that force created embeddings to be normalized (in `MaxS-N` optimization problem in Sec. 3.2.2). Here, instead of bounding the L2 norm, we consider bounding the L1 norm of embedding changes. As we see next, bounding the L1 norm creates sparse embedding updates.

Consider the `MaxS-M` optimization problem but with bounds on L1 norm instead of L2 norm. We obtain the following optimization problem, denoted as `MaxS-S`.

$$\texttt{MaxS-S}(\gamma) = \arg\max \sum_{i \in [n_T]} \boldsymbol{Q}_i^T \cdot (\boldsymbol{D}_{Y_i^T} + \boldsymbol{\Delta}_{Y_i^T})$$

$$\text{s.t. } \boldsymbol{\Delta} \in \mathcal{R}^{n \times d}, \|\boldsymbol{\Delta}_i\|_1 \leq \gamma \quad \forall i \in [n],$$

Where $\|.\|_1$ is the L1 norm. A $\boldsymbol{\Delta} \in \texttt{MaxS-S}(\gamma)$ changes data embeddings by at most $\gamma$, now *in terms of the L1 norm*, while maximizing the similarity between training queries and their ground-truth answer. We have that[2]

**Lemma 4.** *The optimal solution to* `MaxS-S`$(\gamma)$, $\boldsymbol{\Delta}^\gamma$, *is*

$$\boldsymbol{\Delta}_{i,j}^\gamma = \begin{cases} 0 & j \in [d] \setminus j^* \\ \gamma \operatorname{sign}(\boldsymbol{G}_{i,j^*}) & j = j^* \text{ otherwise,} \end{cases} \quad \text{where} \quad j^* = \arg\max_{j \in [d]} |\boldsymbol{G}_{i,j}|, \qquad (31)$$

*for all $i \in [n]$ and $j \in [d]$, where $\boldsymbol{G}$ is defined as in Eq. 3 .*

**Proof.**    Observe that for any $i \in [n]$, we have that, $\boldsymbol{\Delta}_i^\gamma$, the optimal embedding change in `MaxS-S`$(\gamma)$ for the $i$ data record, is the optimal solution to

$$\arg\max_{\boldsymbol{\Delta}_i \in \mathcal{R}^d} \quad \boldsymbol{G}_i \cdot \boldsymbol{\Delta}_i$$

$$\text{s.t. } \|\boldsymbol{\Delta}_i\|_1 \leq \gamma.$$

$\boldsymbol{\Delta}_i^\gamma$ defined in Eq. 31 is a feasible solution to above. Moreover, $\boldsymbol{G}_i \cdot \boldsymbol{\Delta}_i^\gamma = \gamma \max_{j \in [d]} |\boldsymbol{G}_{i,j}|$. Now consider any feasible solution $\boldsymbol{\Delta}_i$.

$$\boldsymbol{G}_i \cdot \boldsymbol{\Delta}_i = \sum_{j \in [d]} \boldsymbol{G}_{i,j} \boldsymbol{\Delta}_{i,j} \leq \sum_{j \in [d]} |\boldsymbol{G}_{i,j}||\boldsymbol{\Delta}_{i,j}| \leq \max_{j \in [d]} |\boldsymbol{G}_{i,j}| \sum_{j \in [d]} |\boldsymbol{\Delta}_{i,j}| \leq \gamma \max_{j \in [d]} |\boldsymbol{G}_{i,j}| = \boldsymbol{G}_i \cdot \boldsymbol{\Delta}_i^\gamma.$$

Thus, $\boldsymbol{\Delta}_i^\gamma$ is an optimal solution. □

We refer to fine-tuning with Eq. 31 as *NUDGE-S*. Observe that $\boldsymbol{\Delta}^\gamma$ as defined in Eq. 31 is very sparse, with only one non-zero entry per row. This means updating data embeddings as $\boldsymbol{D} + \boldsymbol{\Delta}^\gamma$ performs a sparse update. It specifically only changes the embedding of a single dimension of every data record. As our experiments in Sec. E.6 show, this is not beneficial in most settings we consider, since we use dense embeddings. Since the embeddings are dense, large modifications on a single dimension independent of other dimensions will likely lead to poor embeddings. This is because we expect embedding dimensions to be co-dependent, and large modifications in a single embedding dimension can distort the semantics of the embedding space. Nonetheless, we expect the above to be beneficial if the embedding space is sparse. We provide an experimental analysis of NUDGE-S in Appx. E.6.

Overall, the above shows how constraints shape the embedding updates performed by NUDGE, where different constraints can lead to different fine-tuning update rules (compare Eq. 31 with Eqs. 3

---

[2]Recall that $\boldsymbol{X}_{i,j}$ is the element in the $i$-th row and $j$-th column of $\boldsymbol{X}$

and 6). We expect different constraint forms to be beneficial in different settings, depending on the geometry of the embedding space. NUDGE, through the use of constraints in its optimization, provides a framework to incorporate knowledge about the embedding space in the fine-tuning process.

## C.2 ITERATIVE VARIANTS

Besides defining constraint optimization problem variants within the NUDGE family, here, we also consider NUDGE variants that use different optimization methods to solve the optimization problem. We consider variants that use iterative optimization so that they are more flexible in their optimization approach but do not provide closed-form optimal solutions.

Specifically, the NUDGE variants discussed in the paper can be difficult to implement (especially NUDGE-N) and are not flexible, e.g., considering other accuracy metrics or constraints requires a new theoretical study. Here, we discuss simple NUDGE variants that use gradient descent to optimize MaxS-EFT (e.g., to solve the inner optimization in BiMax-M) and hyperparameter tuning to optimize MaxA-EFT (e.g., to solve the outer optimization in BiMax-M). This approaches can be less efficient and suboptimal, but can still provide accurate solutions, and are simple and flexible. In such NUDGE variants, the learning rate and number of iterations act as knobs constraining how much the embeddings change.

**NUDGE-IM**. First, note that MaxS-EFT is equivalent to minimizing the following loss

$$\mathcal{L} = - \sum_{i \in [n_T]} \boldsymbol{Q}_i \cdot (\boldsymbol{D}_{Y_i} + \boldsymbol{\Delta}_{Y_i}). \tag{32}$$

NUDGE-IM performs gradient descent on loss in Eq. 32 with learning rate $\alpha$ for $t$ iterations ($\alpha$ and $t$ determined through hyperparameter tuning to maximize validation accuracy) with normalized gradients:

$$\boldsymbol{\Delta}^{(0)} \leftarrow \boldsymbol{0}, \ \boldsymbol{\Delta}^{(t)} \leftarrow \boldsymbol{\Delta}^{(t-1)} - \alpha \frac{\nabla_{\boldsymbol{\Delta}} \mathcal{L}}{\|\nabla_{\boldsymbol{\Delta}} \mathcal{L}\|}.$$

**Lemma 5.** *NUDGE-IM finds a solution equal to NUDGE-M, and thus optimally solves BiMax-M, whenever $\alpha t = \gamma^*$, where $\gamma^*$ is the optimal $\gamma$ value found by NUDGE-M.*

*Proof.* Observe that $\boldsymbol{G}_i = -\nabla_{\boldsymbol{\Delta}_i} \mathcal{L}$, and therefore the gradient remains constant through optimization. Thus, after $t$ iterations, we have

$$\boldsymbol{\Delta}_i^{(t)} = \alpha t \mathbb{I}[\boldsymbol{G}_i \neq 0] \frac{\boldsymbol{G}_i}{\|\boldsymbol{G}_i\|}.$$

Thus, setting $\gamma^* = \alpha t$, we obtain the same results as NUDGE-M. $\square$

The above lemma implies using gradient descent with suitable $\alpha$ and $t$ can also provide accurate solutions, but at the cost of efficiency due to iterative updates and hyperparameter tuning (instead of using the closed-form solutions), and the added challenge of finding suitable $\alpha$ and $t$. Appx. E.5 presents an experimental study of these trade-offs.

**NUDGE-IN**. Another alternative is *NUDGE-IN, an iteratively normalized NUDGE variant*. NUDGE-IN is similar to NUDGE-IM but normalizes the embeddings after every update:

$$\boldsymbol{D}_i^{(0)} \leftarrow \boldsymbol{D}_i, \quad \boldsymbol{D}_i^{(t)} \leftarrow \frac{\boldsymbol{D}_i^{(t-1)} - \alpha \frac{\nabla_{\boldsymbol{\Delta}_i} \mathcal{L}}{\|\nabla_{\boldsymbol{\Delta}_i} \mathcal{L}\|}}{\|\boldsymbol{D}_i^{(t-1)} - \alpha \frac{\nabla_{\boldsymbol{\Delta}_i} \mathcal{L}}{\|\nabla_{\boldsymbol{\Delta}_i} \mathcal{L}\|}\|} \quad \forall i \in [n].$$

NUDGE-IN ensure the fine-tuned embeddings are normalized similar to NUDGE-N. Although, there is no theoretical equivalency between the solutions of NUDGE-IN and NUDGE-N, we observed similar accuracy in practice. Meanwhile, NUDGE-IN is less efficient but simpler to implement.

## D MULTI-LABEL FORMULATION

In the main body of the paper, we provided fine-tuning solutions assuming each query has a single ground-truth answer. Here, we discuss how our results can be extended to a multi-label setting. We first formalize the problem setting with multiple labels and then discuss how to extend our results.

**Ground-Truth Answers**. For a query, $\boldsymbol{q}$, its ground-truth answer is a ranking of the data records so that the highest-ranked records are the most related to the query. This ranking can be represented

Table 9: NDCG@10 results for GTE-L on text datasets

| Method | ArguAna | Fever | HotpotQA | NF-Corpus | NQ | SciFact | TriviaQA |
|---|---|---|---|---|---|---|---|
| NUDGE-M | **52.4** (+0) | **95.6 (+8.6)** | **60.8 (+10.3)** | 45.2 (+7.9) | 50.6 (+9.5) | 82.4 (+7.9) | 40.1 (+9.9) |
| NUDGE-N | **52.4** (+0) | 93.1 (+6.1) | 57.0 (+6.4) | **46.9 (+9.6)** | **56.2 (+15.1)** | 88.3 (+13.8) | **45.0 (+14.7)** |
| Adapter | **52.4** (+0) | 87.0 (+0) | 53.0 (+2.5) | 38.5 (+1.1) | 41.1 (+0) | **87.8 (+13.4)** | 30.2 (+0) |
| No Fine-Tuning | **52.4** | 87.0 | 50.5 | 37.4 | 41.1 | 74.5 | 30.2 |

Table 10: NDCG@10 results for TE3-L on text datasets

| Method | ArguAna | Fever | HotpotQA | NF-Corpus | NQ | SciFact | TriviaQA |
|---|---|---|---|---|---|---|---|
| NUDGE-M | **42.4 (+0.1)** | **94.8 (+14.8)** | **65.2 (+14.2)** | **50.7 (+10.5)** | 56.6 (+13.0) | 86.5 (+11.1) | 46.1 (+13.1) |
| NUDGE-N | **42.4 (+0.1)** | 93.1 (+13.2) | 63.4 (+12.3) | 49.0 (+8.7) | **59.6 (+16.0)** | 89.8 (+14.4) | **50.1 (+17.1)** |
| Adapter | 41.7 (–0.6) | 88.7 (+8.8) | 54.5 (+3.4) | 42.1 (+1.8) | 42.4 (–1.2) | **90.9 (+15.5)** | 32.7 (–0.3) |
| No Fine-Tuning | **42.3** | 79.9 | 51.0 | 40.2 | 43.6 | 75.4 | 33.0 |

using *relevance scores*, which, for each data record, quantifies how related the record is to the query (relevance score zero means the record is unrelated). The ground-truth ranking can be obtained by sorting the data records based on their relevance scores. More formally, we represent the ground-truth ranking for a query with a *ground-truth rank index set*, $y = \{y_1, ..., y_p\}$ and corresponding *ground-truth relevance score set* $r = \{r_1, ..., r_p\}$ for some integer $p$ denoting the number of data records that are related to the query; the remaining records are not relate to the query. This means that record $\bar{D}_{y_i}$ has relevance $r_i$ to the query, and any record index not present in $y$ is assumed to have zero relevance score. Thus, sorting the dataset based on $r$ gives the ground-truth answer to the query. We drop the set $r$ if $|y| = 1$, or if $r_1 = ... = r_p = 1$.

**Fine-Tuning Query Sets**. For fine-tuning, a query set and corresponding ground-truth answers are available, consisting of queries $\bar{Q}$ and ground-truth answer sets $Y, R$, where for the $j$-th query $\bar{q}_j \in \bar{Q}$, $Y_j$ is the ground-truth index set and $R_j$ is the ground-truth relevance score set for $\bar{q}_j$. We assume this training set is split into two, a training set $\bar{Q}^T, Y^T, R^T$ and a validation set $\bar{Q}^V, Y^V, R^V$, with $n_T$ and $n_V$ queries respectively. Similar to single label setting, let $\boldsymbol{Q}^T \in \mathcal{R}^{n_T \times d}$ and $\boldsymbol{Q}^V \in \mathcal{R}^{n_V \times d}$ be matrices containing embeddings for training and validation queries.

**Problem Formulation**. Both MaxS-EFT and MaxA-EFT can be modified to utilize multiple labels. For MaxS-EFT we can change the objective to $\sum_{i \in [n_T]} \sum_{j^* \in Y_i^T} \boldsymbol{Q}_i^T \cdot (\boldsymbol{D}_{j^*} + \boldsymbol{\Delta}_{j^*})$, where summation over $Y_i^T$ can optionally be weighted by relevance scores. For MaxA-EFT, we can adjust the inequalities in the definition of the correct answer to a query (i.e., Eq. 1), so that for a query $\boldsymbol{q}$ with two relevance scores $r_1$ and $r_2$, $r_1 > r_2$, and for $R_1$ and $R_2$ containing document indexes with $r_1$ and $r_2$ relevance scores, we say $\boldsymbol{q}$ is answered correctly when

$$\boldsymbol{q} \cdot (\boldsymbol{D}_i + \boldsymbol{\Delta}_i) > \boldsymbol{q} \cdot (\boldsymbol{D}_j + \boldsymbol{\Delta}_j), \ i \in R_1, j \in R_2. \tag{33}$$

**NUDGE**. To solve BiMax-M and BiMax-N, observe that the above modifications cause marginal changes for `MaxS-M` and `MaxS-N`, and only require modifying the definition of $\boldsymbol{G}$ in the corresponding optimal solutions. However, the solutions to the outer optimization problem in BiMax-M and BiMax-N require further modifications since now a different set of inequalities needs to be solved to find the range of $\gamma$ for which a query is answered correctly. However, each inequality is still of the same form as before (compare Eq. 33 with Eq. 1), and thus, the same methodology applies.

# E  ADDITIONAL EXPERIMENTS AND DETAILS

Here we present additional experimental details and results:

- Appx. E.1 discussed details on the implementation of Adaptor and PTFT, including hyper-parameter tuning, loss, and efficiency considerations.

- Appx. E.2 contains detailed per dataset results summarized in the paper's main body.

- Appx. E.3 presents experiments on the training processes of Adaptors and PTFT to understand their failure modes.

- Appx. E.4 provides an ablation study of various normalization methods in NUDGE.

- Appx. E.5 provides an experimental comparison between NUDGE-M and its corresponding iterative variant NUDGE-IM.
- Appx. E.6 provides an experimental evaluation of alternative constraint variants.
- Appx. E.7 studies the impact of noise on query embeddings during fine-tuning.
- Appx. E.8 studies combining NUDGE with query rewriting methods.
- Appx. E.9 presents additional out-of-distribution generalization results.
- Appx. E.10 presents micro-benchmark comparing CPU and GPU runtimes.
- Appx. E.11 presents an ablation study of NUDGE.
- Appx. E.12 studies the impact of label distribution on NUDGE's accuracy.

### E.1 ADAPTOR AND PTFT DETAILS AND HYPER-PARAMETER TUNING

**-L loss.** The results presented with -L suffix use a modified version of MNR loss. The MNR loss, for a batch of $b$ queries, $\boldsymbol{Q}^B$, and positive examples (i.e., data records that the correct answer to queries) $\boldsymbol{D}^B$, where $\boldsymbol{D}_i$ is a positive example for $\boldsymbol{Q}_i$ for any $i \in [b]$, treats every other example in the batch as negative examples for $\boldsymbol{Q}_i$. Let $\mathcal{S}_{i,j} = \frac{\boldsymbol{Q}_i \cdot \boldsymbol{D}_j}{\|\boldsymbol{Q}_i\|\|\boldsymbol{D}_i\|}$ for any $i, j \in [b]$. Then, MNR loss minimizes

$$\mathcal{L}_{MNR} = -\sum_{i \in [b]} \log \frac{e^{\tau \mathcal{S}_{i,i}}}{\sum_{j \in [b]} e^{\tau \mathcal{S}_{i,j}}},$$

for some temperature parameter $\tau$. Our modified loss, for the $i$-th query, ignores samples, $j$, for which $\mathcal{S}_{i,j} < \eta$ for some threshold $\eta$:

$$\mathcal{L}' = -\sum_{i \in [b]} \log \frac{\mathbb{I}[\mathcal{S}_{i,i} \geq \eta] e^{\tau \mathcal{S}_{i,i}}}{\sum_{j \in [b]} \mathbb{I}[\mathcal{S}_{i,j} \geq \eta] e^{\tau \mathcal{S}_{i,j}}}.$$

This follows the intuition that, at the fine-tuning stage, the model only needs to get better at distinguishing between records that have high similarity and is already accurate enough to separate relevant from non-relevant items. $\tau$ and $\eta$ are related and are jointly set through hyperparameter tuning.

**Modeling details and hyper-parameter Tuning.** For both Adaptor and PTFT, we did hyperparameter tuning to determine the learning rate (and use of a scheduler), batch size (although for PTFT it is bottlenecked by GPU memory size), number of training steps, model architecture and initialization (for Adaptor, we tried linear up to 8 layer MLPs) and which layers to train (for PTFT, we tried training the full model or training the last layer), and the choice and parameters of the loss function. We only performed hyper-parameter tuning for BGE-S, and used the resulting hyper-parameters for other models. We note that the choice of initialization is particularly important for Adaptor, and we observed a significant advantage to ensuring that at initialization, Adaptor is an identity function. For a single-layer adaptor, this can be achieved by setting the weight matrix to the identity matrix (also done by Llama Index (LlamaIndex, 2024c)). For multi-layer adaptors, using ReLU activation, assuming queries are normalized (so that a query input $q \geq -1$), we achieve this by setting the weights to identity, the bias of first layer to **+1**, the bias of last layer to **-1** and other biases to **0**. We are unaware of this initialization being used by existing work for multi-layer adaptors, and we observed benefits to using this initialization over initialization that modify the embeddings at initialization (e.g., setting all biases to **0**).

**Efficient Implementation.** For Adaptor, our implementation is based on Llama Index LlamaIndex (2024b) and for PTFT based on Sentence Transformers Aarsen (2024), but with additional considerations for initialization (for Adaptor, see above), loss function (see above) and efficiency. We observed that validation passes (for model checkpointing) often take longer than training passes (because the number of data records is often more than the number of training queries in our datasets), we used hyperparameter tuning to set the validation frequency to as low as possible without affecting final accuracy. For Adptor, when applying Adaptor only to queries (so that data records don't get embedded), we also used a vector index (Faiss library Douze et al. (2024)) but did not observe any speed-ups (perhaps due to the already parallelizable nature of answering batched queries, and that we only do top-1 lookup during validation). For PTFT, a single validation pass, which requires re-embedding the entire dataset, can take more than an hour on our large datasets (Fever, NQ, TriviaQA, HotpotQA). To reduce the computational cost, for each validation query, we selected its top-10

Table 11: NDCG@10 results for CLIP-B on image datasets

| Method | COCO | Flickr |
|---|---|---|
| NUDGE-M | **29.7 (+11.6)** | **52.2 (+16.5)** |
| NUDGE-N | **29.9 (+11.9)** | **52.4 (+16.7)** |
| Adapter | 20.0 (+2.0) | 39.8 (+4.1) |
| No Fine-Tuning | 18.0 | 35.7 |

Table 12: NDCG@10 results for CLIP-L on image datasets

| Method | COCO | Flickr |
|---|---|---|
| NUDGE-M | **31.4 (+9.0)** | **55.0 (+12.5)** |
| NUDGE-N | **31.5 (+9.1)** | **55.1 (+12.6)** |
| Adapter | 25.0 (+2.6) | 48.0 (+5.5) |
| No Fine-Tuning | 22.4 | 42.5 |

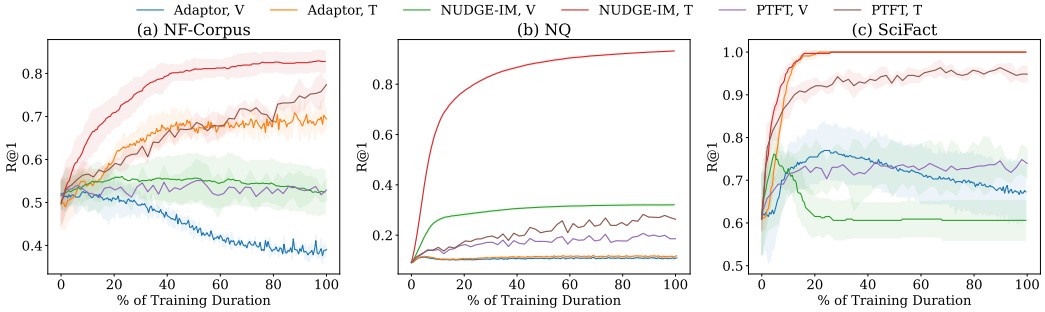

Figure 3: Training and validation accuracy for BGE-S on three datasets

answers based on the pre-trained model and only included those (in addition to ground-truth training and validation answers) data records in the dataset for validation. This provided more than 10x speed-up for validation on large datasets, and we observed similar final accuracy (intuitively, this removes data records from validation that, unless the pre-trained model significantly changes, are unlikely to impact validation accuracy). Finally, we use early stopping if validation accuracy drops by more than 5% compared with the maximum it had achieved.

## E.2 OTHER PER DATASET RESULTS

Tables 9-12 present the per dataset results for the embedding models GTE-L, TE3-L, CLIP-B and CLIP-L. The tables (in addition to Table 5) present the detailed results from which Tables 3-4 are generated.

## E.3 TRAINING AND VALIDATION ACCURACY DURING TRAINING

To better understand the differences and failure modes of the approaches, Fig. 3 shows the validation and training accuracy for Adaptor, PTFT and NUDGE-IM, where T and V respectively signify training and validation sets.

Fig. 3 (a) and (b) show two failure modes for Adaptor, overfitting, and underfitting. Specifically, Fig. 3 (a) shows NUDGE provides much better validation accuracy compared with Adaptors at the same training accuracy, suggesting that Adaptor simply overfits to the training set instead of learning generalizable patterns. Fig. 3 (b) shows the other end of the spectrum, where Adaptor fails to fit the training set at all (while NUDGE both fits the training set and improves validation accuracy). We observed this behavior on large data and query sets. We also observed (but not shown here) that increasing the number of parameters, e.g., by introducing additional layers, did not improve accuracy, suggesting that perhaps using adaptors is a wrong modeling choice. Fig. 3 (c) shows the only dataset where Adaptor performs well, where it both fits the training set and improves validation accuracy.

PTFT, on the other hand, has a much smaller generalization gap compared with Adaptor. However, both validation and training accuracy increase at a much slower pace, and eventually plateau. Especially for NQ, we observe that the model underfits the training set. We hypothesized that one reason could be due to the loss function used, where indeed Table. 5 shows our attempt at modifying the loss function does help improve accuracy on NQ, but not consistently across datasets (and worsens the accuracy on other datasets).

## E.4 NORMALIZATION ABLATION STUDY

We compare NUDGE-N, with NUDGE-IN (as described in Sec. C.2) with two other potential variants to understand the impact of normalization. NUDGE-M+N is a variant that first performs

Table 13: Normalization Study, avg. BGE-S accuracy

| Method | NDCG@10 |
|--------|---------|
| NUDGE-N | **61.1** |
| NUDGE-IN | **61.8** |
| NUDGE-M+N | 58.8 |
| NUDGE-IM+R | 52.4 |

Table 14: Accuracy/efficiency of iterative and non-iterative NUDGE variants (BGE-S on text datasets)

| Method | R@1 | R@10 | NDCG@10 | Fine-Tuning Time (s) |
|--------|-----|------|---------|----------------------|
| NUDGE-IM-$(10^{-4}, 10^3)$ | **50.4** | **66.5** | **57.2** | 11.3 |
| NUDGE-IM-$(10^{-3}, 10^2)$ | **50.4** | **66.5** | 57.1 | 4.16 |
| NUDGE-IM-$(10^{-2}, 10)$ | 48.0 | 59.2 | 51.1 | **3.46** |
| NUDGE-M | **50.5** | **66.6** | **57.3** | **3.49** |

NUDGE-M and then normalizes embeddings post-hoc. NUDGE-IM+R is a variation of NUDGE-IM with L2 regularization added to the loss to penalize embeddings with large norms. Table 13 shows the results for this experiment, showing that normalizing embeddings after optimization, i.e., NUDGE-M+N performs worse than when normalization is considered as part of optimization, which is the case for both NUDGE-IN and NUDGE-N. NUDGE-IM+R performs worse than all methods, showing an advantage for enforcing a normalization constraint over L2 regularization. Meanwhile, NUDGE-IN and NUDGE-N perform similarly (see Sec. C.2 for a discussion between the two).

### E.5    ITERATIVE VS. CLOSED-FORM NUDGE VARIANTS

We present results comparing NUDGE-M and NUDGE-IM. We use NUDGE-IM-$(\alpha, t)$ to refer to NUDGE-IM with learning rate $\alpha$ run for $t$ iteration (although we use model checkpointing based on validation accuracy, so the presented accuracy can be for a model trained with fewer iterations than $t$). Overall, the results show that NUDGE-M saves time while providing the best accuracy, while some NUDGE-IM variants are as accurate while being less efficient. Nonetheless, NUDGE-IM variants are simple to implement and more flexible (e.g., in terms of validation accuracy metric to use), and thus may be preferred in some applications.

### E.6    NUDGE CONSTRAINT VARIANTS

#### E.6.1    SPARSITY CONSTRAINTS

We evaluate the accuracy of the NUDGE variant discussed in Appx. C.1 which performs sparse embedding updates in practice. Specifically, we apply the NUDGE-S update formula in Eq. 31 to fine-tune the embeddings with various values of $\gamma \in [0, 0.5]$ for various text datasets using BGE-S. The results of this experiment are shown in Fig. 4.

Overall, as the figure shows, in almost all datasets, updating the embeddings with sparse updates worsens accuracy, with drop of NDCG for any $\gamma > 0$. This is expected since the updates change only a single dimension of the embedding of every data record. Since the embeddings are dense vectors where different dimensions are expected to be dependent on each other, modifying only a single embedding dimension can distort the semantics in the embedding vectors and worsen accuracy. Interestingly, the only exception is ArguAna, where NUDGE-S indeed improves accuracy. This contrasts with NUDGE-M and NUDGE-N (as well as PTFT and Adaptor, See Table 5) that do not improve accuracy on ArguAna. This suggests that embedding space for ArguAna has different characteristics, and emphasizes the relationship between the regularization method and the embedding geometry.

#### E.6.2    ADAPTIVE THRESHOLDS

The constraint formulation for NUDGE-N and NUDGE-M enforces a uniform bound of $\gamma$ on the magnitude of change of embeddings across the embedding space. Here, we explore whether al-

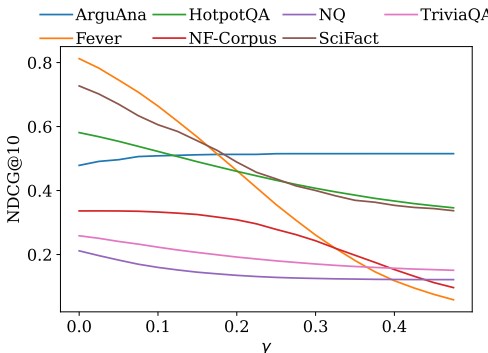

Figure 4: Fine-Tuning with NUDGE-S

Table 15: Using adaptive thresholds across datasets with BGE-S

| Methods | NFCorpus | SciFact | ArguAna | Fever | NQ | TriviaQA | HotpotQA |
|---|---|---|---|---|---|---|---|
| NUDGE-M | 40.9 | 79.2 | **47.9** | **95.0** | 38.6 | 35.5 | 63.1 |
| NUDGE-M-C2 | 34.5 | 76.5 | **47.9** | 87.3 | 31.8 | 30.3 | 61.3 |
| NUDGE-M-C5 | 34.4 | 75.7 | **47.9** | 86.1 | 30.9 | 30.2 | 61.2 |
| NUDGE-M-C10 | 34.0 | 74.7 | **47.9** | 85.7 | 30.1 | 29.8 | 61.0 |
| NUDGE-N | **45.7** | **87.7** | **47.9** | 93.5 | **45.9** | **43.0** | **65.2** |
| NUDGE-N-C2 | 43.1 | **87.7** | **47.9** | 93.5 | **45.9** | **42.8** | **65.1** |
| NUDGE-N-C5 | 41.0 | 86.9 | **47.9** | 93.2 | **45.9** | **43.0** | **65.2** |
| NUDGE-N-C10 | 39.5 | 86.6 | **47.9** | 93.0 | **46.0** | **42.9** | **65.0** |

lowing for adaptive thresholds can improve accuracy, where different parts of the embedding space are allowed to use different bounds on the magnitude of change to the embeddings. To do so, we first cluster the data embeddings using K-Means clustering into $r$ clusters for an integer $r$. Then, we iteratively apply NUDGE-M and NUDGE-N to each cluster. This provides the model with the flexibility to set a different threshold $\gamma$ for different clusters.

The result of this experiment is presented in Table 15 for text datasets using BGE-S. NUDGE-M-C$r$ and NUDGE-N-C$r$, refer, respectively, to NUDGE-M and NUDGE-N applied to $r$ different clusters as described above. Interestingly, we see that using clustering worsens the performance, especially for NUDGE-M, where the more clusters are used, the worse the model gets. We suspect this is because fine-tuning is done on each cluster separately, and thus, the $\gamma$ values across clusters are set independently. As a result, this leads to a suboptimal selection of $\gamma$ values for different clusters. We expect an optimal solution to the problem to be non-trivial, given that $\gamma$ values for all the clusters need to be set co-dependently. We leave such a study to the future work. We also note that, as shown in Fig. 2, NUDGE-N is less sensitive to the value of $\gamma$, thus, its accuracy is only marginally impacted by the use of clustering.

### E.7 IMPACT OF NOISE IN TRAINING QUERIES

In this experiment we study the impact of unreliable training and validation queries on the fine-tuning outcome. Specifically, for each training/validation query embedding $q$, we add $\mathcal{N}(0, \sigma^2)$ noise, independently to every dimension of $q$. We vary $q$ to study the impact of noise magnitude on the behavior of NUDGE-M and NUDGE-N. The result of this experiment, varying $\sigma \in [0.025, 0.05, 0.1, 0.2, 0.5]$ is plotted in Fig. 5.

The results show that both NUDGE variants still outperform No Fine Tuning, even up until noise with standard deviation 0.2. Note that noise with standard deviation of 0.2 has average absolute value of around 0.16, while the average absolute value of the training query embeddings across all datasets is less than 0.04, showing that the fine-tuning processes is robust to i.i.d and unbiased noise. Interestingly, our results also show that NUDGE-M is more robust to noise than NUDGE-N, and starts to outperform NUDGE-N for noise with standard deviation larger than 0.05. We believe this can be because normalization forces the embeddings to fall in a smaller subset of space. This can

Table 16: NUDGE with HyDE using BGE-S embeddings

| Method | NFCorpus | SciFact |
|---|---|---|
| NUDGE-M | 40.0 (6.2) | 79.2 (6.5) |
| NUDGE-N | 44.6 (10.8) | 87.8 (15.1) |
| HyDE+NUDGE-M | 46.3 (12.5) | 80.9 (8.2) |
| HyDE+NUDGE-N | **49.6 (15.8)** | **88.5 (15.8)** |
| HyDE | 35.5 (1.7) | 74.8 (2.1) |
| No Fine-Tuning | 33.8 | 72.7 |

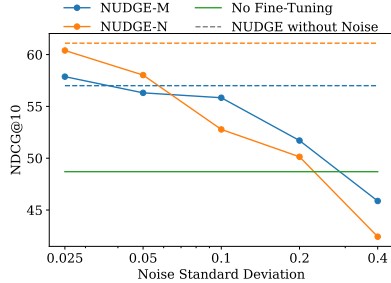

Figure 5: Impact of noise in training queries, average NDCG@10 across text datasets using BGE-S.

force unrelated embeddings to become more entangled in the presence of noise, leading to worse accuracy.

### E.8 USING QUERY REWRITING WITH NUDGE

Here we explore the potential of using NUDGE together with query rewriting approaches. Since NUDGE only modifies data embeddings, it is orthogonal to query rewriting approaches and can easily be combined with such methods. In this experiments, we study impact of combining NUDGE with HyDE Gao et al. (2022). HyDE is a popular query rewriting method that rewrites the query as *hypothetical documents* that answer the query using an LLM (we used GPT-4o mini), and uses the generated hypothetical documents as queries to retrieve relevant documents from the dataset. Note that HyDE changes how the query embeddings are created, and operates completely independently of NUDGE. Thus, in this experiment, HyDE is used to generate the query embeddings (i.e., the embedding matrices $Q^T$ and $Q^V$ are now generated using HyDE). NUDGE is then directly applied on the generated embeddings as before, without any modifications. We note that HyDE's query rewriting method is time-consuming (and expensive), as it requires multiple LLM generation calls for every query (HyDE in fact perform 8 different generations for every query and averages the embeddings). Thus, we only report results on two datasets, NF-Corpus and SciFact.

Table 16 shows the result of this experiment. We see that combining HyDE with NUDGE improves accuracy over only using NUDGE, although NUDGE alone is still more accurate than using HyDE alone.

### E.9 OTHER GENERALIZATION RESULTS

This section contains generalization results for other models and dataset in the same setting as Sec. 4.2. Specifically, Table 17 contains per dataset generalization results not presented in the paper and Table 18 contains generalization results using GTE-L model averaged over the text datasets. Overall, we see that the trends in Tables 18 and 17 are in general similar to the results presented in Tables 6 and 7. Here, we note that on HotpotQA, TriviaQA and Fever, both parametric approaches Adapter and PTFT are unable to improve accuracy over fine-tuning. This is consistent with the results in Table 5, and is further discussed and explained in Sec. 4.1. Thus, in the results presented for Adapter and PTFT in Table 17, we see zero change in accuracy over No Fine-Tuning.

### E.10 RUNTIME MICRO-BENCHMARK

This section presents a micro-benchmark, showing the runtime of matrix operations on GPU and CPU, in order to justify the runtime numbers shown in Table 8. Specifically, we perform 13 consecutive matrix addition and multiplication operations on matrices of dimensions $d \times d$, and measure

Table 17: Generalization results using BGE-S for three text datasets. Results report change in NDCG@10 over No Fine-Tuning (ID: In-Dist., OD: Out-of-Dist., L: Labeled, UL:Unlabeled)

| Method | HotpotQA | | | | Triviaqa | | | | Fever | | | |
|---|---|---|---|---|---|---|---|---|---|---|---|---|
| | IQ | OQ | IL | OL | IQ | OQ | IL | OL | IQ | OQ | IL | OL |
| NUDGE-M | 4.1 | **0.5** | **25.0** | -5.2 | 9.1 | -22.2 | **49.4** | -13.7 | **12.5** | -40.1 | **14.0** | 6.1 |
| NUDGE-N | **5.5** | **0.9** | 19.9 | 0.6 | **16.6** | **1.6** | 41.1 | **3.1** | 11.5 | **4.1** | 12.4 | **9.7** |
| NUDGE-M+N | 2.0 | 0.4 | 8.2 | 0.9 | **16.7** | **1.2** | 42.0 | **2.8** | 10.2 | 3.4 | 10.6 | **9.5** |
| A+NUDGE-N | **5.5** | **0.9** | 19.9 | 0.6 | **16.6** | **1.6** | 41.1 | **3.1** | 11.5 | **4.1** | 12.4 | **9.7** |
| P+NUDGE-N | **5.5** | **0.9** | 20.9 | 0.2 | **16.6** | **1.6** | 41.1 | **3.1** | 11.5 | **4.1** | 12.4 | **9.7** |
| Adapter | 0.0 | 0.0 | 0.0 | 0.0 | 0.0 | 0.0 | 0.0 | 0.0 | 0.0 | 0.0 | 0.0 | 0.0 |
| PTFT | 0.0 | 0.0 | 0.0 | 0.0 | 0.0 | 0.0 | 0.0 | 0.0 | 0.0 | 0.0 | 0.0 | 0.0 |

Table 18: Distribution shift results on GTE-L, NDCG@10 averaged over text datasets

| Method | In-Distribution Query | Out-of-Distribution Query |
|---|---|---|
| NUDGE-M | 59.3 (+6.2) | 38.9 (-14.4) |
| NUDGE-N | 61.8 (+8.7) | 54.4 (+1.1) |
| Adapter | 55.2 (+2.1) | 52.8 (-0.5) |
| No Fine-Tuning | 53.1 | 53.3 |

the total runtime (average across 10 runs). Fig. 6 shows the runtime of the approaches for various values of $d$. We see that at $d = 384$, GPU is 2.3x faster, but at $d = 3072$ the GPU is 1600x faster. This justifies the less than expected speed-ups of using GPUs in Table 8, where due to embedding dimension being 384, most operations in NUDGE-M, NUDGE-N, and Adaptor involve only small matrix operations.

### E.11 ABLATION STUDY OF NUDGE

We provide an ablation study to better understand the impact of various constraints in NUDGE. In addition to NUDGE-M and NUDGE-N, we present NUDGE-M+N, which performs normalization on the output of NUDGE-M. That is, instead of incorporating normalization as a constraint in the optimization problem, NUDGE-M+N simply normalizes the embeddings after performing NUDGE-M. We also present NUDGE-NU, which is a variation of NUDGE-N that only includes the normalization constraint, but not the constraint on the magnitude of the change in the embeddings (the solution is equivalent to the first branch of Eq. 6).

Table 19 shows the results of comparing the above variants across text datasets using the BGE-S model. The table shows normalizing the embeddings as part of the optimization (NUDGE-N) is better than simply normalizing the output after optimization (NUDGE-M+N), although both work better than using unnormalized embeddings (NUDGE-M). Moreover, allowing the magnitude of change to the embeddings to be unbounded (NUDGE-NU) performs worse than all other variants, showing the benefit of constraining how much embeddings can change during fine-tuning. Overall, comparing NUDGE-N, No Fine-Tuning, and all other variants, we see that the benefits of NUDGE-N come from a combination of all design choices made.

### E.12 IMPACT OF LABEL DISTRIBUTION

Here, we study the impact of label distribution in the results presented in Sec. 4. To do so, Fig. 7 plots the distribution of test labels in relation to the training set. A point $(x, y)$ on the plot means that $y$ percent of test queries on the dataset have a label that appears $x$ times in the training set. For instance, all test queries for ArguaAna have ground truths unseen during training, while for CoCo and Flickr, all test queries have a ground truth that appeared between 1-4 times in the training set (i.e, there are 1-4 training queries with ground truth being the same data record as the test query).

Next, we discuss the relationship between label distribution and normalization to explain the performance difference between NUDGE-M and NUDGE-N. Note that if a data record is not a ground truth to any training query, NUDGE-M will not modify its embedding. On the other hand, NUDGE-M will likely increase the magnitude of the data embeddings that are the ground truth for some training query. This can create a discrepancy between the magnitude of different data embeddings in NUDGE-M depending on whether they were present in the training set. Embeddings with smaller

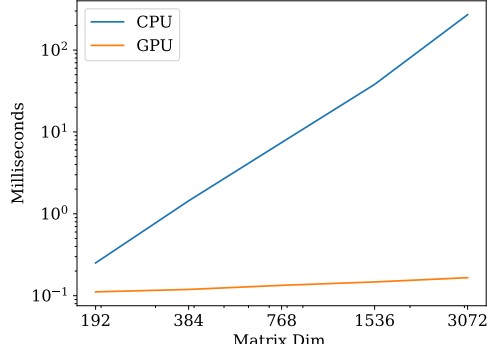

Figure 6: Runtime micro-benchmark

Table 19: Ablation of NUDGE

| Method | NDCG@10 |
|---|---|
| NUDGE-N | **61.1** |
| NUDGE-M+N | 58.8 |
| NUDGE-M | 57.0 |
| NUDGE-NU | 56.6 |
| No Fine-Tuning | 48.7 |

magnitudes will generally have lower inner product similarity with test queries, which is detrimental if many test queries ask for data embeddings whose ground truth was not seen during training. The exact impact depends on whether label distribution is heavy-tailed (with many test labels not present in training set) or if it is skewed towards a fixed set of labels (with test labels likely seen during training). For example, we see a large gap between NUDGE-M and NUDGE-N in SciFact, TriviaQA, NQ and HotpotQA, where, as Fig. 7 shows, the ground truth for between 40-60% of test queries was not seen during training. Meanwhile, NUDGE-M performs similarly to NUDGE-N on CoCo, Flicker and Fever datasets, where, as shown in Fig. 7, almost no test query asks for data records not seen during training.

Regarding image datasets CoCo and Flickr, we also note that, all images have exactly 5 different queries. Thus, the ground truth for all test queries is present in the training set. This means NUDGE-M modifies the embedding of all data records during fine-tuning, leading to an increase in the magnitude of all embeddings. Interestingly, in both CoCo and Flickr datasets, our results show that the increase in the magnitude has low variation across embeddings. In fact, the standard deviation of the magnitude of the embeddings after fine-tuning is 0.02 (average embedding magnitude is 2.24 an 1.81 respectively for CoCo and Flickr datasets). Since there is a low variation in the magnitudes of the embeddings, normalization does not make a significant difference. As a result, NUDGE-M and NUDGE-N (which requires normalized embeddings) perform similarly.

Finally, Fig. 7 also helps explain why none of the approaches perform well on ArguAna, where no test label appears in the training set. As such, all models have to rely on generalization to improve accuracy, which is more challenging.

**Impact of label distribution on** $\gamma$. Here, we also discuss the impact of label distribution on the best value of $\gamma$ for NUDGE-M. Recall that, as we saw in Fig. 2, for NUDGE-M the best value of $\gamma$ changes more significantly across datasets compared with NUDGE-N. In fact, for NUDGE-M, we see an interesting relationship between $\gamma$ and label distribution. For the label distributions concentrated on a fixed set of labels (i.e., CoCo, Flickr, Fever, NF-Corpus without test queries asking for labels not seen during training), the larger values of $\gamma$ do not hurt accuracy, since, as discussed above, larger magnitudes of fine-tuned embeddings in those dataset is not detrimental. Nonetheless, for other datasets, increased $\gamma$ can worsen accuracy for test queries that ask for data records not seen during training. This leads to a trade-off, where increasing $\gamma$ can improve accuracy for queries on data records seen during training and worsen accuracy for records not seen during training. We see that this trade-off leads to different suitable $\gamma$ values for datasets SciFact, HotpotQA, NQ and TrivaiaQA.

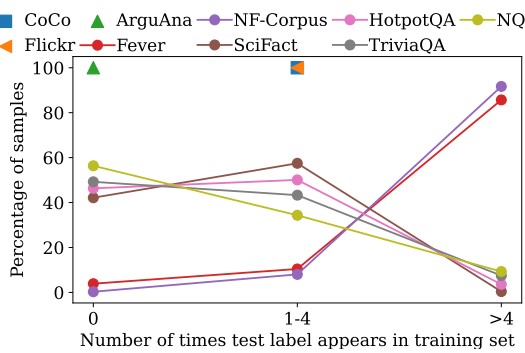

Figure 7: Label dist. across datasets

