# OpenReview forum: "NUDGE: Lightweight Non-Parametric Fine-Tuning of Embeddings for Retrieval"
_ICLR.cc/2025/Conference — ICLR 2025 Poster_

### Official Review · Reviewer_K1aL · 2024-10-17

**Soundness:** 3
**Presentation:** 3
**Contribution:** 3
**Rating:** 8
**Confidence:** 5

**Summary:**

This paper introduces NUDGE, a novel method for retrieval that directly fine-tunes index embeddings by treating them as "model parameters". The theoretical analysis yields two variants, NUDGE-M and -N, and elucidates their time complexities. Experimental results demonstrate that NUDGE significantly outperforms existing methods for in-distribution data and achieves comparable performance for out-of-distribution scenarios.

**Strengths:**

[S1] The proposed method is novel, simple, and easy to implement.

[S2] The experiments effectively demonstrate the efficacy of NUDGE for in-distribution data.

[S3] The theoretical analysis clearly delineates the characteristics of NUDGE-N and NUDGE-M.

[S4] The paper is well-structured and easy to follow.

[S5] NUDGE represents a potential new paradigm for fine-tuning.

[S6] The authors have provided the code with their submission.

**Weaknesses:**

[W1] Fine-tuning the embedding directly makes the database difficult to update. Specifically, if new photos are added to the database, their embeddings must also be trained. This requires relabeling (or pseudo labeling) at least a few queries whose GT answers include the new DB images, which incurs additional cost (or the new DB embedding will not be updated). This weakness, however, does not appear in previous two lines of works, ie. fine-tuning the foundation model, or training a new lightweight model to refine the embeddings. Consequently, this limitation somewhat impacts the practicality of NUDGE.



[W2] NUDGE is unable to refine query embeddings. It has been well established that query embedding refinement can significantly enhance retrieval performance (as evidenced by query expansion and its variants). Both fine-tuning the foundation model and training an embedding update model are capable of updating query embeddings. However, NUDGE cannot accomplish this because the GT label for the input query is unknown, making it impossible to train. This limitation leaves considerable room for improvement.



[W3] The generalizability of the proposed method is questionable. Intuitively, embedding update methods are likely to suffer from significant overfitting issues. The authors have invested considerable effort to address this, including constraining the $\Delta$ and normalizing the updated embeddings (either pre- or post-update). While NUDGE-N ultimately achieves acceptable results for OOD data, overall confidence in its generalizability remains limited. For OOD scenarios, NUDGE-N shows only minor improvements on the BGE-S benchmark, with many other benchmarks left untested. When the embedding is not normalized (i.e., NUDGE-M), the generalizability appears notably poor. While it's plausible that the transition from NUDGE-M to NUDGE-N could yield such significant improvements, additional experiments elaborating on generalizability would be beneficial, as this might constitute a major drawback of NUDGE. Nevertheless, if NUDGE can perform adequately across most OOD benchmarks, or even in its current form, it may still be considered acceptable given the other strengths of this work.



[W4] Nit: The authors should follow the ICLR 2025 template guidelines by placing table captions above the tables.

**Questions:**

[Q1] It seems appropriate to position NUDGE as the third paradigm for retrieval task fine-tuning. The three paradigms can be summarized as follows: 1) Fine-tuning the foundation model; 2) Training a tiny model to refine the embedding; 3) directly fine-tuning the embedding. These approaches present interesting trade-offs. In terms of training cost, the order is 1 > 2 > 3, while for cost of adding more photos to the DB, the order is reversed: 3 > 2 > 1. Fine-tuning a foundation model is computationally intensive, but it easily accommodates new images added to the database. NUDGE, on the other hand, is less demanding to train, but requires additional computation to update embeddings for newly added DB images. In this context, the second paradigm appears to offer the best balance among the three. Do you have any insights on this?



[Q2] How is the generalizability of other NUDGE variants?



Justification of my score:

Although concerns about generalizability and practicality have been raised, the overall contributions of NUDGE—such as its efficient training, strong performance, and introduction of a new fine-tuning paradigm—significantly outweigh its weaknesses. Based on these considerations, this paper merits acceptance. However, given the identified limitations, I look forward to discussing with the authors and other reviewers to further evaluate its merits. At present, I believe NUDGE represents a valuable and striking contribution to the retrieval community. I hope the authors can address my concerns or, at the very least, discuss them as limitations of this work in future revisions.

---

> ### Author Response · Authors · 2024-11-21
>
> We thank the reviewer for their comments. We have revised the paper to address all the reviewer's comments, as discussed below. Modifications are marked in blue in the revised paper.
>
> **Question [Q1].** "It seems appropriate to position NUDGE as the third paradigm for retrieval task fine-tuning. The three paradigms can be summarized as follows: 1) Fine-tuning the foundation model; 2) Training a tiny model to refine the embedding; 3) directly fine-tuning the embedding. These approaches present interesting trade-offs. In terms of training cost, the order is 1 > 2 > 3, while for cost of adding more photos to the DB, the order is reversed: 3 > 2 > 1. Fine-tuning a foundation model is computationally intensive, but it easily accommodates new images added to the database. NUDGE, on the other hand, is less demanding to train, but requires additional computation to update embeddings for newly added DB images. In this context, the second paradigm appears to offer the best balance among the three. Do you have any insights on this?"
>
> **Response**. Thank you for your comment. Overall, we agree with the reviewer that NUDGE is not designed to generalize to data records not seen in its training set. On the other hand, the parametric nature of both paradigms 1 and 2 mentioned by the reviewer provides a mechanism for the model to generalize beyond the labels seen during training, e.g., to accommodate new images added to the database.
>
> Nonetheless, we have included new experimental results in Sec. 4.2 that show, if such a generalization is needed in an application, NUDGE can be combined with parametric methods to inherit their generalization ability while providing a strong accuracy boost for data records that appear in the training set. Specifically, we studied the performance of A+NUDGE-N and P+NUDGE-N, two hybrid methods that first perform, respectively, Adaptor or PTFT on the embeddings, and apply NUDGE-N on top of the generated embeddings. The new experimental results using these methods in Table 6 (please see the updated Sec. 4.2 for details of this experiment and exact results) show that A+NUDGE-N and P+NUDGE-N are in fact able to provide the generalization benefits of Adapter and PTFT, while also maintaining significant accuracy boosts on in-distribution samples. This empirically confirms that NUDGE can be combined with existing approaches to provide the best of both worlds, i.e., improved accuracy on records present in the training set through NUDGE, and improved accuracy on records unseen during training through parametric methods. In relation to the reviewer’s ranking of 3 > 2 > 1 in the presence of insertions, our result suggests that indeed we have the ranking 3+1 > 2+1>3 > 2 , where 3+1  and 2+1 denote combining the non-parametric approach of NUDGE with the existing parametric method.
>
> Finally, we also point out that although PTFT and Adaptor do allow for generalization to new labels on some datasets, there is also a risk of overfitting on the training set for those methods. For example, our results in Table 7 show that (see Sec 4.2 for details), when measuring the accuracy on data records not seen during training, PTFT and Adaptor improve the accuracy on NFCorpus and NQ datasets (to being better than NUDGE) but also cause a significant drop in accuracy (which ends up being worse than NUDGE) on the SciFact dataset (we note that SciFact is our smallest dataset and generalization from such a small training set of <1000 queries is difficult). NUDGE, on the other hand, avoids such a performance regression. Avoiding such performance regressions is a benefit of NUDGE that allows for its safe deployment in practical settings, as it will not lead to unexpected behavior due to poor generalization when new data is inserted. To summarize, in relation to the reviewer's ranking of 3 > 2 > 1 in the presence of insertions, our experimental results show that although this may hold in most datasets, it depends on how well the model generalizes and it is difficult to know apriori. In some cases, we may have 1>2>3, with the accuracy of 1 significantly better than 2 and 3.
>
> **Question [Q2]**. "How is the generalizability of other NUDGE variants?"
>
> **Response**. We have now included three more NUDGE variants in Tables 6 and 7, including the two variants A+NUDGE-N and P+NUDGE-N discussed in response to Q1, as well as NUDGE-M+N. As discussed in our response to Q1, A+NUDGE-N and P+NUDGE-N improve the generalization ability of NUDGE.  Here, we discuss NUDGE-M+N.
>
> NUDGE-M+N normalizes data embeddings after applying NUDGE-M, so that it only differs from NUDGE-M by a normalization step. Tables 6 and 7 show  the results for generalization of NUDGE-M+N compared with NUDGE-M. Indeed, the result shows significantly better out-of-distribution generalization for NUDGE-M+N compared with NUDGE-M, showing that normalization is needed to achieve good out-of-distribution performance with NUDGE.

---

> > ### Author Response · Authors · 2024-11-21
> >
> > **Comment [W1]**. "Fine-tuning the embedding directly makes the database difficult to update. Specifically, if new photos are added to the database, their embeddings must also be trained. This requires relabeling (or pseudo labeling) at least a few queries whose GT answers include the new DB images, which incurs additional cost (or the new DB embedding will not be updated). This weakness, however, does not appear in previous two lines of works, ie. fine-tuning the foundation model, or training a new lightweight model to refine the embeddings. Consequently, this limitation somewhat impacts the practicality of NUDGE."
> >
> > **Response.** Thank you for your comment. We have provided a detailed discussion on generalization in our response to Q1. Here, we point out that there are many applications where the set of documents to be queried is relatively fixed, which creates a setting particularly well-suited for NUDGE. Some examples include a corpus of hospital or medical regulations, laws passed by the US Congress, the documentation for software, and textbooks on a specific topic. Journalists, legal experts, software users, and students frequently query such fixed corpora, and improving retrieval accuracy is beneficial in all such settings. Moreover, even if the corpus set changes, NUDGE is useful as long as the query distribution does not change. A journalist studying historical crime rates in a city still benefits from better retrieval of court cases in the past, even if new court cases are added to the corpus.  In fact, NUDGE has already been incorporated into a popular open-source library for RAG applications, showing the interest to include NUDGE in real-world applications for better retrieval.
> >
> > **Comment [W2].** "NUDGE is unable to refine query embeddings. It has been well established that query embedding refinement can significantly enhance retrieval performance (as evidenced by query expansion and its variants). Both fine-tuning the foundation model and training an embedding update model are capable of updating query embeddings. However, NUDGE cannot accomplish this because the GT label for the input query is unknown, making it impossible to train. This limitation leaves considerable room for improvement."
> >
> > **Response.** We thank the reviewer for pointing out query expansion techniques. Indeed, since NUDGE only modifies data embeddings, it is orthogonal to query rewriting and expansion techniques, and can be used together with such approaches out-of-the-box. To show this, we have added a new experiment, combining NUDGE with HyDE, a popular query rewriting technique. This experiment is included in the newly added Appendix E.8. The results in Table 16 show that incorporating HyDE within NUDGE improves the accuracy over using NUDGE alone (although NUDGE alone is more accurate compared with using HyDE without NUDGE).  This shows that, as the reviewer suggests, NUDGE can be combined with existing query rewriting methods to improve accuracy.
> >
> > Moreover, we also note that, as discussed in our response to Q1, NUDGE can be performed on top of embeddings created by PTFT and Adaptor, allowing NUDGE to combine query embedding refinements done by PTFT and Adaptor with improved data embeddings created by NUDGE.
> >
> > **Comment [W3].** "The generalizability of the proposed method is questionable. Intuitively, embedding update methods are likely to suffer from significant overfitting issues. The authors have invested considerable effort to address this, including constraining the Δ  and normalizing the updated embeddings (either pre- or post-update). While NUDGE-N ultimately achieves acceptable results for OOD data, overall confidence in its generalizability remains limited. For OOD scenarios, NUDGE-N shows only minor improvements on the BGE-S benchmark, with many other benchmarks left untested. When the embedding is not normalized (i.e., NUDGE-M), the generalizability appears notably poor. While it's plausible that the transition from NUDGE-M to NUDGE-N could yield such significant improvements, additional experiments elaborating on generalizability would be beneficial, as this might constitute a major drawback of NUDGE. Nevertheless, if NUDGE can perform adequately across most OOD benchmarks, or even in its current form, it may still be considered acceptable given the other strengths of this work."
> >
> > **Response.** Thank you for your comment. Our response to Q1 and Q2 provides a detailed discussion on generalization. Here, we additionally note that we have also included Table 18 in Appendix E.9, which presents generalization results using the GTE-L model. The table shows that similar observations on the BGE-S model also hold for GTE-L.
> >
> > **Comment [W4]**. "Nit: The authors should follow the ICLR 2025 template guidelines by placing table captions above the tables."
> >
> > **Response.** Thanks for pointing this out. We have moved the captions to above the tables.

---

> > > ### Comment · Reviewer_K1aL · 2024-11-22
> > >
> > > My concerns have been thoroughly addressed by the authors. I appreciate their efforts in conducting extensive experiments, which significantly enhance the quality of the work. I have also reviewed the weaknesses highlighted by other reviewers, noting that the concerns raised by Reviewer ZHaK align with mine (especially W1). During the rebuttal period, I did not identify any additional issues remained completely unsolved by authors. Taking all these factors into account, I am pleased to maintain my positive score and increase my confidence level to 5.
> > >
> > > I would greatly appreciate it if future works could thoroughly address the generalizability within the paradigm of embedding fine-tuning, as this would represent another valuable contribution to the community
> > >
> > > In my humble opinion, the key issue in deciding whether to accept this work lies in assessing if the relatively limited generalizability and challenges in adapting to new data are significant enough to warrant rejection. From my perspective, as the authors have demonstrated, these issues could be addressed by integrating other methods (albeit not in the most elegant way, but still a viable solution). Furthermore, the strengths—namely the innovative approach to fine-tuning for retrieval systems and the high efficiency—are notably compelling. Therefore, despite some weaknesses, I believe the strengths are substantial enough to justify acceptance. That said, I would not oppose a rejection if the ACs consider the weaknesses to be critical, though I personally do not find them to be so.

---

### Official Review · Reviewer_ZHaK · 2024-10-29

**Soundness:** 3
**Presentation:** 2
**Contribution:** 3
**Rating:** 6
**Confidence:** 4

**Summary:**

The context is embedding-based retrieval, for Q&A on documents, RAG, text-to-image retrieval, etc. The authors propose to do domain adaptation by "fine-tuning" the document embeddings with a supervision. The results show that this is more accurate (and faster) than to fine-tune the feature extraction model or to train a transformation model on top of frozen features.

**Strengths:**

S1. The study is an interesting exploration of dataset side adaptation, in the case of a "closed world" dataset that is frozen.

S2. The approach and baselines are relevant. The adaptation relies on simple linear algebra, so it is fast and much can be done in closed form.

S3. Convincing results: since the training acts directly on the embeddings (as opposed to network parameters) it has more degrees of freedom than when training on the model (indirect fine-tuning)

**Weaknesses:**

W1. The "elephant in the room" for this method is that it does not work on out-of-sample documents. Since the embeddings of the database are adjusted at fine-tuning time, new documents cannot benefit from it, unless new training data is added for them as well, which becomes a really narrow application setting. This is all the more apparent since out-of-distribution on the queries side is considered in this work.

W2. The method works only on densely annotated training data, since a document that is never returned to a query will not be adapted. This is apparent in the KILT dataset where documents that do not appear in any result are simply removed.

W3. Because of this particular setting, the queries in experiments must be split in train+val+test, which is unusual. Therefore there are no comparisons with state-of-the-art methods.

**Questions:**

I would be willing to raise my rating if the "closed-word" setting of the paper was clearly discussed and mentioned as a limitation compared to the baselines it compares with. It is not even sure it's legitimate to call the method "fine-tuning". However, it *is* an interesting study of what can happen if embeddings are optimized directly (it's just not practical beyond a benchmark setting).

For out-of-sample results, it would be interesting to see what would happen if an actual feature transformation model was trained on top of the embeddings found by this method.

**Details Of Ethics Concerns:**

jylh.kj

---

> ### Author Response · Authors · 2024-11-21
>
> We thank the reviewer for their comments. We have revised the paper to address all the reviewer's comments, as discussed below. Modifications are marked in blue in the revised paper.
>
> **Question Q1**.  “I would be willing to raise my rating if the "closed-word" setting of the paper was clearly discussed and mentioned as a limitation compared to the baselines it compares with … it is an interesting study of what can happen if embeddings are optimized directly (it's just not practical beyond a benchmark setting).”
>
> **Response**. First, we agree with the reviewer regarding the closed-world setting of NUDGE, and we have modified the introduction and Sec. 3.3 to explicitly state this as requested by the reviewer. We have also extended our out-of-distribution experiments in Sec. 4.2 to clearly evaluate all methods both when this “closed-world“ setting holds and also when it doesn’t. Thus, Sec. 4.2 presents a thorough discussion of both the benefits and limitations of NUDGE and existing work. Overall, NUDGE, clearly, only modifies the data embeddings for which training labels are available, and does not make any modifications to unlabeled data embeddings. As such, it focuses on improving accuracy for embeddings present in the training set and does not attempt to generalize to data embeddings that do not have associated queries in the training set. In contrast, Adaptor and PTFT do have the potential to generalize beyond the labels observed during training and can potentially improve accuracy on labels not seen during training.
>
> However, (1) we believe there are many practical applications where this closed-world setting holds and generalization to new embeddings is not needed, (2) if generalization is needed, NUDGE can be combined with Adaptor or PTFT to combine the benefits of NUDGE on labeled data with the generalization ability of Adaptor or PTFT on unlabeled data, and (3) although Adaptor and PTFT have the ability to generalize to unseen data embeddings, this generalization ability also comes with the risk of performance regression when the model overfits, as seen in our experiments.
>
> (1) *Practicality*. There are many applications where the set of documents to be queried are relatively fixed. Some examples include a corpus of hospital or medical regulations, laws passed by the US Congress, the documentation for software, and textbooks on a specific topic. Journalists, legal experts, software users, and students frequently query such fixed corpora, and improving retrieval accuracy is beneficial in all such settings. Moreover, even if the corpus set changes, NUDGE is useful as long as the query distribution does not change. A journalist studying historical crime rates in a city still benefits from better retrieval of court cases in the past, even if new court cases are added to the corpus.  In fact, NUDGE has already been incorporated into a popular open-source library for RAG applications, showing the interest to include NUDGE in real-world applications for better retrieval.
>
> (2) *Combining NUDGE with PTFT and Adaptors*. We have conducted new experiments to study the impact of combining NUDGE with PTFT and Adaptors. Specifically, we empirically evaluate A+NUDGE-N and P+NUDGE-N methods, which perform NUDGE-N on top of embeddings found by, respectively, Adaptor and PTFT. Overall, the new experimental results in Table 6 (please see the updated Sec. 4.2 for details of this experiment and results) show that A+NUDGE-N and P+NUDGE-N are in fact able to provide the generalization benefits of Adapter and PTFT, while also maintaining significant accuracy boosts on in-distribution samples. This shows that if generalization is needed, NUDGE can be combined with existing approaches to provide the best of both worlds, i.e., improved accuracy on records present in the training set through NUDGE, and improved accuracy on records unseen during training through parametric methods.

---

> ### Author Response · Authors · 2024-11-21
>
> **Response to Q1 Continued...**
>
> (3) *Generalization*. Finally, we have significantly expanded our empirical analysis of generalization in Sec. 4.2, presenting per dataset results and complementing our previous results on OOD query generalization to also include OOD label generalization. Our results show that although PTFT and Adaptor do provide OOD generalization benefits, there is also a risk of overfitting. Specifically, our results in Table 7 show that PTFT and Adaptor improve accuracy on OOD samples for NFCorpus and NQ datasets (better than NUDGE), but also cause a significant drop in OOD accuracy (which ends up being worse than NUDGE) on the SciFact dataset (we note that SciFact is our smallest dataset and generalization from such a small training set of <1000 queries is difficult). NUDGE, on the other hand, avoids such a performance regression, and it, in general, marginally impacts OOD accuracy. Avoiding such performance regressions is a benefit of NUDGE that allows for its safe deployment in practical settings, as it will not lead to unexpected behavior in OOD settings.
>
> To conclude, we believe NUDGE presents an interesting, practical, and surprisingly effective method within the retrieval fine-tuning design landscape. It is complementary to existing approaches, and can be combined with them to complement their strengths and address their weaknesses. Our experimental results provide a thorough analysis of the strengths and weaknesses of NUDGE as well as other existing methods, which can be used as a basis to design better fine-tuning methods in the future.

---

> > ### Author Response · Authors · 2024-11-21
> >
> > **Question Q2**. "For out-of-sample results, it would be interesting to see what would happen if an actual feature transformation model was trained on top of the embeddings found by this method."
> >
> > **Response.** Based on the reviewer’s suggestion, and as discussed in our response to Q1, we have now included in Tables 6 and 7 results that show how NUDGE can be combined with both Adaptor (which learns a linear feature transformation) and PTFT. Please see our response to Q1 for a discussion of these results, which shows that combining NUDGE with Adaptor and PTFT allows for improving accuracy both on in-distribution samples through NUDGE, and on out-of-distribution samples by inheriting the generalization ability of Adaptor and PTFT.
> >
> > Here, we note that the reviewer seems to suggest performing Adaptor after using NUDGE on the embeddings, referred to NUDGE-N+A (instead of A+NUDGE-N which performs NUDGE after transforming embeddings by Adaptor). In fact, our experiment showed that NUDGE-N+A performs identical to NUDGE-N (we have omitted this result for brevity). We suspect this is because NUDGE already changes the embeddings until a point where further modifications worsen accuracy. In contrast, in A+NUDGE-N, there is still room for improvement after applying Adaptor, so performing NUDGE on top of Adaptor is beneficial.
> >
> > **Comment W1**. "The "elephant in the room" for this method is that it does not work on out-of-sample documents. Since the embeddings of the database are adjusted at fine-tuning time, new documents cannot benefit from it, unless new training data is added for them as well, which becomes a really narrow application setting. This is all the more apparent since out-of-distribution on the queries side is considered in this work."
> >
> > **Response**. Thank you for the comment, please see our response to Q1 where we discuss in detail the points raised.
> >
> > **Comment W2**. "The method works only on densely annotated training data, since a document that is never returned to a query will not be adapted. This is apparent in the KILT dataset where documents that do not appear in any result are simply removed."
> >
> > **Response**. Thank you for the comment, please see our response to Q1 where we discuss in detail the points raised, especially with the added experiment evaluating the approach’s accuracy on unlabeled documents, as well as new experiments showing that NUDGE can be combined with parametric methods if generalization is needed in an application.
> >
> > Here, we additionally note that although it is correct that NUDGE does not modify embeddings for which there is no ground-truth in the training set, it is not necessary that all of the corpus be annotated to improve accuracy. This is because, in practice, query labels are often not uniformly distributed, but instead are concentrated on a smaller subset of the dataset. For instance, consider the benchmark datasets NQ, NFCorpus and SciFact that are created from real-world queries. As the newly added Fig. 7 shows, a significant portion (around 40% or more) of queries on these datasets ask for data records accessed by other queries. Improving retrieval accuracy for those embeddings alone is sufficient to increase retrieval accuracy in general, as evidenced by the benefits of NUDGE on these datasets.
> >
> > **Comment W3**. "Because of this particular setting, the queries in experiments must be split in train+val+test, which is unusual. Therefore there are no comparisons with state-of-the-art methods."
> >
> > **Response**. We are unsure what state-of-the-art methods the reviewer is referring to, but we would be happy to compare with them given the chance. Our method is a fine-tuning approach for which train+val+test split is standard (where validation set is used for hyperparameter tuning). For example, many of the BEIR and KILT datasets already come with a train+val+test split. In case the reviewer is referring to recent query rewriting methods, we note that NUDGE is indeed orthogonal to such methods, and can be used together with them. We have added a new experiment showing how NUDGE can be combined with HyDE, a popular query rewriting technique that rewrites the queries in terms of hypothetical documents that answer the queries. The new experiment is presented in Appendix E.8, and shows that combining NUDGE with HyDE performs better than using NUDGE alone (which still performs significantly better than HyDE alone).

---

> > > ### Comment · Reviewer_ZHaK · 2024-11-26
> > >
> > > Reviewer K1aL had the same reservation as me about the fact that NUDGE does not work on out-of-sample data.
> > >
> > > In their copious rebuttals and subsequent additions to the paper, the authors addressed this (1) by acknowledging this is a limitation and (2) by adding new experiments that address the limitation. Therefore, my opinion on the paper improved.
> > >
> > > I think the paper could be accepted.

---

### Official Review · Reviewer_ubhs · 2024-11-04

**Soundness:** 2
**Presentation:** 3
**Contribution:** 2
**Rating:** 5
**Confidence:** 3

**Summary:**

This paper introduces NUDGE, a non-parametric embedding fine-tuning approach designed to enhance k-nearest neighbor (k-NN) retrieval accuracy in text and image retrieval tasks. Two main variants, NUDGE-M and NUDGE-N, apply controlled modifications to embeddings, with NUDGE-M using bounded changes and NUDGE-N adding normalization to improve out-of-distribution generalization. Experimental results across multiple datasets show that NUDGE consistently outperforms traditional PTFT and Adapter methods.

**Strengths:**

1. **Intuition of the Proposed Method**: The paper clearly presents NUDGE's core idea of directly modifying embeddings as parameters, offering a simple, intuitive approach to fine-tuning that improves retrieval accuracy without altering the model.

2. **Extensive Experimental Evaluation**: Experiments are comprehensive, covering diverse retrieval tasks (text, cross-modal) and configurations, validating NUDGE's effectiveness, efficiency, and adaptability across different datasets and models.

**Weaknesses:**

1. **Theoretical Issues**:
   - Equation (1), defining MaxA-EFT, is disconnected from the actual task as it primarily maximizes inner product similarity without a clear link to retrieval-specific objectives.
   - The NP-hard nature of MaxA-EFT, while noted, lacks clear relevance to the search task, and the authors fail to define an explicit optimization problem for solving it.
   - Equation (2) being unbounded is more of a general mathematical property and seems unrelated to the paper’s focus on non-parametric methods.
   - Line 212’s constraint on \(\Delta\) uses matrix norm notation, which doesn’t adequately capture the constraints on embedding norms, indicating a need for clearer notation.

2. **Experimental Issues**:
   - Table 3’s results are overly generalized under “across text datasets,” obscuring which datasets contributed to the reported metrics.
   - In BEIR benchmark results (Table 5 and Table 9), the first column shows identical performance across all methods, which is counterintuitive and suggests either data or methodology issues.
   - The GPU speedup in Table 6 is marginal compared to the CPU, which raises efficiency concerns, as an A100 GPU should provide significantly higher speedup for such operations relative to CPU processing.

**Questions:**

As the modifications of the embeddings still require learning and would be sufficiently large given a large dataset, how much benefit does this method still have when the dataset is large?

---

> ### Author Response · Authors · 2024-11-21
>
> We thank the reviewer for their comments. We believe most of the reviewer's comments stem from a misunderstanding of the paper. We provide a discussion below that we hope to clarify the confusion.
>
> **Comment W1**. "Equation (1), defining MaxA-EFT, is disconnected from the actual task as it primarily maximizes inner product similarity without a clear link to retrieval-specific objectives."
>
> **Response**. We believe the reviewer has misunderstood our problem setting. Inner product similarity *is* the retrieval-specific objective. Specifically, our work follows a very common retrieval paradigm, where the retrieval method retrieves data records whose embeddings have the maximum inner product similarity with the query embedding. As such, MaxA-EFT, is the problem of fine-tuning data embeddings to maximize the retrieval accuracy. In other words,  MaxA-EFT *is a formalization of the actual retrieval task*.
>
> **Comment W2.** "The NP-hard nature of MaxA-EFT, while noted, lacks clear relevance to the search task, and the authors fail to define an explicit optimization problem for solving it."
>
> **Response**. We are confused by this comment from the reviewer. MaxA-EFT is the problem of fine-tuning data embeddings to maximize the retrieval accuracy, which is exactly the search task studied in the paper. Moreover, we are unclear about what the reviewer means by “the authors fail to define an explicit optimization problem for solving it”. First, MaxA-EFT *is an optimization problem*. Second, It is NP-hard, which means it cannot be solved unless P=NP. We are unsure what defining an explicit optimization problem for solving an NP-hard problem means. Moreover, we do, in fact, solve a variant of this NP-Hard problem that is tractable, via an explicit optimization problem; this is the whole point of the paper.
>
> **Comment W3**. "Equation (2) being unbounded is more of a general mathematical property and seems unrelated to the paper’s focus on non-parametric methods."
>
> **Response**. On the contrary, Equation (2) being unbounded has *direct impact on our methods*, because it means there is no optimal solution to the problem. Eq. (2) is the problem of fine-tuning data embeddings while maximizing similarity between data and training query embeddings. This is exactly the paper’s focus, i.e., solving a non-parametric optimization problem to improve retrieval, and unboundedness poses a challenge as it does not allow us to solve the optimization problem as stated.
>
> **Comment W4**. "Line 212’s constraint on ($\mathbf{\Delta}$) uses matrix norm notation, which doesn’t adequately capture the constraints on embedding norms, indicating a need for clearer notation."
>
> **Response**. Line 212 contains a constraint on $\mathbf{\Delta}_i$, which, as defined in Sec. 2 is the $i$-th row of the matrix $\mathbf{\Delta}$, and therefore is a vector. L2 norm of $\mathbf{\Delta}_i$ is the L2 norm of a vector, and clearly well-defined.

---

> > ### Author Response · Authors · 2024-11-21
> >
> > **Comment W5**. "Table 3’s results are overly generalized under “across text datasets,” obscuring which datasets contributed to the reported metrics."
> >
> > **Response**. The paper already contains the detailed per dataset results for all the datasets in Table 3; specifically, Table 5 shows per dataset results for BGE-S, Table 9 shows per dataset results for GTE-L and Table 10 shows per dataset result for TE3-L. In fact, the reviewer already refers to such per dataset results in their comment below, indicating that per dataset results are clearly presented in the paper.
> >
> > **Comment W6**. "In BEIR benchmark results (Table 5 and Table 9), the first column shows identical performance across all methods, which is counterintuitive and suggests either data or methodology issues."
> >
> > **Response**. First, we note that due to the use of a validation set, a method that performs worse than No Fine-Tuning on a dataset will default to No Fine-Tuning. For example, as Fig. 2 shows, for ArguAna, NUDGE-M does worse than No Fine-Tuning for all values of $\gamma>0$. Thus, the approach sets $\gamma=0$, resulting in a method identical to No Fine-Tuning. This means that different approaches do in fact perform differently, but since we use a validation set, a performance worse than No Fine-Tuning will not be reported for any approach. This is a fair comparison, as in practice, users can default to No Fine-Tuning, if a fine-tuning method does not improve accuracy.
> >
> > Consequently, the results in Table 5 and Table 9  show that no approach among NUDGE-M, NUDGE-N, PTFT and Adaptor improves over No Fine-Tuning. Interestingly, based on the Reviewer j9KJ’s suggestion, we have included the method NUDGE-S (see Appendix C.1 for details of the approach and Appendix E.6 for experiments), which performs sparse data updates for fine-tuning, and indeed performs better than No Fine-Tuning on ArguAna. This suggests that sparse updates are more suitable for that dataset, while NUDGE-M, NUDGE-N, PTFT and Adaptor perform dense updates. Overall, we have included ArguAna as an example of a dataset where no standard approach performs well, presenting a diverse set of datasets that thoroughly evaluate different approaches.
> >
> > **Comment W7**. "The GPU speedup in Table 6 is marginal compared to the CPU, which raises efficiency concerns, as an A100 GPU should provide significantly higher speedup for such operations relative to CPU processing."
> >
> > **Response**. First, we emphasize that the goal of Table 6 (Table 8 in the revised paper) is to show that NUDGE runs within a few minutes on both GPU and CPU, while Adaptor, although fast on GPU, takes more than an hour on CPU, and PTFT takes hours on GPU. The focus of our paper has not been on providing a GPU-optimized implementation of any of the algorithms, and the runtime analysis is for a standard implementation of all algorithms. For PTFT and Adaptors, the implementation closely follows that of the SentenceTransformers and LlamaIndex libraries, respectively.
> >
> > Having said that, we disagree with the reviewer’s suggestion that just because an A100 GPU has significantly more compute capacity, it can provide significantly higher speed-ups. In fact, the speed-ups the GPU can provide does not depend only on its capacity, but on how much of the capacity can be used to perform the operations (which depends on how much parallelism is possible for the specific operations). For the experiments in the paper where runtime is presented, the embedding dimension is 384, so most operations involve matrices where each row is 384 dimensional. A dimensionality of 384 is much smaller than the maximum capacity of A100 GPUs and processing such matrices does not utilize the full capacity of the GPU. This leads to lower speed-ups than using A100 GPUs on larger matrices.  To show this, we performed a micro-benchmark to study the effect of matrix dimension on runtime difference between GPU and CPU. The newly added Fig. 6 in Appendix E.10 in the paper shows the runtime of GPU and CPU when performing 13 consecutive matrix addition and multiplication operations on matrices of dimensions $d\times d$, varying $d$. We see that at $d=384$, GPU is 2.3x faster, but at $d=3072$ the GPU is 1600x faster. To summarize, for lightweight methods such as NUDGE and Adaptor which involve small matrix operations, speed-ups of using an A100 GPU are not proportional to its capacity.

---

> > > ### Author Response · Authors · 2024-11-21
> > >
> > > **Question Q1**. "As the modifications of the embeddings still require learning and would be sufficiently large given a large dataset, how much benefit does this method still have when the dataset is large?"
> > >
> > > **Response.** We note that various datasets used in our experiments have millions of records, showing the benefits of NUDGE on datasets of that scale. In fact, the larger the dataset, we expect more benefits to NUDGE’s lightweight approach because it can be applied efficiently, while other approaches like PTFT might take days to run, and require GPUs with large memory capacity which renders them infeasible for many users. NUDGE runs in a single pass over the entire dataset, and without the need for any model forward pass. As such, it scales much better with data size compared with PTFT and Adaptor which typically need 100s of epochs over the training set to converge.
> > >
> > > Moreover, because the approach is non-parametric, it can easily scale to larger datasets without the need for additional modeling considerations. For example, Adaptor’s training strategy and modeling choices may need to change to allow for better training as data size increases. NUDGE’s non-parametric approach avoids the need for such hyper-parameter tuning altogether.
> > >
> > > Finally, NUDGE improves accuracy much more than other approaches, and our experiments show that the benefits of NUDGE do not depend on the size of the dataset, providing consistent accuracy boost across datasets with thousands to millions of records. Overall, considering the efficiency and accuracy gains achieved by NUDGE, NUDGE is expected to remain the best choice for fine-tuning, especially when the datasets are large.

---

> > ### Comment · Reviewer_ubhs · 2024-11-26
> >
> > Thanks for the response,
> >
> > For W1 and W2, I am questioning the **necessity of NP-hard modeling**.
> >
> > Inner product similarity-based search (MIPS) does not inherently require NP-hardness, as gradient-based approaches are sufficient to optimize for inner product similarity. Could you clarify why NP-hard modeling is necessary for addressing MaxA-EFT?
> >
> > For W4, I am asking clarification of the notation. If $\Delta_i$ represents the $i$-th row of $\Delta$, what does $\Delta_{Y_i}$ in Equations (2) and (32) refer to, given that $Y$ is the set of ground-truth labels? The authors could have resolved this ambiguity with just a few wording adjustments, but they chose not to do so.

---

> > > ### Author Response · Authors · 2024-11-26
> > >
> > > Thank you for your clarifications. Please see our response below. We have updated the paper to include your requested clarification on the notation. The modifications in the paper are marked in blue.
> > >
> > > **Comment** “Inner product similarity-based search (MIPS) does not inherently require NP-hardness, as gradient-based approaches are sufficient to optimize for inner product similarity. Could you clarify why NP-hard modeling is necessary for addressing MaxA-EFT?”
> > >
> > > **Response.** First, we note that this paper is the first work studying *non-parametric* optimization of embeddings for MIPS-based search. As the reviewer points out, a large body of work has previously focused on gradient-bassed *parametric* optimization approaches. Given our novel *non-parametric* formulation, the optimization problem is new and not previously explored in the literature. It is not necessarily true, as we discuss below, that observations from *parametric* optimization approaches (e.g., suitability of gradient-based approaches) hold in the *non-parametric* setting. One of our contributions is to provide a thorough study of the non-parametric optimization problem to understand how it should be solved.
> > >
> > > As such, the first important question addressed in the non-parametric formulation is whether the embeddings can be fine-tuned *optimally* to maximize accuracy. Our NP-Hardness result shows that it is not possible to do so. This result is consequential in how we solve the problem; if the problem wasn’t NP-hard, an optimal solution should’ve been provided. The NP-hardness is important as it informs our approach to solving the optimization problem.
> > >
> > > Moreover, regarding gradient-based approaches, we, in fact, do study their applicability to our non-parametric optimization problem, where we provide a theoretical discussion in Appendix C.2 and evaluate them empirically in Appendix E.5. Our result shows that a gradient-based approach is theoretically equivalent to NUDGE-M (see Lemma 5), and solves a constrained variation of the optimization problem, *not MaxA-EFT itself*. Furthermore, the gradient-based approach is slower than NUDGE-M, because NUDGE-M provides a closed-form solution, while the gradient-based approach is iterative (this observation is also confirmed empirically in Appendix E.5). Thus, for the non-parametric optimization problem, gradient-based approaches are less suitable, even though they are ubiquitous in the parametric setting. This emphasizes the difference between parametric and non-parametric optimization formulations and the need for separate theoretical analysis in the non-parametric setting.
> > >
> > > **Comment** “If $\mathbf{\Delta_i}$ represents the $i$-th row of $\mathbf{\Delta}$, what does $\mathbf{\Delta_{Y_i}}$ in Equations (2) and (32) refer to, given that $Y$ is the set of ground-truth labels? “
> > >
> > > **Response.** Thank you for the clarification on your comment. Recall that $Y_i$ is the index of the ground-truth answer to the $i$-th query, so that $\mathbf{D_{Y_i}}$ and $\mathbf{\Delta_{Y_i}}$ are $d$-dimensional vectors, respectively, referring to the data embedding of the ground-truth answer to $i$-th query, and the modification to be learned to the data embedding of the ground-truth answer to $i$-th query. That is, after fine-tuning, the embedding for the ground-truth answer to the $i$-th query is  $\mathbf{D_{Y_i}}+\mathbf{\Delta_{Y_i}}$. We have now added this clarification to the beginning of Sec. 3.1.

---

> ### Comment · Reviewer_ubhs · 2024-11-26
>
> Thank you for reiterating the novelty of your work, which I agree with. However, my question is specifically about **why NP-hardness is necessary for addressing MaxA-EFT**.
>
> Appendices C.2, E.5, and Lemma 5 discuss how the optimal solution can be equivalent to the iterative method. **These claims seem to support my concern** that the iterative method provides a simple solution that is as good as the optimal one.
>
> The authors should also avoid using unnecessarily complex theoretical constructs and uncommon notation. For example, the definition of $ Y_i $ differs from its commonly used meaning, which is inconsistent with that $ Y $ is defined as the label set.

---

> > ### Author Response · Authors · 2024-11-26
> >
> > **Comment** "Thank you for reiterating the novelty …  that is as good as the optimal one."
> >
> > **Response**. The reviewer’s comment conflates the multiple optimization problems studied in the paper. Specifically, in the paper, we study (1) MaxA-EFT optimization problem, which is the *unconstrained* optimization problem of fine-tuning embeddings while maximizing accuracy and (2) the *constrained* variations of the optimization problem, BiMax-M and BiMax-N.
> >
> > (1) MaxA-EFT is the **unconstrained** optimization problem of fine-tuninig embeddings to maximize retrieval accuracy. It is a natural formulation of the problem of fine-tuning embeddings to maximize accuracy. Given that fine-tuning embeddings to maximize retrieval accuracy is the focus of the paper, MaxA-EFT is a problem that should be solved in the paper. However, because it is NP-hard, it cannot be solved optimally. Thus, instead of solving MaxA-EFT directly, we solve tractable variations of the problem. In other words, *because MaxA-EFT is NP-hard, we don’t solve it directly but instead formulate tractable variants of the problem.* Clearly, NP-hardness is necessary to motivate this discussion in the paper.
> >
> > (2) The paper specifically formulates two tractable variants of the optimization problem BiMax-M and BiMax-N, both of which are **constrained** optimization problems. We provide a thorough theoretical and empirical analysis of how these problems can be solved, presenting both closed-form and iterative gradient-based solutions. As the reviewer points out and already discussed in Appendix C.2, the gradient-based solutions can be used to solve BiMax-M and BiMax-N. Appendix C.2 specifically presents NUDGE-IM and NUDGE-IN, two variants that use iterative gradient-based optimization, and we theoretically show that NUDGE-IM is equivalent to NUDGE-M under the correct hyper-parameter setting (although suboptimal hyper-parameter setting leads to suboptimally). Nonetheless, iterative variants are less efficient compared with closed-form solutions, since their iterative nature means they need to go over the training/validation set multiple times. Comparing iterative variants with closed-form solutions, Appendix C2 already states, “These approaches can be less efficient and suboptimal, but can still provide accurate solutions, and are simple and flexible.” Overall, we have provided a thorough study of how the constrained optimization problems can be solved, and we leave it to the practitioners to decide if simplicity of solution is preferred in practice, or optimality and efficiency.
> >
> > **Comment** "The authors should also avoid …  is defined as the label set."
> >
> > **Response.** The reviewer seems to have misunderstood the notation. In our notation, $Y$ is the *label set*, and $Y_i$ is the $i$-th element of the label set. In a retrieval setting, the ground-truth answer (i.e., the label) for a query is the index of the data record that matches the query.
> >
> > In case the reviewer is implying that using $Y$ as a label set is uncommon, we note that the contrary is true, and using $Y$ to refer to the label set is very common in machine learning, please see the Deep Learning book by Goodfellow et al.

---

> ### Comment · Reviewer_ubhs · 2024-11-28
>
> This response does not adequately address **why NP-hardness is necessary**. Using similar reasoning, any classification or distance-based retrieval problem that aims to maximize validation accuracy could also be proven to be NP-hard. But very few of such papers start the paper with an unnecessary, almost irrelevant theorem of NP-hardness.
>
> Frankly, the core idea of the paper is simple. However, including such unnecessary complexities distracts readers and diminishes the potential impact of the paper.

---

> > ### Author Response · Authors · 2024-11-29
> >
> > First, if by “necessary” the reviewer means *the experimental results presented in the paper do not depend on NP-hardness of MaxA-EFT*, then we agree with the reviewer that NP-hardness is not “necessary”. If a reader is only interested in our proposed algorithms and their practical utility, then they can safely ignore our NP-hardness result and only focus on other relevant sections. We believe our experimental results are impressive as they are and our methodology is novel and interesting; so the paper already makes significant contributions to improve retrieval accuracy through fine-tuning for practical applications.
> >
> > However, the reviewer seems to question the need for NP-hardness altogether and suggests studying NP-hardness is not relevant to machine learning problems. We note that the contrary is true, and studying NP-hardness in various settings is, and has been for decades, an active area of research (e.g., Brand et al 2024 NeurIPS, Froese & Hertrich 2023 NuerIPS, Shamir 2018 JMLR and Blum & Rivest 1988 NeurIPS below and the references therein). In fact, training neural networks, in general, is known to be NP-hard (Blum & Rivest 1988), meaning that when using *parametric* neural network-based methods to train classification or retrieval models, *NP-hardness is already known*. This explains why no new NP-hardness result is presented in papers when using parametric methods.
> >
> > Moreover, we also disagree with the reviewer's suggestion that just because one problem that maximizes validation accuracy is NP-hard, other related problems also become NP-hard. This is already apparent from the paper, since all three of MaxA-EFT, BiMax-M and BiMax-N maximize validation accuracy, but the latter two are not NP-hard and can be solved optimally in polynomial time. Consequently, NP-hardness must be studied specifically for the problem at hand, and unfounded extrapolations can lead to incorrect conclusions.
> >
> > Overall, MaxA-EFT is a novel problem (thus unknown if it is NP-hard) and solving it is the central focus of this paper. We believe the NP-Hardness result is relevant and useful because it allows us to know that an exact solution to MaxA-EFT cannot be obtained and justifies the use of our alternative problem formulations to solve the fine-tuning problem. Moreover, the NP-Hardness result also contributes to our general understanding of the non-parametric fine-tuning problem (e.g., future work now knows the solution cannot be improved by solving MaxA-EFT optimally).
> > Nonetheless, if a reader is only interested in practical utility and not NP-Hardness of the problem, they are free to ignore that portion of the paper.
> >
> >
> > Brand, Cornelius, Robert Ganian, and Mathis Rocton. "New complexity-theoretic frontiers of tractability for neural network training." Advances in Neural Information Processing Systems 36 (2024).
> >
> > Froese, Vincent, and Christoph Hertrich. "Training neural networks is np-hard in fixed dimension." Advances in Neural Information Processing Systems 36 (2024).
> >
> > Shamir, Ohad. "Distribution-specific hardness of learning neural networks." Journal of Machine Learning Research 19.32 (2018): 1-29.
> >
> > Blum, Avrim, and Ronald Rivest. "Training a 3-node neural network is NP-complete." Advances in neural information processing systems 1 (1988).

---

> > > ### Comment · Reviewer_ubhs · 2024-12-03
> > >
> > > Making simple concepts unnecessarily complex cannot be considered a meaningful contribution. Moreover, beginning the paper with such a statement distracts readers from the actual contributions of the work.
> > >
> > > Additionally, the authors have not adequately addressed concerns regarding scalability. The claim that "the larger the dataset, the more benefits we expect from NUDGE’s lightweight approach" is not true, as the size of adapaters are fixed while the proposed method has to update all embeddings.
> > >
> > > Taking into account the strengths and weaknesses of the paper, I will maintain my current rating.

---

> > > > ### Author Response · Authors · 2024-12-04
> > > >
> > > > We believe **the reviewer has not identified a single meaningful weakness in our paper**, and as we have pointed out throughout the discussion, their comments are either illogical or obviously false, stemming from obvious misunderstandings of the paper and the related work. Here, we discuss the two concerns raised by the reviewer in their final comment.
> > > >
> > > > The reviewer finds our NP-hardness result (occupying three lines in the paper) to make the paper “unnecessarily complex” and “distracting”. As we have explained throughout the discussion, the NP-hardness result is relevant and useful (it is a result on whether the central optimization problem in the paper can be solved optimally), and NP-hardness results are common in machine learning literature. We find it bewildering that devoting fewer than 3 lines of the paper on NP-hardness, a concept commonly taught in undergraduate computer science degrees, makes the paper “unneccessarily complex” and “distracting”. We therefore believe the **reviewer's comment is illogical and we disagree with their assertion that the 3 lines of the paper on such a well-known concept makes the paper “unnecessarily complex” or is “distracting”. While we could have removed this 3 line hardness result to please the reviewer, we find that doing so would both diminish the quality of the paper and its contributions, and would, in our opinion, be silly — since any reader who doesn’t care about hardness can safely ignore these lines.**
> > > >
> > > > Moreover, the **reviewer's comment on scalability is obviously false** and suggests a complete misunderstanding of our paper and the related work. The reviewer suggests that since “the size of adapters are fixed while the proposed method has to update all embeddings”, adaptors are more scalable than our proposed method. This claim is false for *many* different reasons: (1) *any approach* that fine-tunes the data embeddings has to modify the data embeddings, this obviously has nothing to do with whether the size of the model used for fine-tuning is fixed or not, (2) training time depends on the number of training samples and how many times a method iterates over them: adaptors iterate multiple times over the training set but NUDGE does it only once, making it obviously faster, (3) NUDGE, unlike what the reviewer suggests, does not modify all data embeddings, but only changes data embeddings for which a training sample is available, and (4) a fixed-size adapter is unlikely to be beneficial as data size increases, because a more complex transformation (i.e., with more parameters) is likely needed to make fine-grained adjustments for a larger dataset. In fact, regarding scalability, our paper *theoretically proves* that NUDGE is faster than adaptors for any data size (a claim supported by our experiments in Table 8, that shows NUDGE is around  3 and 11 times faster than Adaptors, respectively when using GPUs and CPUs), where we compare the time complexities of different methods in Theorem 2 and the discussion directly below it.

---

### Official Review · Reviewer_j9KJ · 2024-11-04

**Soundness:** 3
**Presentation:** 3
**Contribution:** 3
**Rating:** 5
**Confidence:** 3

**Summary:**

This work presented a family of novel non-parametric embedding fine-tuning approaches that are significantly more accurate and efficient than both sets of existing approaches.

**Strengths:**

1.The s NUDGE, a non-parametric embedding fine-tuning method, which innovatively addresses the limitations of parametric approaches in k-Nearest Neighbor (k-NN) retrieval. This originality stems from shifting focus to the embeddings themselves rather than model parameters, offering a novel optimization perspective for retrieval tasks.

2.Comprehensive experimental and theoretical analyses validate NUDGE's efficacy across five pre-trained models and nine retrieval datasets.

3.The paper is well-organized and presents its methodology and experimental findings clearly.

4.By providing a lightweight, model-agnostic solution with strong generalization to out-of-distribution queries, NUDGE offers substantial improvements in both computational efficiency and accuracy.

**Weaknesses:**

1. It does not extensively explore or compare alternative constraint types. For example, additional regularization techniques, such as sparsity constraints or adaptive thresholds.

2. Lack a detailed ablation study that examines how skewed distributions impact both in-distribution and out-of-distribution performance could reveal if other normalization techniques or dataset-specific adjustments would better manage this overfitting.

3. While the optimization and constraint methods are described, some of the choices behind the approach could use further explanation.

**Questions:**

1. What is the sensitivity of NUDGE's performance to hyperparameter selection, particularly the choice of γ. If 'NUDGEM sets γ by solving MaxA-EFT on the validation set', how should γ be set on the training set?

2. In Eq.1, the range of j is confusing, what does '\Y_j' represent?

3. Why do NUDGE-M and NUDGE-N methods have similar average results on image retrieval pre-trained models. （CLIP-B and CLIP-L）

4. The criteria for selecting the constraint boundaries in NUDGE-M and the specific reasoning behind the grid search implementation for NUDGE-N's gamma optimization are not fully detailed.

5. How to mitigate the impact of unreliable training query embeddings when fine-tuning based on these embeddings?

---

> ### Author Response · Authors · 2024-11-21
>
> We thank the reviewer for their comments. We have revised the paper to address all the reviewer's comments, as discussed below. Modifications are marked in blue in the revised paper.
>
> **Comment W1**. "It does not extensively explore or compare alternative constraint types. For example, additional regularization techniques, such as sparsity constraints or adaptive thresholds."
>
> **Response.** Thank you for the suggestion on alternative constraint types. We have extended the paper to include the alternative constraint types, discussing both sparsity constraints and adaptive thresholds as suggested by the reviewer. We have included both a theoretical discussion (in the newly added Appendix C.1) and new empirical results (in the newly added Appendix E.6) using the alternate constraint types, as well as a discussion in Sec. 3.3. Overall, we observed limited benefits in using such constraints compared with NUDGE-N and NUDGE-M. We provide an overview of the added discussion below.
>
>
> **Summary of added discussion in Sec 3.3, Appendix C.1 and Appendix E.6.** We agree with the reviewer that constraints are an important part of NUDGE, controlling how embeddings are updated. The L2 norm constraints in NUDGE-M and NUDGE-N are chosen since we focus on dense embeddings and bounding L2 norm leads to dense embedding updates. However, other constraints are possible, for example, using L1 norm instead of L2 to create sparse updates or using adaptive thresholds in NUDGE-M and NUDGE-N to increase flexibility in fine-tuning. We studied both such approaches, deriving the fine-tuning update rule using L1 norm in Appendix C.1 and empirically evaluating both the use of L1 norm and adaptive thresholds in Appendix E.6. We observed limited benefits from these alternative constraint formulations. We describe each case in more detail next.
>
> *Sparsity*.  Using L1 norm instead of L2 norm leads to sparse embedding updates. In Appendix C.1, we formalize the optimization problem when replacing L2 with L1 in NUDGE-M. We theoretically derive the corresponding embedding update rule, which as Lemma 4 shows, is sparse. We refer to the corresponding sparse embedding fine-tuning method as NUDGE-S. We empirically evaluate NUDGE-S in Appendix E.6. NUDGE-S performs poorly in practice, providing worse accuracy on all datasets except ArguAna. This is likely due to the dense embedding space in our experiments, where sparse updates ignore dependencies across embedding dimensions during fine-tuning. Interestingly, NUDGE-S outperforms other NUDGE variants on ArguAna, which is the only dataset on which NUDGE-M and NUDGE-N do not improve accuracy over No Fine-Tuning. This suggests that although L2 norm constraints perform better in general and across datasets, alternative constraints can also be beneficial in some cases to improve accuracy.
>
> *Adaptive thresholds*. We assume that by adaptive threshold the reviewer refers to adaptively changing the $\gamma$ threshold in different parts of the embedding space. We empirically explore this approach in Appendix E.6.2, where we cluster data embeddings into $r$ clusters, and perform NUDGE-M and NUDGE-N independently on each cluster. Such an approach allows different parts of the space to set different $\gamma_1$,.., $\gamma_r$ thresholds for its embedding updates. The results are presented in Appendix E.6.2, where we see a marginal change in accuracy in NUDGE-N, and a significant drop in accuracy in NUDGE-M. Overall, NUDGE-N, as also discussed in response to the reviewer's Q1 below, is not sensitive to $\gamma$, and as a result, setting it adaptively makes little difference. On the other hand, NUDGE-M is sensitive to $\gamma$. We suspect the drop in accuracy is because fine-tuning is done independently across clusters, where we expect to see  dependencies between $\gamma$ values across clusters. However, an approach that does consider dependencies across clusters is non-trivial. We leave such a study to future work and emphasize that NUDGE-M already provides significant accuracy improvements over existing work.

---

> > ### Author Response · Authors · 2024-11-21
> >
> > **Comment W2**. Lack a detailed ablation study that examines how skewed distributions impact both in-distribution and out-of-distribution performance could reveal if other normalization techniques or dataset-specific adjustments would better manage this overfitting.
> >
> > **Response**. We assume that by skewed distribution the reviewer is referring to the skewness in the distribution of data labels. Indeed, our experiments already consider various label distributions with different skewness. We have now added Fig. 7 which depicts the label distribution in the datasets used in our experiments, as well as a discussion regarding the impact of skewness in label distribution on our results in Sec. 4.1 with details in Appendix E.12. Our newly added discussion reveals an interesting relationship between the impact of normalization and skewness in label distribution. We also present the discussion below.
> >
> > **Summary of added discussion in  Sec 4.1 and Appendix E.12**. We consider the relationship between skewness in label distribution and normalization to understand the difference in accuracy between NUDGE-M and NUDGE-N and the need for normalization. Comparing accuracy results in Table 4-5 and label distribution in Fig. 7, we see that the impact of normalization depends on whether label distribution is heavy-tailed (leading to many test labels not being present in the training set) or if it is skewed towards a fixed set of labels (in which test labels are likely seen during training). To understand why, first, note that if a data record is not seen during training (i.e., it's not a ground truth to any training query), its embedding will not be modified by NUDGE-M. Meanwhile, NUDGE-M will likely increase the magnitude of the data embeddings that are represented in the training set. As a result, the data embeddings seen during training will have a larger magnitude than data embeddings not seen during training. Smaller magnitude causes the inner product similarity of such unseen data embeddings with test queries to be generally lower, which is detrimental if many test queries ask for data embeddings whose ground truth was not seen during training. Thus, heavy-tailed distributions benefit from normalization, while distributions skewed towards a fixed set of labels are less impacted by it.
> >
> > Using the above argument, the label distribution can provide an explanation for the gap between NUDGE-M and NUDGE-N in SciFact, TriviaQA, NQ, and HotpotQA, where, as Fig. 7 shows, the ground truth for between 40-60% of test queries was unseen during training. It can also provide an explanation for why NUDGE-M performs similarly to NUDGE-N on CoCo, Flicker, and Fever datasets, where, as shown in Fig. 7, almost no test query asks for data records not seen during training. Moreover, our Out-of-Distribution experimental results (Sec. 4.2) can also be explained using the label distribution, where out-of-distribution test queries ask for labels unseen during training, leading to poor accuracy for unnormalized embeddings. We also see that NUDGE-M performs worse than NUDGE-N in that setting. Overall, this suggests that NUDGE-M  (with unnormalized embeddings) has poor generalization outside of the training set, and normalization, as done in NUDGE-N, helps solve this problem.

---

> > > ### Author Response · Authors · 2024-11-21
> > >
> > > **Comment W3**. "While the optimization and constraint methods are described, some of the choices behind the approach could use further explanation."
> > >
> > > **Response**. We have included a discussion as requested by the reviewer in their Q1 and Q4, please see our response below.
> > >
> > > **Question Q1.** “What is the sensitivity of NUDGE's performance to hyperparameter selection, particularly the choice of γ”.
> > >
> > > **Response**. We have now added Fig. 2 and Sec. 4.3  to the paper to discuss how sensitive NUDGE is to the choice of $\gamma$, where we see that NUDGE-N is not very sensitive to $\gamma$ while NUDGE-M is more sensitive. Interestingly, the best value of $\gamma$ for NUDGE-N is similar across all datasets, suggesting that $\gamma$ depends on an inherent characteristic of the embedding space independent of the dataset.
> > >
> > > On the other hand (as we now discuss in detail in Appendix E.12), the best choice of $\gamma$ for NUDGE-M is dataset-dependent, where we see a relationship between $\gamma$ and label distribution (see also our response to W2). For the label distributions concentrated on a fixed set of labels (i.e., CoCo, Flickr, Fever, NF-Corpus where ground-truth for test queries was seen during training), the larger values of $\gamma$ do not hurt accuracy, since, as discussed in our response to W1, larger magnitudes of fine-tuned embeddings in those datasets is not detrimental. Nonetheless, for other datasets, increased $\gamma$ can worsen accuracy for test queries with ground truths unseen during training. This leads to a trade-off, where increasing $\gamma$ can improve accuracy for queries on data records seen during training and worsen accuracy for queries on records unseen during training. We see that this trade-off leads to different suitable $\gamma$ values for datasets SciFact, HotpotQA, NQ and TrivaiaQA.
> > >
> > > **Question Q1 Continued**. "If 'NUDGEM sets γ by solving MaxA-EFT on the validation set', how should γ be set on the training set?"
> > >
> > > **Response**. We would like to clarify that  NUDGE-M sets $\gamma$ during training to the value that maximizes validation accuracy (this is what we mean when we say 'NUDGE-M sets $\gamma$ by solving MaxA-EFT on the validation set'). This is a typical hyperparameter tuning procedure where a hyperparameter value that achieves the highest validation accuracy is chosen to train the model.
> > >
> > > As such, setting $\gamma$ in NUDGE-M is a typical hyperparameter search problem. To illustrate this further, we note that a simple way to set $\gamma$ (which is not what NUDGE-M does) is through grid search. Grid search considers a set, S, of different $\gamma$ values, applies fine-tuning (i.e., Eq. 3) for each $\gamma\in S$ and checks the resulting validation accuracy, and chooses the $\gamma$ value in $S$ with the highest validation accuracy. Such a grid search is a typical hyperparameter selection approach, and is applicable to NUDGE-M. However, it is inefficient, as it requires fine-tuning and checking validation accuracy for various $\gamma$ values. It is also not guaranteed to find the best possible $\gamma$, since it only checks $\gamma$ values in $S$ (but the optimal $\gamma$ may have been a value not in $S$). Instead of grid search, NUDGE-M provides an algorithm that finds the optimal $\gamma$,  and does it more efficiently than grid search. That is, NUDGE-M finds the best $\gamma$ value without calculating the validation accuracy for every possible $\gamma$. The exact algorithm used is overviewed in Sec. 3.2.1 with details provided in Appendix B.2.
> > >
> > > **Question Q2**. "In Eq.1, the range of j is confusing, what does '\Y_j' represent?"
> > >
> > > **Response**. First, we clarify that (as defined in Sec. 2), $[n]$ refers to the set ${1, 2, 3, …, n}$, so that $j\in[n]$ means $j$ is an integer up to $n$, and $\setminus$ refers to set difference. In Eq. 1, we have $j\in[n]\setminus Y_i$, where $Y_i$ is the ground-truth for the i-th query. Thus,$[n]\setminus Y_i$ is the set of data indexes that are *not* relevant to the $i$-th query. That is, $j\in [n]\setminus Y_i$ sets $j$ as data indices that are the incorrect answers to $Q_i$. We have added this clarification to the paper and also included parentheses in the range of $j$ to avoid any confusion.

---

> > > > ### Author Response · Authors · 2024-11-21
> > > >
> > > > **Question Q3**. "Why do NUDGE-M and NUDGE-N methods have similar average results on image retrieval pre-trained models （CLIP-B and CLIP-L）"
> > > >
> > > > **Response**. This can be explained by the label distribution for the image datasets, as also discussed in our response to W2. In both image datasets CoCo and Flickr, all images have exactly 5 different queries. Thus, the ground truth for all test queries is also present in the training set. This means NUDGE-M modifies the embedding of all data records during fine-tuning, leading to an increase in the magnitude of all data embeddings. Interestingly, in both CoCo and Flickr, our results show that the increase in the magnitude has low variation across embeddings. In fact, the standard deviation of the magnitude of the embeddings after fine-tuning is 0.02 (average embedding magnitude is 2.24 and 1.81 respectively for CoCo and Flickr datasets). Since there is a low variation in the magnitudes of the embeddings, normalization does not make a significant difference. As a result,  NUDGE-M and NUDGE-N (which requires normalized embeddings) perform similarly.  The above discussion, together with a more general discussion on impact of label distribution (in response to W2) is now included in Appendix E.12.
> > > >
> > > > **Question Q4**. “The criteria for selecting the constraint boundaries in NUDGE-M …. are not fully detailed“
> > > >
> > > > **Response**. As we now discuss in Sec. 3.3 (with additional details in newly added Appendices C1 and E.6), we use an L2 norm constraint because (1) our embedding space is dense, and L2 norm provides dense embedding updates and (2) the constraint allows for finding efficient optimal solutions. Our experiments show that NUDGE-M constraints do perform better than alternatives, e.g., when considering sparse constraints, or when using adaptive thresholds to increase constraint flexibility but sacrificing optimality. We next present the above two arguments in more detail (please also see our response to W1).
> > > >
> > > > First, because our focus is on dense vector embeddings, our constraints in NUDGE-M use L2 norm to bound the embedding change. In a dense embedding space, we expect embedding dimensions to be dependent on each other, and thus dense updates to the embeddings will be more suitable. This contrasts with using L1 norm, which leads to sparse embedding updates. Our experiments confirm this, showing significantly more benefit to using L2 norms compared with L1 norm constraints on all datasets except ArguAna. Nonetheless, sparse updates do perform better on ArguAna. This suggests that L2 norm constraints perform better in general and across datasets, but alternative constraints can also be used in some cases to improve accuracy.
> > > >
> > > > Second, an important consideration in designing the constraints in NUDGE is allowing efficient optimal solutions (recall that the unconstrained optimization problem was NP-hard). Our experiments show that allowing more flexibility, such as using adaptive thresholds in our constraints but sacrificing optimality can lead to a worse solution. The specific constraint in NUDGE-M is chosen as it allows for an efficient optimal solution, which also provides significant accuracy boosts in practice.
> > > >
> > > > **Question Q4 Continued**. “the specific reasoning behind the grid search implementation for NUDGE-N's gamma optimization are not fully detailed.”
> > > >
> > > > **Response**. As the newly added experiments in Fig. 2 show (also see our response Q1), NUDGE-N is not very sensitive to $\gamma$, and similar values of gamma work well across datasets. This presents a suitable setting for using grid search, since a search over only a few values for $\gamma$ in a small range allows for finding a good $\gamma$ value.  We have added this reasoning behind our use of grid search in Sec. 3.2.

---

> ### Author Response · Authors · 2024-11-21
>
> **Question Q5**. "How to mitigate the impact of unreliable training query embeddings when fine-tuning based on these embeddings?"
>
> **Response**. NUDGE does not modify query embeddings, and uses the given query embeddings to fine-tune data embeddings. As the reviewer suggests, the quality of the query embeddings may impact the benefits obtained by NUDGE. We investigated this in two settings, where we show that (1) if unreliable query embeddings are due to random noise, NUDGE is surprisingly robust, providing benefits even when the scale of noise is larger than the scale of query embeddings (see newly added Appendix E.7); and (2) if unreliable query embeddings are not caused by the embedding model but instead due to poor query formulation, NUDGE can be combined with query rewriting methods to mitigate its impact (see newly added Appendix E.8). We provide a summary of the newly added experiments below.
>
> **Summary of discussion in Appendix E.7 and  E.8**. First, we investigated how noisy training query embeddings impact the accuracy of fine-tuning. We added i.i.d Gaussian noise from $\mathcal{N}(0, \sigma^2)$ to training query embeddings, for various values of $sigma$. The noise is added independently to each dimension of training and validation query embeddings. The result of this experiment is shown in Fig. 5 in Appendix E.7, where we see that both NUDGE variants still outperform No Fine Tuning, even until noise with standard deviation (std) 0.2. Note that noise with std of 0.2 has average absolute value of around 0.16, while the average absolute value of the training query embeddings across all datasets is less than 0.04, showing that the fine-tuning process is robust to some amount of noise. Interestingly, our results also show that NUDGE-M is more robust to noise than NUDGE-N, where NUDGE-M outperforms NUDGE-N for noise with standard deviation larger than 0.05. We believe this can be because normalization in NUDGE-N forces the embeddings to fall in a smaller subspace. This can force unrelated embeddings to become more entangled in the presence of noise, leading to worse accuracy.
>
> Next, we also consider the scenario that unreliable query embeddings exist due to poor query formulation. In such cases, existing work uses query rewriting techniques to mitigate the impact of poor query formulation. We show that such approaches can be used together with NUDGE out-of-the box, further improving accuracy. This is because NUDGE only modifies data embeddings, and is orthogonal to any query rewriting method. We explore using one such approach in the newly added Appendix E.8, where we discuss how NUDGE can be used together with HyDE, a popular query rewriting method. Our results in Table 16 show that incorporating HyDE within NUDGE improves the accuracy over using NUDGE alone (although NUDGE alone is more accurate compared with using HyDE without NUDGE).  This shows that NUDGE can be combined with query rewriting methods to mitigate the impact of unreliable query embeddings.

---

> ### Author Response · Authors · 2024-12-04
>
> Dear Reviewer,
>
> As we have discussed in our previous comments, we have thoroughly addressed all your concerns mentioned in your review and have accordingly made modifications to the paper, with new experiments added based on your suggestions. We would appreciate it if you could acknowledge the changes and how they address your concerns.
>
> The Authors

---

### Meta-Review · Area_Chair_MiTw · 2024-12-20

**Metareview:**

Thanks for your submission to ICLR.

This paper received borderline reviews, with two positive and two negative reviews.  Some of the issues raised in the original reviews included i) missing experimental results (e.g., ablation studies), ii) missing details throughout the paper, iii) questions about the theoretical results, iv) inability to apply out-of-sample data, v) generalizability.

The authors provided an extensive rebuttal to all of the points raised by the reviewers, including adding several new results.  I read through these carefully, along with the paper.  One of the negative reviewers never responded to the author rebuttal, though in my opinion the rebuttal clarified most of the issues raised by this reviewer.  The other negative reviewer seemed largely concerned that the paper was unnecessarily complicated (e.g., they felt the NP-hardness results were not necessary).  I disagree with the reviewer on this point, and in general agree with the authors that the weaknesses raised by this reviewer were adequately addressed by the rebuttal.  Further, the other reviewers maintained their positive score, and furthermore, the authors adequately addressed the issues in those reviews as well.

So, overall, I am willing to advocate for accepting this paper, as I think the authors have done a good job in rebutting the reviews and responding to the various concerns raised.

**Additional Comments On Reviewer Discussion:**

See above.  There were two positive and two negative reviews, but in the end one of the negative reviewers did not engage (but their concerns were adequately addressed by the authors), and the other negative reviewer, while not ultimately willing to advocate for the paper, had concerns that were also adequately addressed by the authors.

---

### Decision · Program_Chairs · 2025-01-22

Accept (Poster)